# CrossPL: Systematic Evaluation of Large Language Models for Cross Programming Language Interoperating Code Generation

**Zhanhang Xiong**[1,2]   **Dongxia Wang**[1,2,*]   **Yuekang Li**[3]   **Xinyuan An**[1,2]   **Wenhai Wang**[1]
[1]Zhejiang University   [2]Huzhou Institute of Industrial Control Technology
[3]The University of New South Wales

## Abstract

Large language models (LLMs) have shown strong performance in single-language code generation, but how well they produce cross-programming-language (CPL) interoperating code, which is widely used in cross-platform and complex software systems, remains underexplored. Therefore, a benchmark for evaluating CPL interaction code generation is essential. However, Constructing such a benchmark is challenging owing to sparse interoperating code in real-world multi-programming-language projects, diverse Inter-process Communication (IPC) mechanisms, vast Foreign Function Interface(FFI) language pairs, and the difficulty of evaluation. To address this gap, we introduce CrossPL, the first benchmark for systematically assessing LLM performance of CPL code generation across two primary interoperation modes and 2534 tasks, specifically 1,982 IPC tasks spanning six languages and 522 Python–C FFI tasks. Its construction involved a review of CPL documentation, 156 finite state machines, and analysis of 19,169 multi-language GitHub repositories. Two LLM-based workflows are designed for automating the benchmark construction and evaluation, and assess 20 state-of-the-art LLMs. Results reveal clear limitations: the best model achieves only 19.5% Pass@1 and 26.46% Pass@5 on the FFI subset, in sharp contrast to the strong performance of these models on single-language benchmarks. These findings underscore the urgent need for improving LLMs regarding CPL interoperating code generation. The benchmark and code are available at `https://github.com/newxzh/crosspl`.

## 1 Introduction

Large Language Models (LLMs) have recently attracted substantial attention, showing impressive capabilities in natural language processing and code generation tasks (Jin et al., 2024; Lozhkov et al., 2024). In particular, using LLMs for programming has emerged as a promising direction, with many software engineers already adopting them as auxiliary tools for software development (Aguiar et al., 2024).

To evaluate the programming ability of LLMs, researchers have proposed a variety of benchmarks focusing on single programming language code generation (Chen et al., 2021; Austin et al., 2021; Cao et al., 2024; Du et al., 2024; Xue et al., 2024). On these tasks, LLMs demonstrate strong performance, particularly in generating syntactically valid and functionally correct code(Lozhkov et al., 2024; Jin et al., 2024; Islam et al., 2024).

However, in modern software system development, the use of multiple programming languages (MPL) has become increasingly common, both in industry and open-source projects. Studies show that over 80% of software systems involve two or more programming languages (Tomassetti & Torchiano, 2014). The rationale lies in exploiting the unique strengths of different languages to enhance performance, modularity, and scalability (Li et al., 2024; Delorey et al., 2007; Ray et al., 2014; Li et al., 2021; Grichi et al., 2020). Consequently, it is crucial to extend evaluation beyond single-language tasks and towards cross-programming-language (CPL) settings.

---

*Corresponding author: dxwang@zju.edu.cn

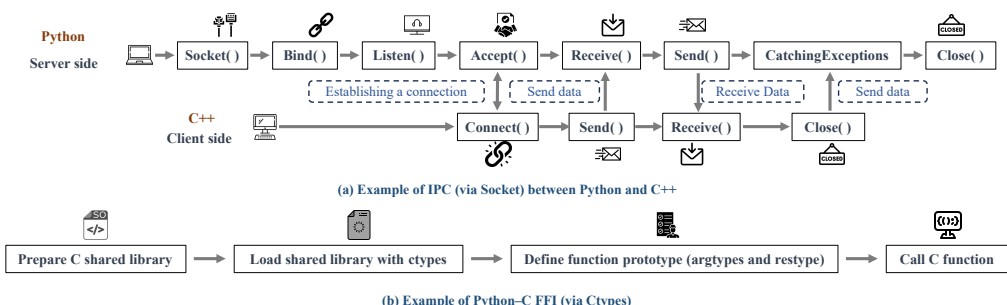

Figure 1: Examples of CPL interoperating (IPC and FFI)

Despite this demand, existing benchmarks for evaluating MPL-related code generation are scarce. More importantly, the few available works (Zheng et al., 2023; Cassano et al., 2023; Xu et al., 2024) primarily focus on code translation across languages, rather than assessing LLMs' ability to generate interoperating code across heterogeneous languages. This leaves a critical research gap: there is still no benchmark specifically designed to evaluate LLMs' capacity to generate code involving CPL collaboration.

In real-world MPL systems, such collaboration is typically enabled by two mechanisms: Inter-process Communication (IPC) and Foreign Function Interface (FFI) (Yang et al., 2024a; Li et al., 2022b;a). IPC facilitates communication between independent processes using techniques such as *Sockets*, *gRPC*, *message queues*, and *Websocket*, requiring careful handling of data serialization, synchronization, and protocol compliance to avoid deadlocks or message loss. FFI, in contrast, enables direct invocation of functions across languages within the same process by managing function signatures, type conversions, and memory layouts, where even minor mistakes may lead to undefined behavior (Yang et al., 2024b). Fig. 1 provides representative examples of IPC and FFI in MPL systems[1].

Constructing a benchmark that incorporates both IPC and FFI poses several challenges: ❶ In real-world MPL software systems, CPL interaction code snippets are typically sparse and implicitly embedded within large codebases, making it difficult to accurately detect the boundaries of CPL interoperation. Manual collection is time-consuming and incurs substantial overhead. ❷ IPC encompasses a wide range of mechanisms, including but not limited to *pipe*, *message queue*, *gRPC*, *TCP*, *UDP*, *HTTP* and *Websocket*, each exhibiting distinct characteristics, usage patterns, and programming paradigms (Yang et al., 2024a). This diversity adds complexity to interface categorization and identification. ❸ Moreover, constructing an FFI benchmark is particularly challenging, because the space of possible language combinations is vast[2] (Yang et al., 2024b; Li et al., 2022a; 2023; Bae et al., 2019; Hwang et al., 2021; Bai et al., 2018) , making exhaustive coverage infeasible and necessitating careful selection of representative cases. ❹ MPL projects are typically large-scale, involving complex dependency environments, making direct code execution often infeasible.

To address these challenges in benchmark construction, we propose three core strategies: ❶ We designed two LLM-based workflows to support both the construction and evaluation of the benchmark. ❷ We conduct comprehensive interface analysis by reviewing official documentation of mainstream IPC technologies and designing 156 scenario-specific finite state machines (FSMs) for real-world IPC pattern detection. ❸ As both Python and C are among the top ten most widely adopted programming languages, C offers exceptional performance and memory management, while Python is lauded for its simplicity and readability (Yang et al., 2024b). This combination provides access to a vast ecosystem of libraries and modules essential for tasks in machine learning and artificial intelligence (e.g., Numpy(Harris et al., 2020), SciPy(Virtanen et al., 2020), Pytorch(Paszke et al., 2019), Tensorflow(Abadi et al., 2016), etc.). Consequently, we constructs a set of FFI-related tasks specifically within the context of Python-C interoperation. ❹ For IPC tasks, we employ the cor-

---

[1]Fig. 1 illustrates representative examples for the two mechanisms of CPL interoperability. The upper part depicts the workflow of IPC between Python and C++ using sockets, while the lower part shows the key steps for Python invoking low-level C functions via *ctypes*.

[2]High-level languages (e.g., Python, Java, Ruby, Go) interact with C/C++ via FFI to leverage the high-performance capabilities of low-level modules.

responding FSMs as static analysis tools to verify protocol compliance. For FFI tasks, focusing on Python-C as the target setting, we carefully construct controlled compilation and execution environments together with assertion-based test cases to validate both the syntactic correctness and functional soundness of the LLM-generated code.

Following these strategies, we present CrossPL, the first benchmark for systematically evaluating LLM performance in CPL interoperating code generation, comprising 2,534 tasks. It encompasses two subsets: CrossPL-IPC with 1,982 tasks spanning six languages, and CrossPL-FFI with 522 Python–C tasks. Furthermore, we raise three key research questions (RQs) as follows:

- **RQ1**: How effectively can LLMs generate complete IPC code covering key state transitions, and what are the error distributions in cases of failure?

- **RQ2**: How effectively can LLMs generate FFI-based CPL interoperating code, and what are the error distributions in cases of failure?

- **RQ3**: How do model characteristics impact the CPL interoperating code generation performance of LLMs?

We systematically investigate these RQs and have the following key findings: ❶ LLMs tend to underperform on most tasks involving different programming languages, and their capabilities vary significantly across different subsets of programming languages; ❷ Similarly, LLMs show limited effectiveness in generating code for various IPC techniques, with substantial differences in performance depending on the specific technique; ❸ The performance of LLMs on the CrossPL-FFI subset is notably poor, with the best-performing model achieving only 19.50% at Pass@1 and 26.46% at Pass@5. However, many LLMs have already surpassed 90% pass@1 on single language code generation benchmarks[3]. ❹ Recent advances in model thinking techniques do not significantly improve the accuracy of IPC code generation by LLMs and, in some cases, even perform worse than the base models. These findings reveal the significant inadequacy of existing LLMs in generating CPL interoperating code compared with their performance on single-language code generation.

Our main contributions can be summarized as:

- **CrossPL benchmark:** To the best of our knowledge, we propose CrossPL as the first benchmark designed to evaluate LLMs' ability to generate CPL interoperating code involving IPC and FFI. CrossPL encompasses two subsets: CrossPL-IPC with 1,982 tasks spanning six languages, and CrossPL-FFI with 522 Python–C tasks.

- **Methodology for Automated Benchmark Construction:** We propose a unified methodology for automated benchmark construction, combining FSM-based IPC interface characterization and LLM-based workflows. First, we designed 156 FSMs based on official CPL interface specifications to formally characterize IPC-based interaction interfaces. These FSMs enable automatic detection and extraction of IPC code snippets from real-world GitHub repositories. They also serve as a tool to evaluate LLMs' capability in generating CPL code under specific IPC scenarios. Second, we developed two LLM-based workflows: one leveraging FSMs to identify, extract, and validate IPC snippets, followed by instruction generation and performance evaluation; the other focusing on constructing FFI tasks and systematically assessing LLMs' effectiveness on FFI code generation.

- **Large-scale empirical study:** We evaluate 20 representative LLMs to answer the key question: whether LLMs can accurately generate cross-language interoperating code. The findings highlight the need for more dedicated effort in this critical yet underexplored area.

## 2 RELATED WORK

**Single Language Code Generation Benchmarks.** Early benchmarks, such as BLEU (Rabinovich et al., 2017) and CodeBLEU (Ren et al., 2020) focus on surface or structural similarity, but fail to capture functional correctness. HumanEval (Chen et al., 2021) introduces Pass@k for functional evaluation, which has become a standard metric. However, these benchmarks primarily target

---

[3]EvalPlus leaderboard (`https://evalplus.github.io/leaderboard.html`) is an open, continuously updated ranking that evaluates and compares LLMs' single programming language code generation performance across expanded and more rigorous benchmarks.

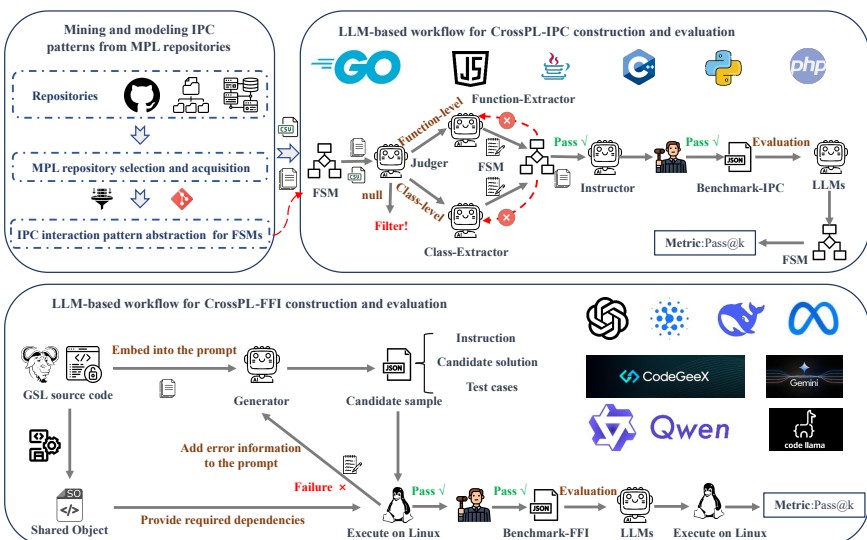

Figure 2: Framework for CrossPL construction and evaluation.

single-language, function-level tasks, mostly in Python (e.g. MBPP(Austin et al., 2021) and HumanEval(Chen et al., 2021)). As evaluation expands, ClassEval (Du et al., 2024) CodeAgentBench (Zhang et al., 2024a) assess class-level and repository-level performance. SWE-bench (**?**) evaluates real-world software engineering problem solving by requiring models to generate patches that correctly fix issues in large, open-source repositories. However, ClassEval, CodeAgentBench and SWE-bench are all still within Python. Domain-specific benchmarks in data science (Huang et al., 2024; Zhang et al., 2024b; Nejjar et al., 2025), embedded systems (Englhardt et al., 2024), robotics (Hu et al., 2024), and other areas (Zhu et al., 2025) are likewise restricted to a single language in their respective fields.

**Multilingual Code Generation Benchmarks.** While early efforts focus on single-language settings, recent work shifts toward multilingual code generation. Xu et al. (Zheng et al., 2023) introduce HumanEval-X, extending evaluation to five programming languages. Cassano et al. (Cassano et al., 2023) propose MultiPL-E, translating HumanEval and MBPPAustin et al. (2021) tasks into 18 languages to analyze multilingual performance. CRUXEVAL-X (Xu et al., 2024) further assesses LLMs' reasoning and generation across languages. HumanEval-XL (Peng et al., 2024) expands this effort, linking 23 natural and 12 programming languages to create a large-scale multilingual benchmark. Multi-SWE-bench (Zan et al., 2025) extends SWE-bench to multilingual settings, but still evaluates bug fixing within single-language projectsand few CPL interaction problems.

Although these benchmarks span multiple programming languages, they evaluate each language in isolation rather than testing CPL interoperability. As a result, it remains unclear whether LLMs can reliably generate interoperating CPL code. To the best of our knowledge, no existing benchmark directly assesses the model capability of CPL interoperability.

## 3 METHODOLOGY

Fig.2 presents an overview of the core pipeline of this work, spanning from code analysis and collection to code generation and evaluation. The framework consists of three main components: (1) Mining and modeling IPC interoperating patterns from MPL repositories; (2) LLM-based workflow for CrossPL-IPC construction and evaluation; (3) LLM-based pipeline for CrossPL-FFI construction and evaluation.

### 3.1 MINING AND MODELING IPC INTEROPERATING PATTERNS FROM MPL REPOSITORIES

**Repository Selection and Localization for CPL Analysis.** We begin by curating 19169 MPL projects from GitHub, each with 1000-30000 stars and employing 2–5 programming languages. For

each project repository, we record the key metadata $\mathcal{R}$ (e.g., ID, name, star count, language set, description, timestamps) and store it in a structured CSV format. Using the provided clone URLs, we locally replicate all the repositories via Git to enable downstream analysis and extraction of IPC interoperating code.

**Modeling CPL interoperating Patterns.** The main advantages of FSMs in clear structure and logic make them well-suited for managing complex processes with distinct stages and transitions. They thus are particularly suitable for analyzing IPC workflows, where interactions typically involve well-defined states (e.g., initialization, data transmission, termination) and deterministic transitions between them. Fig. 5 illustrates an example of modeling CPL interoperating using FSMs, a detailed explanation of this example is given in subsection A.1.

PolyFax (Li et al., 2022a) first proposes leveraging FSMs to identify and classify interoperating patterns in MPL projects. However, it defines only 8 IPC-related FSMs, whose state patterns are overly coarse, often contain few states, and rely on fuzzy matching. This can lead to misclassification, especially since the FSMs do not account for code comments, making them prone to incorrectly matching commented-out code. In this study, after analyzing extensive documentation and real-world MPL projects, we design a suite of 156 FSMs. The detailed distinctions between the FSMs we designed and those developed in PolyFax are provided in the subsection A.2.

The entire workflow of IPC patterns modeling, CrossPL-IPC construction and evaluation is formalized in Algorithm 1.

## 3.2 LLM-BASED WORKFLOW OF CROSSPL-IPC CONSTRUCTION AND EVALUATION

This component aims to identify IPC interoperating instances across the repositories using the 156 FSMs, generate their corresponding instructions and also to append the validation FSM of the corresponding task, for CrossPL-IPC construction. For each matched instance, we record the file path $p$, interaction type $\tau$, interaction technique name $\theta$, programming language $L$, FSM identifier $\sigma$, and key procedural steps $K$.

To reduce the manual load and the limitations of single LLM in handling complex tasks with long contextual dependencies, we develop a LLM-based workflow powered by the DeepSeek-V3 (Liu et al., 2024)[4] to construct CrossPL-IPC. Input code instances identified by FSMs as containing IPC techniques are first validated by the *Judger* to confirm IPC implementation and classify them as function-level or class-level. Subsequently, instances are processed by the corresponding *Function Extractor* or *Class Extractor*, which extract minimal, logically complete IPC code snippets guided by technique-specific descriptions and one-shot examples. Extracted snippets undergo FSM-based validation, with up to five extraction attempts at increasing temperature if validation fails. Validated snippets are then passed to the LLM-based *Instructor* to generate natural language task descriptions.

Though currently a common practice[5], to ensure the effectiveness of such generated descriptions, we randomly sample and manually inspect a subset of them, confirming that they do clearly and accurately describe the intended tasks. Furthermore, the benchmark's task information, comprising metadata, code, and the generated instructions, are stored in structured JSON format.

Finally, We provide the instruction corresponding to each task in the benchmark to an LLM under evaluation, prompting it to generate the code for the respective functionality. Correctness is assessed by matching the generated code against the predefined FSMs to ensure protocol compliance. The use of FSMs is particularly advantageous here, as they not only offer a formal and modular way to represent the state transition logic of IPC technologies but also effectively capture subtle protocol violations or misuse patterns in the generated code.

The entire workflow is formalized in Algorithm 2. Figs 6-9 present the prompts of all LLMs in the workflow; detailed information is provided in subsection A.3.

---

[4]Selected for its state-of-the-art performance among open-source LLMs available prior to April 2025 and also its cost-effectiveness.

[5]The feasibility of LLM-based code instruction generation has been shown in (Zhu et al., 2025) and (Xie et al., 2024).

### 3.3 LLM-BASED WORKFLOW OF CROSSPL-FFI CONSTRUCTION AND EVALUATION

Algorithm 3 illustrates the construction of the CrossPL-FFI for Python-C external function calls. The underlying C code is sourced from the GNU Scientific Library (GSL)(Gough, 2009), a widely used and self-contained library of mathematical and statistical functions. GSL is chosen because real-world Python-C FFI scenarios involve complex dependencies, making large-scale execution environments difficult to reproduce, whereas GSL provides stable, compilable C code suitable for benchmark construction.

The workflow begins by compiling the GSL library into shared object (.so) files using Autotools and Make, establishing the runtime environment. C source files are then cleaned and applied using an initial FFI prompt and an error-revision prompt. Execution of the candidate solution is performed in the environment where the precompiled .so files are available for FFI calls; successful executions are saved as benchmark entries, while failures are iteratively refined via the LLM (powered by Deepseek-V3(Liu et al., 2024)). This approach ensures a scalable, reproducible, and controlled benchmark for assessing LLMs' ability to generate correct Python-C FFI code.

Additionally, key information from the canonical solution, including class names, function names, and parameter names, is incorporated into the "Instruction" field of the benchmark. To ensure the effectiveness of such generated descriptions, we randomly sample and manually inspect a subset of them, confirming that they clearly and accurately describe the intended tasks. Finally, these benchmark entries are provided as tasks to the LLMs under evaluation. The outputs from the LLMs are combined with automatically generated assertion test cases to verify correctness, enabling systematic execution and testing. This approach ensures a scalable, reproducible, and controlled benchmark for assessing LLMs' ability to generate correct Python-C FFI code. Figs 10-13 present the prompts of all LLMs in the workflow; detailed information is provided in subsection A.4.

### 3.4 STATISTICS OF CROSSPL BENCHMARK

CrossPL comprises 2,534 tasks for systematically evaluating LLMs' CPL code generation performance. CrossPL encompasses two subsets: CrossPL-IPC with 1,982 tasks spanning six languages and seven IPC technologies, and CrossPL-FFI with 522 Python–C tasks. Figs. 14-16 present the statics of the CrossPL from different view. Additional details regarding Figs. 14-16 are provided in Appendix B. Table 1 and 2 shows the detailed statistics on the number of samples retained after each key filtering step in CrossPL-IPC and CrossPL-FFI construction pipeline respectively.

## 4 EXPERIMENT

We design experiments to answer three research questions:

- **RQ1**: How effectively can LLMs generate complete IPC code covering key state transitions, and what are the error distributions in cases of failure?

- **RQ2**: How effectively can LLMs generate FFI-based CPL interoperating code, and what are the error distributions in cases of failure?

- **RQ3**: How do model characteristics impact the CPL interoperating code generation performance of LLMs?

### 4.1 EXPERIMENTAL SETTINGS

In our evaluation, we adopt the unbiased Pass@k[6] metric (Chen et al., 2021) to precisely measure the functional correctness of code snippets generated by LLMs. Consistent with previous studies (Zhu et al., 2025), we report both Pass@1 and Pass@5 results under a zero-shot setting and use macro-averaging to calculate the overall scores. For Pass@1, we employ greedy decoding by setting the temperature to 0. For Pass@5, we set the temperature at 0.2 and top-p sampling at 0.95. For proprietary models and large-scale open-source models, we perform inference using their official

---

[6]pass@k $:= \mathbb{E}_{\text{Problems}} \left[ 1 - \frac{\binom{n-c}{k}}{\binom{n}{k}} \right]$, where $n$ is the total number of elements, $c$ is the number of correct elements and $k$ is the number of top-ranked predictions. $n$ is set at 10 in pass@5.

APIs. For open-source models such as LLaMA3-8b-Instruct (Grattafiori et al., 2024), Gemma-7b (Team et al., 2024b), CodeLLaMA-7b-Instruct (Roziere et al., 2023), and CodeGemma-7b (Team et al., 2024a), we deploy them locally using Ollama on the NVIDIA RTX 3090 GPU.

## 4.2 IPC CODE GENERATION PERFORMANCE (RQ1)

**Overall findings.** Fig. 3a and Fig. 3b show that LLMs varying effectiveness in generating IPC code across different programming languages and IPC techniques. Generally, larger and domain-specialized models perform better. LLMs exhibit the best performance in generating IPC code written in C++, while their performance is relatively weaker when generating IPC code based on Go. This could be due to Go's reliance on struct-based abstractions, whereas LLMs are predominantly trained on class-based languages, leading to structural mismatches in Go's IPC code generation. Regarding IPC techniques, higher-level protocols such as *gRPC* perform best, while lower-level protocols like *Pipe* and *HTTP* exhibit weaker results due to their minimal abstraction and system dependence. In terms of error distribution, Taking GPT-4o and Llama3-8b-instruct as examples, GPT-4o demonstrates relatively balanced state coverage, with errors primarily occurring in IPC protocol link configuration and data transmission setup. In contrast, Llama3-8b-instruct is worse, it encounters errors during library file configuration in many cases, preventing effective state machine matching and resulting in incomplete IPC code generation.

All experimental results related to RQ1 are summarized in Tables 3-6, with detailed analyses provided in the subsection C.1. Futhermore, a detailed case study of CrossPL-IPC is also provided in the subsection D.1.

> **Answer to RQ1**
> LLMs generate IPC code with varying effectiveness, performing better with C++ and *gRPC* and worse with Go and low-level protocols like *Pipe*. Failures for models like GPT-4o often stem from protocol and data transmission setup, while Llama3-8b-instruct frequently fails earlier at library configuration.

## 4.3 FFI CODE GENERATION PERFORMANCE (RQ2)

**Overall findings.** From Fig. 3c and the Table 7, we find that LLMs perform unsatisfactorily in generating Python-C FFI code. Even the strongest model, GPT-4o, achieves only 19.54% in Pass@1, while Llama3-8b-instruct performs worst with a Pass@1 of merely 0.77%. Although increasing the number of sampled outputs leads to marginal improvements, the results remain far from satisfactory: GPT-4o reaches only 26.46% in Pass@5, and Llama3-8b-instruct achieves 3.95%. To better understand these limitations, we conducted an error analysis of GPT-4o's outputs and categorized the failures into six types: (1) symbol resolution errors; (2) GSL runtime errors; (3) Python-level calling errors; (4) NameError/undefined symbols; (5) assertion/test failures; and (6) memory/crash errors. These error categories highlight the fundamental challenges that constrain the success rate of LLMs in generating reliable Python-C FFI code and underscore the need for more robust reasoning and domain-specific knowledge integration in future model development. We infer the main reasons are that LLMs have insufficient reasoning capability when generating FFI code, lack domain-specific knowledge about CPL interoperating, and have limited understanding of low-level code. This results in frequent symbol resolution failures, runtime errors, calling mistakes, undefined symbols, test assertion failures, and memory or crash errors.

The detailed analysis of the data in Table 7 can be found in subsection C.2. Detailed case studies and error distribution analysis for CrossPL-FFI are presented in subsection D.2 and subsection D.3.

> **Answer to RQ2**
> LLMs struggle with FFI-based CPL code generation (GPT-4o 19.54% Pass@1; Llama3-8b-instruct <1%), with failures dominated by symbol resolution, runtime, calling, and memory errors.

## 4.4 IMPACT OF MODEL CHARACTERISTICS (RQ3)

Our experiments with Qwen3 series models yield two main observations. First, think mode markedly improves performance on CrossPL-FFI tasks (Fig. 4c), indicating that explicit intermediate reasoning helps manage the complexities of external library linking, type specification, and memory oper-

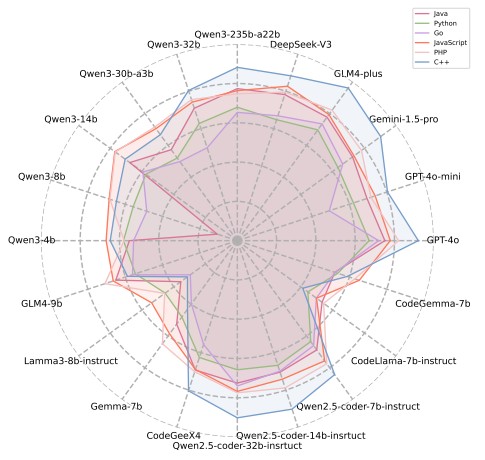

(a) Radar Plot of Pass@1 on CrossPL-IPC (programming languages view)

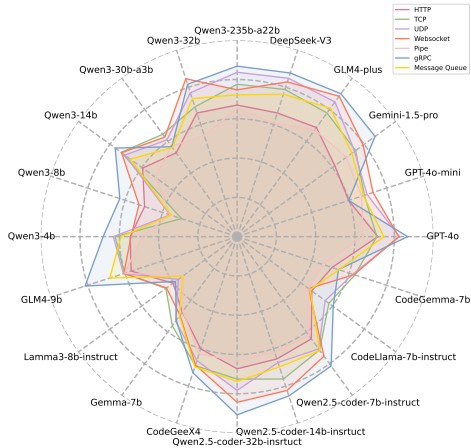

(b) Radar Plot of Pass@1 on CrossPL-IPC (IPC technologies view)

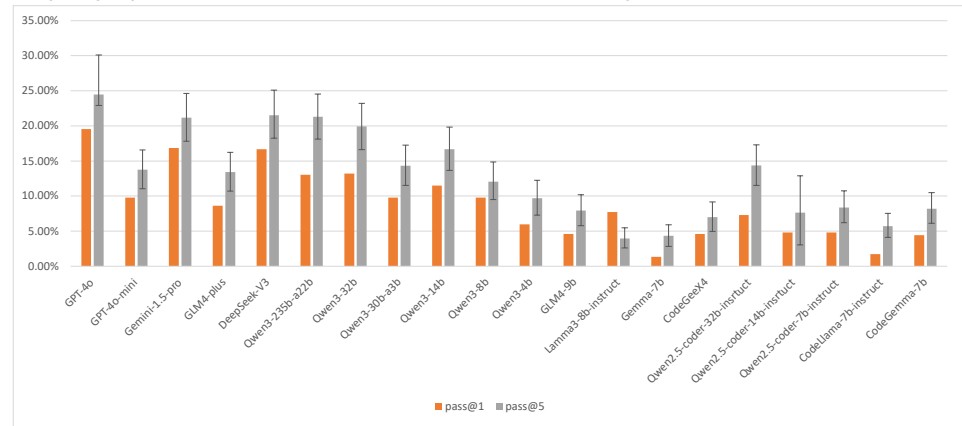

(c) Pass@1 and Pass@5 results of various models on CrossPL-FFI.

Figure 3: Performance of LLMs on CrossPL

ations inherent to FFI. Second, such improvements are not evident for CrossPL-IPC tasks (Fig. 4a, 4b); in some cases, performance even degrades. This may be attributed to IPC's reliance on well-structured communication patterns (e.g., sockets, queues), which are sufficiently captured by the training corpus and do not benefit from additional reasoning steps. Overall, Qwen3 performs better on CrossPL-IPC, while think mode proves particularly beneficial for reasoning-intensive FFI scenarios. The detailed analysis of the data in Tables 8–10 can be found in  subsection C.3.

> **Answer to RQ3**
> Model characteristics, such as think mode, improve performance in reasoning-intensive FFI tasks but have limited or even negative impact on IPC tasks, which rely on well-structured communication patterns.

## 5 DISCUSSION

**Effects of Model Scale in the Two Benchmark Pipelines:** In the CrossPL-IPC benchmark construction pipeline, smaller models could in principle substitute for the "Judger" and "Instructor", since these components require shallow reasoning and few output tokens. However, smaller models exhibit significantly higher false-positive rates, which propagate through the pipeline and dramatically inflate unnecessary extraction attempts. For efficiency and precision, a stronger "Judger" remains preferable. The CrossPL-FFI benchmark pipeline imposes much greater cognitive de-

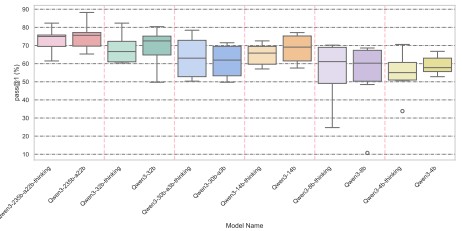

(a) Boxplot of Pass@1 performance of Qwen3 series models across programming languages.

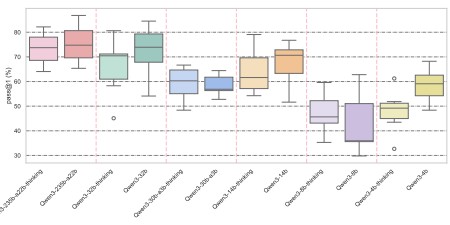

(b) Boxplot of Pass@1 performance of Qwen3 series models across IPC techniques.

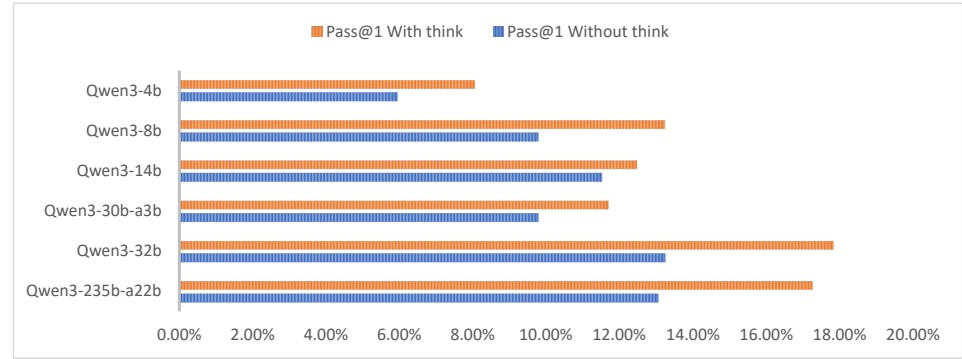

(c) Performance of Qwen3 series models on CrossPL-FFI (Pass@1)

Figure 4: Qwen3 series models on both CrossPL-IPC and CrossPL-FFI.

mands: analyzing low-level C code, generating tasks, producing reference solutions, and constructing assertion-based test cases. These steps benefit substantially from larger models with stronger instruction following and symbolic reasoning capabilities.

We conducted ablation studies to quantify the contribution of model scale. ❶ "Extractor"-only replacement: Replacing the "Extractor" with Qwen3-4B reduces the number of valid samples from 1,982 to 1,723 (–259). ❷ "Instructor"-only replacement: replacing only the Instructor with Qwen3-4B reduces samples from 1,982 to 1,890 (–72). ❸ "Extractor" replacement: Using Qwen3-4B as the "Extractor" causes a drastic drop in valid FFI samples: 522 to 65, confirming that high-quality FFI construction requires a strong, stable model. These findings clarify the differential roles of model scale across IPC and FFI pipelines. Further analysis can be found in C.4.

**Risks of Data Leakage in CPL Code Generation Benchmarks:** Because CrossPL-IPC samples are extracted from public GitHub repositories and For the CrossPL-FFI subset, the primary risk of data leakage arises from the fact that Python–GSL interactions are already implemented in existing open-source libraries (e.g., *pygsl*).

we analyzed potential data leakage through two experiments. ❶Temporal leakage in CrossPL-IPC: CrossPL-IPC samples were grouped by creation year: before 2022, 2022, 2023, and 2024–2025. The group of 2024-2025 contains 1,377 samples (69.48%). We evaluated Pass@1 using greedy decoding. As shown in Table 14 and Figure 20, there is no evidence that models systematically perform better on newer samples, indicating minimal temporal leakage.❷Training-data contamination in CrossPL-FFI: The FFI pipeline is designed to avoid exposure to existing code: the model receives an isolated C function and must independently generate the task description, ctypes wrapper, and reference solution. We further computed lexical overlap between our prompts and the *pygsl* codebase used for verification. The overlap is only 0.61% for function names and 0.99% for class names, indicating negligible contamination risk. Together, the two analyses show that data leakage does not meaningfully affect model performance. Further analysis can be found in C.5.

**Effects of Sampling Temperature and top-p on CPL Code Generation:** We examined the effect of temperature and top-p across eight locally deployable models on the CrossPL-IPC and CrossPL-FFI subsets (Tables 16- 19). There are three key observations: ❶Both sampling parameters affect model behavior, yet nonlinear. Performance varies irregularly across settings, and the optimal con-

figuration differs by model. ❷ IPC tasks are largely insensitive to sampling variation. Their inter-action patterns follow fixed communication protocols and deterministic state transitions, yielding a narrow solution space where stochasticity provides little benefit. Consequently, temperature and top-p adjustments lead to only marginal performance shifts. ❸ FFI tasks remain fundamentally constrained by low-level correctness. FFI requires strict adherence to function signatures, memory layouts and type coercions; minor deviations result in undefined behavior. Higher sampling diversity increases the likelihood of violating these constraints, while more conservative sampling does not substantively improve reliability. This explains the consistently large performance gap—for example, Qwen3-4B averages 55.24% on IPC but only 3.08% on FFI across temperatures, and Qwen2.5-coder-7B shows 68.05% on IPC but only 3.89% across top-p.

Overall, sampling strategies alone cannot overcome the intrinsic difficulty of FFI generation. Sub-stantial improvements will likely require stronger low-level reasoning ability, expanded CPL training data, and dedicated finetuning on FFI-oriented interactions. Further analysis can be found in C.6.

**Effects of Think Mode on CPL code generation Quality:** Section 4.4 shows that think mode of-fers limited benefit for IPC code generation and can even degrade performance, while substantially improving FFI tasks. This contrast stems from task characteristics: IPC tasks follow rigid procedural patterns with a narrow solution space, where additional reasoning may introduce unnecessary devi-ations. FFI tasks, by comparison, demand multi-step reasoning over function signatures, memory layout, and type semantics, making think mode inherently advantageous. Appendix D.4 provides a representative failure example of think mode in IPC tasks.

# 6 LIMITATIONS AND FUTURE WORK

**Limitations.** While this is the first benchmark tailored for assessing LLMs in CPL code generation, it has limitations: (1) The distribution of programming languages and IPC techniques in open-source projects is uneven, leading to limited samples in some sub-tasks; (2) To align with experimental settings in related work, we did not explore the impact of hyperparameter configurations on perfor-mance; (3) Since the Qwen3 series models encompass different parameter scales, we selected this series to evaluate the impact of think mode, while not extending the test to other models.

**Future Work.** Despite progress in LLM-based CPL code generation, our findings reveal several limitations and suggest future directions. First, LLMs exhibit suboptimal performance on FFI tasks due to insufficient reasoning and limited domain-specific knowledge about low-level code and li-brary linking, leading to symbol resolution failures, runtime errors, and memory crashes. Enhanc-ing models with explicit reasoning mechanisms, domain knowledge integration, and library-aware representations could mitigate these issues. Second, IPC code generation remains uneven across lan-guages and protocols: models perform well on C++ and high-level protocols like gRPC but struggle with Go and low-level mechanisms such as Pipe. Future work may explore specialized pretraining or fine-tuning strategies that strengthen LLMs' ability in underrepresented languages and system-level IPC patterns. Third, while model scale generally improves performance, gains are not uniform across tasks, indicating that architectural innovations or task-specific reasoning strategies may be required. Finally, systematic evaluation frameworks and benchmark expansion, including more di-verse CPL projects, protocols, and library interactions, are essential to better understand LLMs' strengths and limitations to guide their development toward robust CPL interoperability.

# 7 CONCLUSION

CPL interoperation code is prevalent in MPL projects and is essential for their construction. How-ever, current LLMs often fail to generate such code accurately, with performance being highly sen-sitive to both programming languages and interaction techniques. In particular, LLMs perform especially poorly on FFI tasks, where challenges such as external library linking, symbol resolu-tion, and memory management frequently lead to compilation or runtime failures. Moreover, the model's thinking mode provides limited benefit and may even negatively impact CPL code genera-tion. Enhancing the ability of LLMs to generate CPL interaction code, and especially to handle the complexities of FFI, will be critical for scaling up their application to the generation of large-scale MPL software systems in future work.

ACKNOWLEDGMENTS

This work is supported by the Huzhou Institute of Industrial Control Technology, China (Grant No. K-ZY-2024-009).

ETHICS STATEMENT

CrossPL is collected entirely from public repositories whose licenses permit software use, and our contributions fully comply with these license terms. During the collection or evaluation process, we do not collect any information about GitHub users, and CrossPL task instances only use GitHub data available via the public API and website. Our work does not involve any human subjects. CrossPL does not implicitly or explicitly use any discriminatory or biased heuristics in the selection of repositories.

REPRODUCIBILITY STATEMENT

We provide complete code and configuration files to fully reproduce all experiments. In section 3, we describe in detail the data collection process and evaluation methodology. In section 4, we present the full experimental settings. The code repository also includes scripts to regenerate all reported figures and results in the paper.

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

# A  ALGORITHM

## A.1  INTRODUCTION OF FSMS

As described in subsection 3.1, Fig. 5 illustrates an example of modeling CPL interoperating using FSMs, specifically modeling typical client-server interactions based on the *java.net*. The upper part of the diagram models the server-side workflow, beginning with the import of networking libraries, followed by creating a *ServerSocket*, accepting connections, establishing input or output streams,

and eventually closing the *Socket*. The lower part represents the client-side process, including socket creation, connection initiation, stream handling, and termination. These workflows converge at the data exchange stage, representing the communication interface. This example demonstrates how FSMs can model API usage patterns in real-world code, enabling structured and interpretable modeling of communication logic.

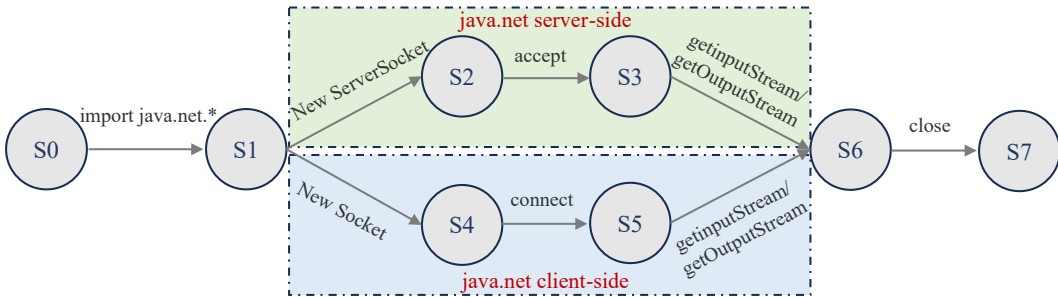

Figure 5: An example of FSM-modeled CPL interoperating.

## A.2 FSMs DIFFERENCES BETWEEN THIS PAPER AND POLYFAX

Compared to that of PolyFax, our models offer far greater state and transition granularity, enabling accurate capture of each critical step in diverse CPL interoperating scenarios. Furthermore, our FSMs include three key enhancements:

(1) Adding semantic descriptions for each key step of CPL interoperating code, providing contextual semantics for the LLMs (like for instruction generation) in benchmark construction workflow.

(2) Increasing the number of states to explicitly cover finer-grained IPC-specific steps, and designing clearer transition logic to reduce ambiguity in pattern detection.

(3) Skipping code comments to avoid false positives from comment text matching critical step patterns.

## A.3 LLM-BASED WORKFLOW OF CROSSPL-IPC CONSTRUCTION AND EVALUATION

To ensure the reproducibility and transparency of our benchmark pipeline, we provide the exact prompt templates employed across all LLMs in the CrossPL-IPC construction and evaluation workflow. As described in subsection 3.2, Figs. 6- 9 provide the detailed prompt information.

## A.4 LLM-BASED WORKFLOW OF CROSSPL-FFI CONSTRUCTION AND EVALUATION

To ensure the reproducibility and transparency of our benchmark pipeline, we provide the exact prompt templates employed across all LLMs in the CrossPL-FFI construction and evaluation workflow. As described in subsection 3.3, Figs. 10- 13 provide the detailed prompt information.

## B BENCHMARK STATIC

### B.1 DETAILED INFORMATION OF BENCHMARK

In constructing CrossPL-IPC benchmark, we conducted a thorough search and review of official documentation related to IPC technologies in CPL projects using keywords such as *gRPC*, *Pipe*, *message queue*, *TCP*, *UDP*, *WebSocket*, and *HTTP* across different programming languages.

Fig. 14a summarizes the distribution in CrossPL-IPC from different perspectives. Overall it covers six programming languages and seven IPC technologies, comprising a total of 1982 tasks. Among the programming languages, Java accounts for the highest proportion of IPC-related tasks with 615 instances (31.03%), whereas C++ the fewest, with 51 tasks (2.57%). Among IPC technologies,

---

**Algorithm 1:** IPC code analysis, extraction, generation and evaluation

---

**Input :**

$\omega_{\min}, \omega_{\max}$: project star count bounds;

$Ln_{\min}, Ln_{\max}$: number of languages per project bounds;

$\mathcal{P}$: all public GitHub repositories within bounds;

$\mathcal{FSM}$: A list of FSMs.

**Output:**

$True\_List$: LLM's correct task information list;

$False\_List$: LLM's wrong task information list

1  **Step 1: MPL Repositories Crawling**

2  $\mathcal{R} \leftarrow \text{Crawler}(\mathcal{P}, \omega_{\min}, \omega_{\max}, Ln_{\min}, Ln_{\max})$

3  save $\mathcal{R}$ to CSV                    ▷ Record metadata of satisfied MPL projects

4  **Step 2: Clone to Local**

5  **foreach** $r_i \in \mathcal{R}$ **do**

6     $local\_dir \leftarrow r_i.\text{name} + \_ + r_i.\text{ID}$

7     $\text{Clone}(r_i.\text{CloneUrl}, local\_dir)$

8                                 ▷ Clone to local with specific directory name

9  **Step 3: FSM-based Interoperability Analysis**

10  $\mathcal{F} \leftarrow \emptyset$

11  **foreach** $r_i \in \mathcal{R}$ **do**

12     **foreach** $\mathcal{FSM}_j \in \mathcal{FSM}$ **do**

13        **if** $\mathcal{FSM}_j(r_i)$ **then**

14           $(p_i, \tau_i, \theta_i, L_i, \sigma_i, K_i) \leftarrow \mathcal{FSM}_j(r_i)$

15                      ▷ Algorithm of FSM can find in (Li et al., 2022a)

16           $\mathcal{F}.append((p_i, \tau_i, \theta_i, L_i, \sigma_i, K_i))$

17           **break**

18        **else**

19           **continue**

20  **Step 4: Benchmark Construction (Algorithm 2)**

21  **Step 5: Benchmark Evaluation**

22  **foreach** $\mathcal{B}_i \in \mathcal{B}$ **do**

23     $code \leftarrow LLM(I_i)$

24     $\mathcal{V} \leftarrow \mathcal{FSM}(\sigma_i)$

25     **if** $\mathcal{V}(code)$ **then**

26        $True\_List.append(\mathcal{B}_i)$

27     **else**

28        $False\_List.append(\mathcal{B}_i)$

---

**Algorithm 2:** CrossPL-IPC Construction

**Input :**
$\mathcal{F} = \{(p_i, \tau_i, \theta_i, L_i, \sigma_i, K_i)\}_{i=1}^N$: metadata of CPL interoperating instances detected by FSMs;
$N$: number of IPC instances identified by the FSMs;
$p_i$: file path of instance $i$;
$\tau_i$: interaction type;
$\theta_i$: technique used;
$L_i$: programming language;
$\sigma_i$: FSM ID;
$K_i$: key steps of IPC.

**Output:** IPC benchmark set $\mathcal{B}$ saved in JSON format.

```
1  T_id ← 1                                              ▷ Initialize task identifier
2  foreach (p_i, τ_i, θ_i, L_i, σ_i, K_i) ∈ F do
3  │   C_i ← read(p_i)                               ▷ Read the code file of the instance
4  │   cl_i ← A_1(C_i, τ_i, θ_i, L_i, K_i)                              ▷ Judger
5  │   if cl_i == null then
6  │   │   continue
7  │   else
8  │   │   R_i ← select_reference(C_i, cl_i)
9  │   │                                      ▷ Select the required one-shot example
10 │   │   for k ← 0 to 5 do
11 │   │   │   t ← 0.1 · k                                   ▷ Set LLM temperature
12 │   │   │   c_k ← A_2(C_i, τ_i, θ_i, L_i, K_i, R_i, cl_i, t)
13 │   │   │                                                ▷ Code extracter
14 │   │   │   if A_3(c_k, σ_i) then
15 │   │   │                                            ▷ FSM-based validation
16 │   │   │   │   I_k ← A_4(c_k, τ_i, θ_i, L_i, K_i)
17 │   │   │   │                                             ▷ Instructor
18 │   │   │   │   B_k ← {T_id, p_i, τ_i, θ_i, L_i, σ_i, K_i, cl_i, I_k, c_k}
19 │   │   │   │   Save(B_k, save_path, T_id)
20 │   │   │   │   T_id ← T_id + 1
21 │   │   │   │   break
```

| Prompt template for "Judger" |
|---|
| You are a senior software engineer with deep expertise in multi-language interoperability and inter-process communication (IPC) technologies.

Your task is to analyze the provided "Source code", which is written in {PL}.

Please determine whether the LLM can accurately extract either function-level code, or class-level code, which contains a complete implementation of the {IPC_Name} technique.Judgment rules are as follows:
1.According to the "Source code" and the key steps of the inter-process communication technology, if the key steps of using this inter-process communication technology can be completed in one function, extract the function-level code;
2. According to the "Source code" and the key steps of the inter-process communication technology, if there are no functions or classes, and the key steps can be completed solely within the script-level code, please extract the code and encapsulate the code into a function-level code;
3. According to the "Source code" and the key steps of the inter-process communication technology, if the inter-process communication technique requires multiple functions to work together and involves defined classes, directly extract the class-level code;
4. According to the "Source code" and the key steps of the inter-process communication technology, if the inter-process communication technique requires multiple functions to work together, but the source code does not define any classes, please extract these functions and encapsulate these functions into class-level code;
5. According to the "Source code" and the key steps of the inter-process communication technology, if you are absolutely sure that you cannot extract suitable class-level code or function-level code from the source code to accurately implement the key steps of the inter-process communication technology, nothing will be extracted and "null" will be returned.

The key steps required for a valid {IPC_Name} implementation are:
{Steps}

Source code:
{Original_code}

Your judgment is: (Your answer can only contain one of the following: "Function-level" or "Class-level" or "null". Please don't include any other characters, and do not give me the reason for your judgment.) |

Figure 6: Prompt template for "Judger"

**Function-level code snippet extraction prompt template for "Function-Extractor"**

Task description:
Based on the original code and information related to the original code, extract the expected function-level code for multi-programming language interaction from the original code.

Requirements:
1. Only extract function-level code involving multi-programming language interaction in the original code, including function name, parameters, return value and core logic.
2. According to the original code, if there are no functions or classes, and the key steps can be completed solely within the script-level code, please extract the code and encapsulate the code into a function-level code, preserve the original semantics of the code as much as possible; do not introduce any content other than encapsulation
3. Extract all external library files and global variables that use the function, and ensure that these codes are completely derived from the original code.
4. Only extract task-related code, and the extracted code must strictly contain the key steps of the inter-process communication technology.
5. The output result should follow the format of "Output" in the example. Do not include any characters other than code, including "Output", "Python", " "" ", or "Here's the extracted...".
6. Please strictly follow the instructions above.

Instruct:
You are an advanced programmer who focuses on multi-language interaction and inter-process communication technology.
Please complete the task according to the following example.

Example:
Input:
Original code: {Reference_raw_code}
Information related to the original code:
  - Programming language: Python
  - Inter-process communication technology name: TCP Server - side by using socket in python
  - The key steps of this inter-process communication technology:
    Step 1: Import the socket module
    Step 2: Create a TCP socket object using IPv4
    Step 3: Connect the socket to the target server's IP and port
    Step 4: Send data to or receive data from the server
    Step 5: Close the connection with the server

Output:
{Reference_function_level_code}

Request:
Now please follow the task description and the example to extract one expected multi-language interaction function-level code from the original code based on the input.

Input:
Original code: {Original_code}
Information related to the original code:
  - Programming language: {PL}
  - Inter-process communication technology name: {IPC_Name}
  - The key steps of this inter-process communication technology: {Steps}

Figure 7: Function-level code snippet extraction prompt template for "Function-Extractor"

**Class-level code snippet extraction prompt template for "Class-Extractor"**

Task description:
Based on the original code and information related to the original code, extract the expected class-level code for multi-programming language interaction from the original code.

Requirements:
1. Only extract one class-level code involving multi-programming language interaction in the original code, including Class name, all relevant functions, parameters, return values and core logic.
2. According to the "Source code" and the key steps of the inter-process communication technology, if the inter-process communication technique requires multiple functions to work together, but the source code does not define any classes, please extract these functions and encapsulate these functions into a class-level code, preserve the original semantics of the code as much as possible; do not introduce any content other than encapsulation.
3. Extract all external library files and global variables that use the class, and ensure that these codes are completely derived from the original code.
4. Only extract task-related code, and the extracted code must strictly contain the key steps of the inter-process communication technology.
5. The output result should follow the format of "Output" in the example. Do not include any characters other than code, including "Output", "Python", " " ", or "Here's the extracted...".
6. Please strictly follow the instructions above.

Instruct:
You are an advanced programmer who focuses on multi-language interaction and inter-process communication technology.
Please complete the task according to the following example.

Example:
Input:
Original code: {Reference_raw_code}
Information related to the original code:
  - Programming language: Python
  - Inter-process communication technology name: HTTP Server-side by using FastAPI in python
  - The key steps of this inter-process communication technology:
    Step 1: Import FastAPI and necessary components from the FastAPI module
    Step 2: Instantiate the FastAPI application object, which will handle HTTP requests
    Step 3: Define route handlers using decorators (e.g., get, post) for each HTTP method
    Step 4: Optionally, define request and response models using Pydantic BaseModel for validation
    Step 5: Extract query parameters, path variables, and request body using FastAPI's parameter extraction mechanisms
    Step 6: Return the response from the route handler, typically as JSON or using FastAPI's built-in response classes

Output:
{Reference_class_level_code}

Request:
Now please follow the task description and the example to extract one expected multi-language interaction class-level code from the original code based on the input.

Input:
Original code: {Original_code}
Information related to the original code:
  - Programming language: {PL}
  - Inter-process communication technology name: {IPC_Name}
  - The key steps of this inter-process communication technology: {Steps}

Figure 8: Class-level code snippet extraction prompt template for "Class-Extractor"

**Instruction generation prompt template for "Instructor"**

You are a senior programmer specializing in multilingual interaction and inter -process communication (IPC) technologies.
Please carefully analyze the "Reference code", which is written in {PL} and utilizes the { IPC_Name} technique. The key steps of this technique include {Steps}.

Reference code:
{Canonical_solution}

If the reference code is at the function level, Please write a clear instruction detailing the function, including its purpose, input parameters (if any), and output (if any), to guide an LLM in generating reference code.
Use the following format to structure your instruction(Finally, please strictly follow the format requirements and do not insert any characters that are not relevant to the requirements.):

Task Description:...

Function Description:...

Input:...

Output:...

If the reference code is at the class level, please write a clear instruction detailing the class, including its purpose, attributes, and methods (with input parameters and return values). Ensure the instruction guides an LLM in generating reference code.

Use the following format to structure your instruction (Finally, please strictly follow the format requirements and do not insert any characters that are not relevant to the requirements.):
Task Description: ...

Class Description: ...

Attributes:

Attribute1: [Data Type] - [Description]
Attribute2: [Data Type] - [Description]
...

Methods:

Method1: [Name]([Input Parameters]) -> [Return Type] - [Description]
Method2: [Name]([Input Parameters]) -> [Return Type] - [Description]
...

Figure 9: Instruction generation prompt template for "Instructor"

---

**Algorithm 3:** CrossPL-FFI Construction

---

**Input :**
$\mathcal{P} = \{p_1, p_2, \ldots, p_M\}$: file paths of C source files;
$M$: number of C files collected from the source library (derived from GSL);
Prompt_file$_1$: CrossPL-FFI generation prompt template;
Prompt_file$_2$: error-revision prompt template.

**Output:**
FFI benchmark set $\mathcal{B}$ saved in JSON format.

1 **Preprocessing:**
2 Compile GSL source code into Linux shared object (.so) files using Autotools and Make .etc
   compile tools.
3                                               ▷ Prepare runtime environment before iteration
4 $T_{id} \leftarrow 1$                                           ▷ Initialize task identifier
5 **foreach** $p_i \in \mathcal{P}$ **do**
6     $C_i \leftarrow$ read_without_comments$(p_i)$                  ▷ Read C file and remove comments
7     $Instr \leftarrow \varnothing, Sol \leftarrow \varnothing, Err \leftarrow \varnothing$
8     **for** $k \leftarrow 0$ **to** $5$ **do**
9         **if** $k == 0$ **then**
10             $Prompt \leftarrow$ format(Prompt_file$_1$, $C_i$)
11         **else**
12             $Prompt \leftarrow$ format(Prompt_file$_2$, $C_i$, $Instr$, $Sol$, $Err$)
13         $Resp \leftarrow \mathcal{A}_1(Prompt)$                      ▷ LLM-based extraction
14         $Result \leftarrow$ Parse$(Resp)$            ▷ Parse the response generated by the LLM
15         **if** $Result == \varnothing$ **then**
16             **break**
17         $Instr \leftarrow Result[\texttt{"Instruction"}]$
18         $Sol \leftarrow Result[\texttt{"Canonical\_solution"}]$
19         **if** *execute(Sol, .so files) == success* **then**
20              ▷ ".so files serve as indispensable dependencies required for the execution of candidate code."
21             $\mathcal{B}_i \leftarrow \{T_{id}, \text{"FFI"}, Instr, Sol\}$
22             Save$(\mathcal{B}_i, \text{save\_path}, T_{id})$
23             $T_{id} \leftarrow T_{id} + 1$
24             **break**
25         **else**
26             $Err \leftarrow$ execution_error_message

**Prompt template for generating CrossPL-FFI**

You are given one low-level C function from an open-source project (GNU Scientific Library). This function has been compiled into a dynamic library (.so on Linux) that can be loaded in Python using the ctypes module. The dynamic library paths are as follows:
1. /usr/local/gsl/lib/libgslcblas.so
2. /usr/local/gsl/lib/libgsl.so

Extra Knowledge:
libgsl.so: The main GNU Scientific Library, provides high-level numerical routines.
libgslcblas.so: GSL's built-in C BLAS library, implements low-level linear algebra functions (e.g., matrix/vector operations).
Relationship: libgsl.so depends on libgslcblas.so; you must load libgslcblas.so first (or globally) to resolve BLAS symbols, otherwise you get "undefined symbol" errors.
The recommended loading order is as follows:
ctypes.CDLL("/usr/local/gsl/lib/libgslcblas.so", mode=ctypes.RTLD_GLOBAL)
gsl = ctypes.CDLL("/usr/local/gsl/lib/libgsl.so")

The source code of a function from the dynamic library is shown below:
{source_code}

You can refer to the following link for definitions of functions related to the GNU Scientific Library (GSL):
https://www.gnu.org/software/gsl/doc/html/

Your task is to design both an Instruction and a Canonical_solution that evaluate an LLM's ability to write correct Python code
that calls this C function via ctypes.

Instruction should include:
    1. A concise and clear natural language task description specifying the intended functionality(e.g., compute a mathematical result, manipulate data structures, process arrays).
    2. An explicit requirement that the solution must use ctypes to load and call the corresponding C function from the compiled dynamic library.
    3. Well-defined input and output specifications to ensure correctness can be verified automatically.

Canonical_solution should include:
    1. A complete and correct Python solution that fulfills the instruction.
    2. Several assertion-based test cases to validate correctness, robustness, and error handling.
    3. All assertion tests must be encapsulated within a function named `assert_test`.
    4. The `assert_test` function must be executed within the `if __name__ == "__main__"` block.

Respond ONLY with a valid JSON object.
Do not use triple quotes (""") in your response.
Wrap code in a single string with escaped newlines (\n).

The returned content must be a dictionary only with the following fields:
"Instruction":,
"Canonical_solution":

Figure 10: Prompt template for constructing CrossPL-FFI

| Prompt template with error information for generating CrossPL-FFI |
|---|
| previously, an attempt was made to generate a Python Canonical_solution that calls this C function via ctypes. The dynamic library paths are as follows: 1. /usr/local/gsl/lib/libgslcblas.so 2. /usr/local/gsl/lib/libgsl.so

The previously generated Instruction and Canonical_solution are as follows: Instruction: {Instruction}

Canonical_solution: {Canonical_solution}

However, when executing the Canonical_solution, the following error occurred: {error_message}

Your task is to revise and generate a corrected Python Canonical_solution that fixes the above error while still satisfying the original Instruction. Ensure that the new solution correctly handles memory allocation, type conversion, buffer sizes, and any error codes or return values.

Canonical_solution requirements: 1. A complete and correct Python solution that fulfills the Instruction. 2. Several assertion-based test cases to validate correctness, robustness, and error handling. 3. All assertion tests must be encapsulated within a function named `assert_test`. 4. The `assert_test` function must be executed within the `if __name__ == "__main__"` block.

Respond ONLY with a valid JSON object. Do not use triple quotes (""") in your response. Wrap code in a single string with escaped newlines ( \n).

The returned content must be a dictionary only with the following field: "Canonical_solution": |

Figure 11: Prompt template with error information for constructing CrossPL-FFI

| Add class information to the Instruction |
|---|
| To accomplish the above task, the required classes are: {class_names}, and the required functions along with their parameter lists are: {functin_names}.

The recommended dynamic library loading order is as follows: ctypes.CDLL("/usr/local/gsl/lib/libgslcblas.so", mode=ctypes.RTLD_GLOBAL) gsl = ctypes.CDLL("/usr/local/gsl/lib/libgsl.so")

You must return ONLY a valid JSON object. Do not include explanations, comments, or extra text. Do not use triple quotes (""").

The JSON object must contain exactly one field: "Candidate_solution": <string>

Inside "Candidate_solution", provide the complete code as a single string. Escape all line breaks as \n. |

Figure 12: Add class information to the Instruction

---

**Add function information to the Instruction**

To accomplish the above task, the required functions along with their parameter lists are: { functin_names}.

The recommended dynamic library loading order is as follows:
ctypes.CDLL("/usr/local/gsl/lib/libgslcblas.so", mode=ctypes.RTLD_GLOBAL)
gsl = ctypes.CDLL("/usr/local/gsl/lib/libgsl.so")

You must return ONLY a valid JSON object.
Do not include explanations, comments, or extra text.
Do not use triple quotes (""").

The JSON object must contain exactly one field:
"Candidate_solution": <string>

Inside "Candidate_solution", provide the complete code as a single string.
Escape all line breaks as \n.

Figure 13: Add function information to the Instruction

*HTTP* accounts for the highest proportion of tasks with 779 (39.30%), while *UDP* for the lowest with 92 tasks (4.64%).

Fig. 14b further details the distribution of IPC technologies used by different programming languages within the CrossPL-IPC. Due to the distinct characteristics of each language, the proportional use of IPC technologies varies considerably; for example, Java tasks are predominantly associated with *TCP*, while JavaScript tasks are mostly related to *HTTP*. Notably, some IPC techniques cannot be fully implemented within a single function and require implementation at the class level. Consequently, during IPC-related code extraction, we distinguished between class-level and function-level code. Overall, class-level code accounts for 59.99% of the CrossPL-IPC, while function-level code constitutes 40.01%. This distribution reflects language-specific design patterns: for example, class-level implementations dominate in Java and PHP due to their object-oriented paradigms, while languages like Go and JavaScript tend to favor function-level or lightweight constructs. Such variations emphasize the necessity of handling both granularities in CrossPL-IPC to faithfully capture real-world IPC usage across different programming ecosystems.

Fig. 15 presents the distribution of code line counts in the canonical solutions across CrossPL. Fig. 15a presents the distribution of Canonical solution Code Lines of CrossPL-IPC. Fig. 15b presents the distribution of Canonical solution Code Lines of CrossPL-FFI. Specifically, the code line of the CrossPL-IPC exhibits a median of 51 lines, an average of 66.96 lines, and a maximum of 483 lines. In contrast, the CrossPL-FFI shows a more compact distribution, with a median of 44 lines, an average of 47.1 lines, a maximum of 112 lines, and a minimum of 8 lines.

Fig. 16 illustrates the distribution of instruction lengths, measured in characters, across the canonical solutions of CrossPL. Fig. 16a shows the distribution of instruction length (characters) of CrossPL-IPC. Fig. 16b presents the distribution of instruction length (characters) of CrossPL-FFI. For the CrossPL-IPC, the instructions have a median length of 1204 characters, an average of 1284.64 characters, and a minimum of 473 characters. In comparison, the length of instruction in CrossPL-FFI shows a slightly larger scale, with a median of 1277.5 characters, an average of 1329.74 characters, a maximum of 2421 characters, and a minimum of 973 characters.

## B.2 DETAILED STATICS OF INSTANCES FILTERED AT EACH KEY STEP

To enhance transparency and ensure methodological reproducibility, we provide detailed statistics on the number of samples retained after each key filtering step in the pipelines used to construct CrossPL-IPC and CrossPL-FFI. Table 1 shows the detailed statistics on the number of samples retained after each key filtering step in CrossPL-IPC construction pipeline. Table 2 shows the detailed statistics on the number of samples retained after each key filtering step in CrossPL-FFI construction pipeline.

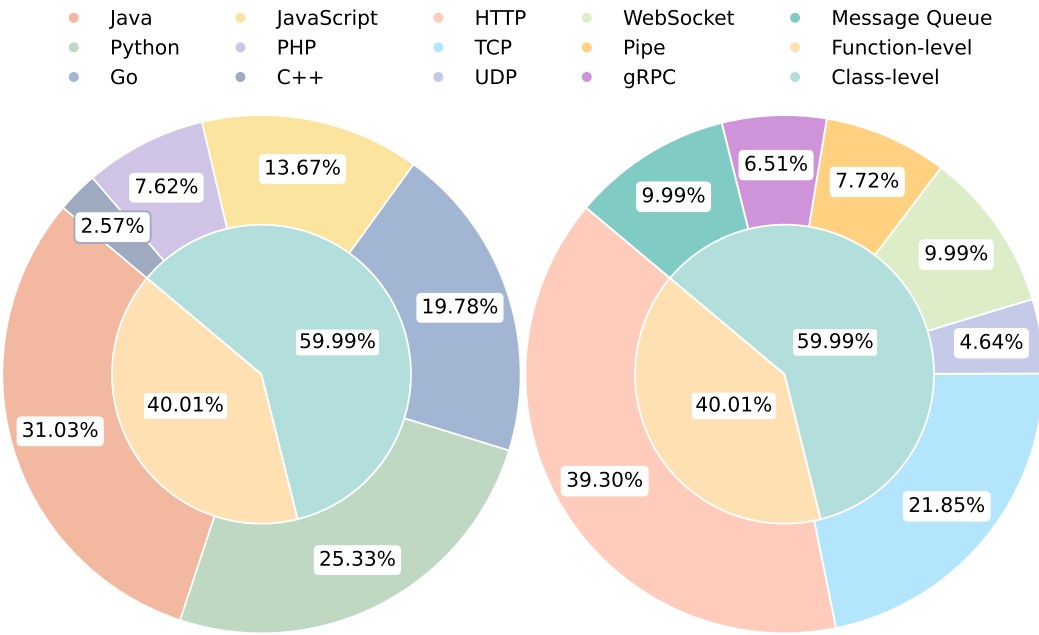

(a) Pie chart of CrossPL-IPC dataset from different view.

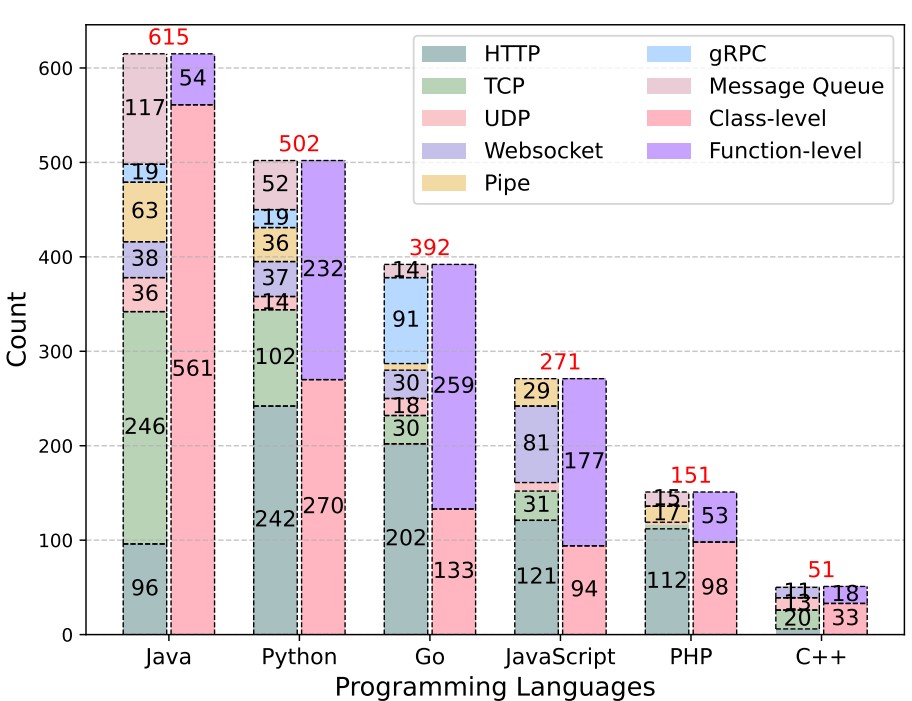

(b) Distribution of different IPC technologies across different programming languages.

Figure 14: Distribution of CrossPL-IPC dataset from different view.

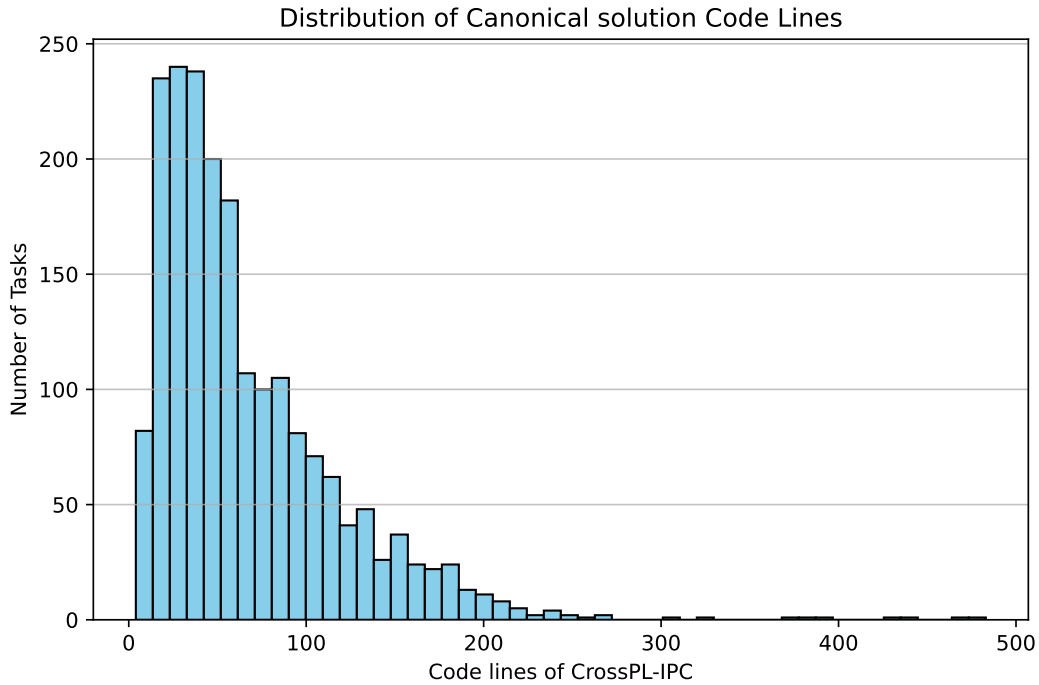

(a) Distribution of Canonical solution Code Lines of CrossPL-IPC.

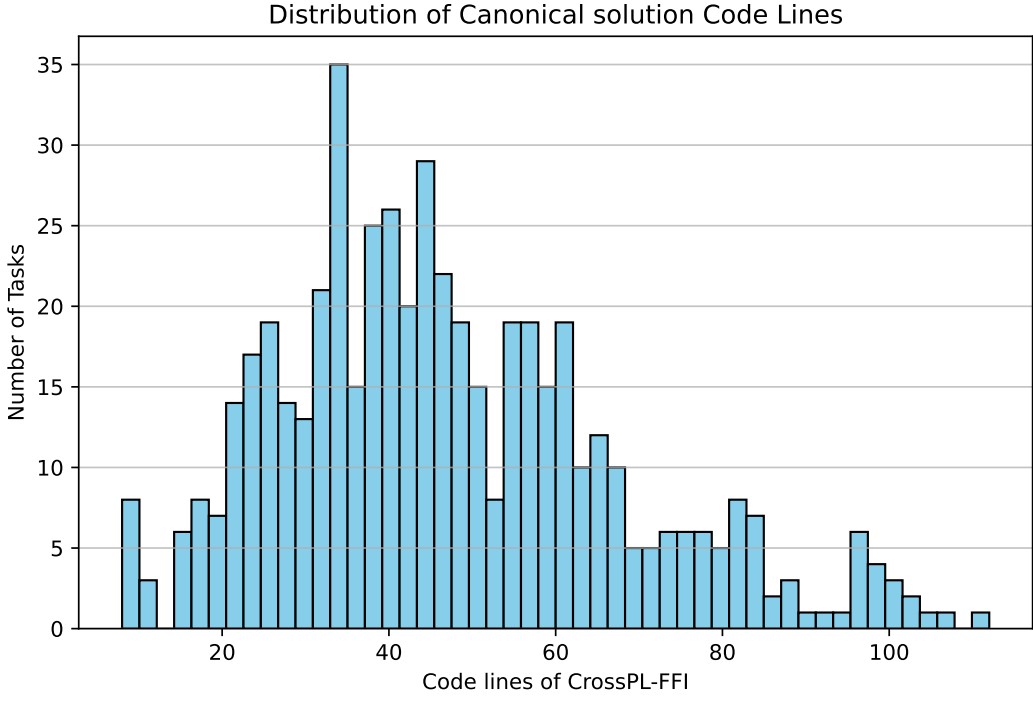

(b) Distribution of Canonical solution Code Lines of CrossPL-FFI.

Figure 15: Distribution of Canonical solution Code Lines of CrossPL-IPC and CrossPL-FFI.

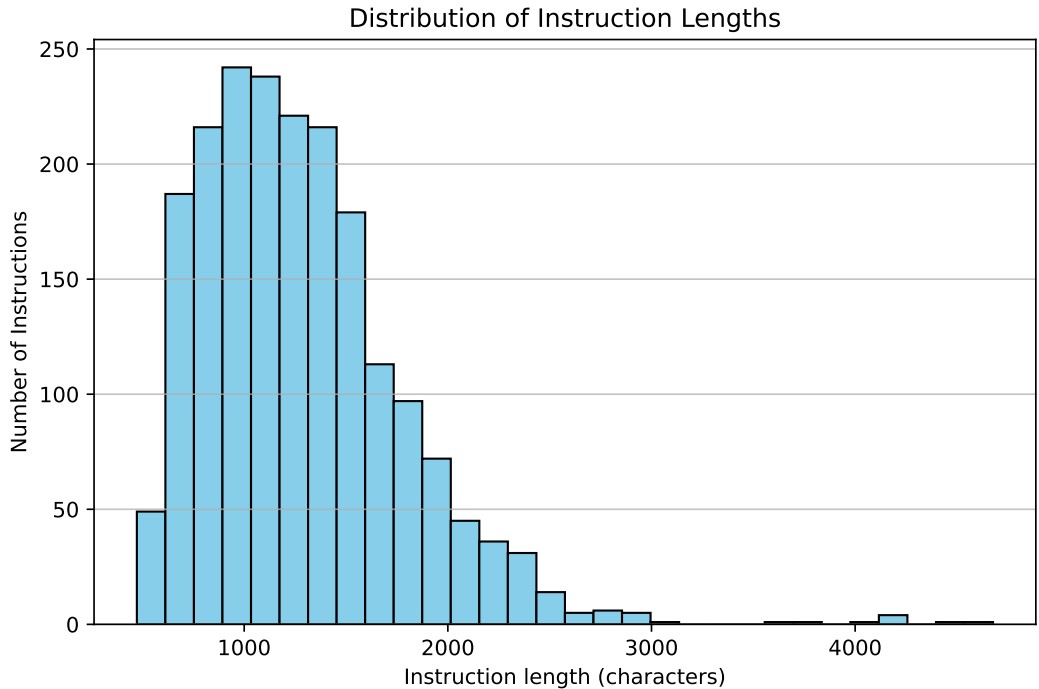

(a) Distribution of instruction length (characters) of CrossPL-IPC.

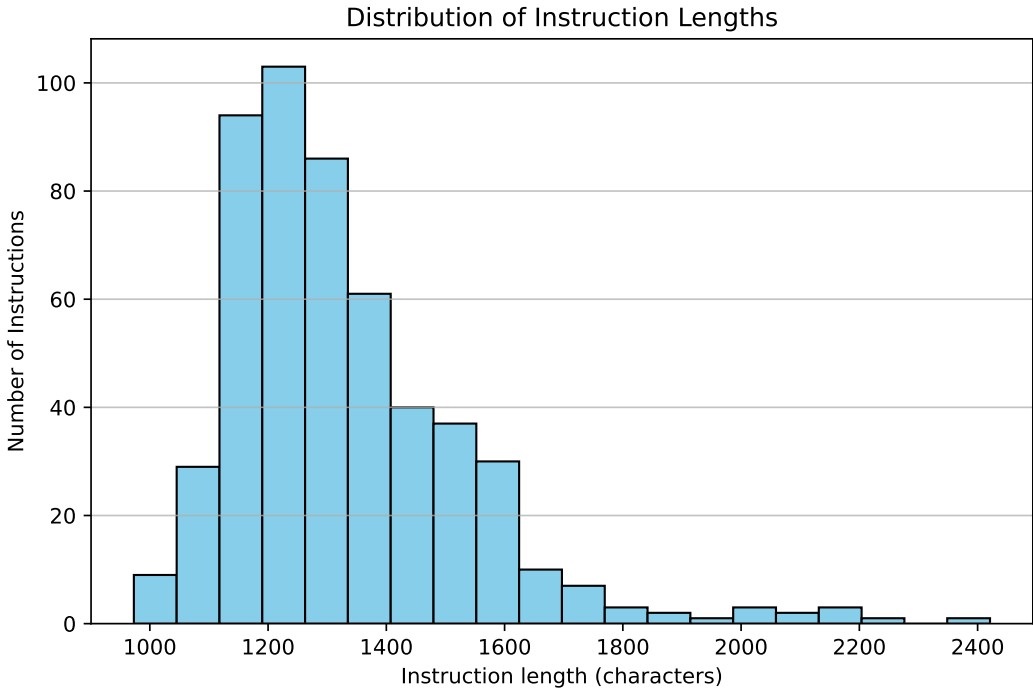

(b) Distribution of instruction length (characters) of CrossPL-FFI.

Figure 16: Distribution of instruction length (characters) of CrossPL-IPC and CrossPL-FFI.

Table 1: Detailed number statistics of instances filtered at each key step in CossPL-IPC construction pipeline

| Language | Total MPL Projects | File Counts (FSM Filtered) | File Counts (Token Limit) | Instance Counts (Judger Filtered) | Instance Counts (FSM Filtered) |
|---|---|---|---|---|---|
| C++ | 1462 | 70 | 70 | 58 | 51 |
| Go | 1534 | 2239 | 998 | 497 | 392 |
| Java | 8807 | 1038 | 982 | 669 | 615 |
| JavaScript | 7560 | 3721 | 807 | 368 | 271 |
| PHP | 704 | 921 | 292 | 252 | 151 |
| Python | 5219 | 1690 | 913 | 522 | 502 |

Table 2: Detailed number statistics of instances filtered at each key step in CossPL-FFI construction pipeline

| Language | Total C Functions | File Counts (Execution Filtered) |
|---|---|---|
| Python-C | 1061 | 522 |

## C  EXPERIMENT RESULTS

**Calculation method for 95% confidence interval:** For all non-greedy decoding settings reported in this paper, we provide 95% confidence intervals for both Pass@1 and Pass@5, computed using a bootstrap procedure. The bootstrap resampling was performed with 5,000 iterations. For each resample, we recomputed the macro-average pass@5 and pass@1; The 2.5th and 97.5th percentiles of the resulting distribution were taken as the 95% confidence interval. This procedure provides a distribution-free and statistically robust estimate of the uncertainty for pass@5.

### C.1  RESULTS FOR ANSWER RQ1

#### C.1.1  HOW EFFECTIVELY CAN LLMS GENERATE IPC CODE ACROSS DIFFERENT PROGRAMMING LANGUAGES IN CPL PROJECTS?

**Overall performance.** Table 3 and Table 4 present[7] the Pass@1 and Pass@5 scores, respectively, for different LLMs across MPL on IPC tasks. Overall, Pass@5 success rates consistently exceed their Pass@1 counterparts across all the task subcategories, reflecting improved performance through multiple sampling attempts. Among the evaluated models, GLM4-plus achieves the highest average Pass@1 score at 75.88%, followed closely by GPT-4o at 73.81%. In contrast, Qwen3-8b exhibits notably poor performance on Java IPC tasks, with a Pass@1 of only 10.73% and a Pass@5 of 13.56%.

Further analysis of Table 3 reveals that, among models with known parameter sizes, larger models generally exhibit better performance. However, this trend is not strictly linear. For instance, the 671b model (DeepSeek-V3) and the 235b model (Qwen3-235b-a22b) perform comparably, with an average difference of only 2.67%. In contrast, both significantly outperform the 30b model (Qwen3-30b-a3b), with an average improvement of 9.87%. Comparing Table 3 with Table 4, Pass@5 typically yields a substantial improvement over Pass@1. However, some models with fewer than 10 billion parameters perform poorly on Java IPC tasks, achieving Pass@5 rates of only around 50% or even lower.

**Effectiveness of code models.** Comparing general-purpose models with code-specific models, code models deliver comparable or even superior performance despite fewer parameters, emphasizing the effectiveness of LLMs' targeted domain adaptation. Analysis of Table 3 indicates that, given comparable parameter sizes, code-specialized models tend to achieve higher performance. Taking the Go subset as an example, Qwen2.5-Coder-32B-Instruct outperforms Qwen3-32B by a substantial margin, with a 24.24% higher Pass@1 score.

---

[7]Values highlighted in bold red indicate the best results, while those underlined in blue represent the second-best results. Similarly for the later tables.

**Impact of Code Structural Granularity on Model Performance.** An analysis of Tables 3 and 4 reveals little correlation between model performance and code structural complexity. Although 91.22% of candidate samples in the Java subset are at the class level (perceived more complex than the function level), the model does not exhibit the poorest performance on this subset. In contrast, the Go subset contains a higher proportion of function-level candidates (66.07%), yet the model performs relatively poorly on it.

**Language-specific performance disparities.** From the analysis, it is evident that the models perform best on C++ CPL tasks, while their performance on Go-related CPL tasks is relatively weaker. A potential explanation for this disparity lies in the structural characteristics of the Go language. Specifically, Go does not natively support the concept of classes, whereas much of the code knowledge learned by current LLMs is derived from class-based programming paradigms. In constructing class-level code samples for Go, we extracted code that emulates class-like behavior using structs. This structural mismatch may impose an additional cognitive burden on LLMs trained predominantly on class-based languages, thereby reducing their effectiveness when generating code in Go for such tasks. Conversely, the better performance on C++ tasks can be attributed to the CrossPL focus on widely used CPL techniques such as *TCP*, *UDP*, *HTTP*, and *WebSocket*. These protocols are well-documented and frequently appear in public code repositories, increasing the likelihood of LLM pretraining exposure. In addition, C++ is closely associated with low-level system programming, which aligns well with the structural characteristics of typical CPL implementations. Fig. 17a illustrates the distribution of Pass@5 scores across different programming languages. The figure clearly highlights performance discrepancies among tasks in various languages. Overall, the models achieve the best performance on C++, while their performance on Go is relatively weaker.

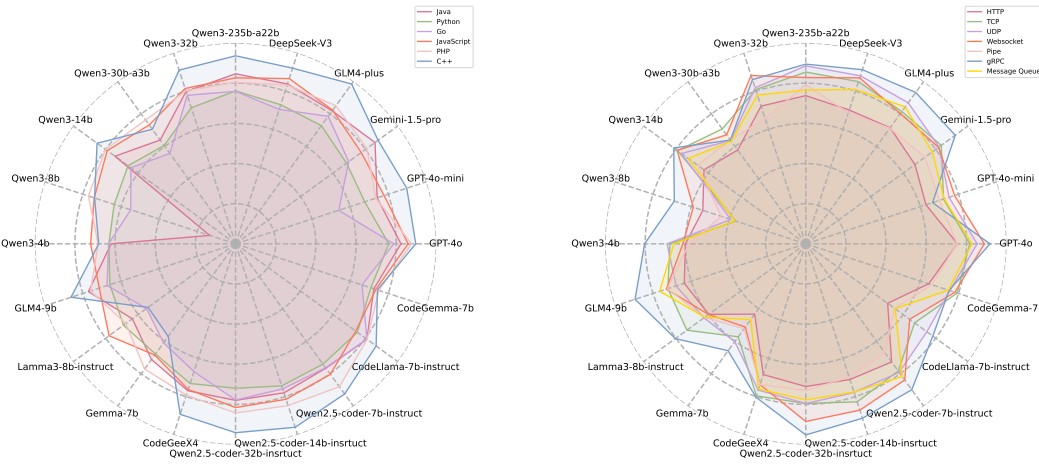

(a) Radar Plot of Pass@5 on CrossPL-IPC (programming languages view)

(b) Radar Plot of Pass@5 on CrossPL-IPC (IPC technologies view)

Figure 17: Radar Plot of Pass@5 on CrossPL-IPC

### C.1.2 HOW DOES THEIR PERFORMANCE VARY ACROSS DIFFERENT IPC TECHNIQUES?

**Overall performance.** Observe from Tables 5 and 6 that GLM4-plus achieves the best overall performance across the subsets of various IPC techniques. It attains the highest *Pass@1* scores in the *UDP*, *WebSocket*, and *Message Queue* subsets, and ranks second-best in the remaining four subsets. In contrast, Llama3-8b-instruct exhibits the lowest average *Pass@1* score at only 40.67%, while Qwen3-8b records the lowest average *Pass@5* score.

**Notable outliers and analysis.** Interestingly, Gemma-7b exhibits lower *Pass@5* than *Pass@1* on the *HTTP* and *Message Queue* subsets, indicating that sampling introduces noise rather than diversity. Across other IPC subsets, limited *Pass@5* improvements further reveal Gemma-7b's difficulty in generating diverse yet correct alternatives, highlighting its reliance on deterministic decoding.

Table 3: Performance of various models across programming languages (pass@1, greedy decode)

| Model Type | Model | Published Time | Size | Java ↑ | Python ↑ | Go ↑ | JavaScript ↑ | PHP ↑ | C++ ↑ | Mean ↑ |
|---|---|---|---|---|---|---|---|---|---|---|
| General Model | GPT-4o Hurst et al. (2024) | 2024 | \ | 75.12% | 67.13% | 71.94% | 77.86% | 82.12% | 92.16% | 73.81% |
| | GPT-4o-mini Hurst et al. (2024) | 2024 | \ | 69.43% | 62.15% | 49.23% | 73.80% | 71.52% | 80.39% | 64.63% |
| | Gemini-1.5-pro | 2024 | \ | 72.52% | 63.35% | 66.33% | 73.43% | 78.15% | 90.20% | 69.98% |
| | GLM4-plus AI (2024) | 2024 | \ | 77.72% | 69.72% | 73.47% | 79.34% | 82.12% | 96.08% | 75.88% |
| | DeepSeek-V3 Liu et al. (2024) | 2024 | 671B | 78.21% | 64.74% | 66.84% | 82.66% | 79.47% | 88.24% | 73.51% |
| | Qwen3-235b-a22b Yang et al. (2025) | 2025 | 235B | 77.40% | 67.93% | 65.31% | 76.38% | 74.83% | 88.24% | 72.55% |
| | Qwen3-32b Yang et al. (2025) | 2025 | 32B | 70.89% | 62.75% | 49.74% | 74.17% | 75.50% | 80.39% | 65.69% |
| | Qwen3-30b-a3b Yang et al. (2025) | 2025 | 30B | 57.24% | 51.99% | 49.75% | 70.48% | 71.52% | 66.67% | 57.57% |
| | Qwen3-14b Yang et al. (2025) | 2025 | 14B | 67.64% | 57.57% | 59.44% | 77.12% | 76.82% | 70.59% | 65.54% |
| | Qwen3-8b Yang et al. (2025) | 2025 | 8B | 10.73% | 55.78% | 48.47% | 68.63% | 68.21% | 64.71% | 43.29% |
| | Qwen3-4b Yang et al. (2025) | 2025 | 4B | 54.96% | 57.97% | 52.81% | 66.79% | 57.62% | 64.71% | 57.37% |
| | GLM4-9b GLM et al. (2024) | 2024 | 9B | 65.04% | 54.58% | 56.12% | 66.42% | 70.86% | 58.82% | 61.10% |
| | Lamma3-8b-instruct Grattafiori et al. (2024) | 2024 | 8B | 35.56% | 45.42% | 29.59% | 53.87% | 45.70% | 31.37% | 40.67% |
| | Gemma-7b Team et al. (2024b) | 2024 | 7B | 52.52% | 48.01% | 39.80% | 58.67% | 64.90% | 43.14% | 50.40% |
| Code Model | CodeGeeX4 Zheng et al. (2023) | 2024 | \ | 69.27% | 62.55% | 55.61% | 69.00% | 70.20% | 80.39% | 65.19% |
| | Qwen2.5-coder-32b-instruct Hui et al. (2024) | 2024 | 32B | 72.52% | 65.74% | 73.98% | 76.75% | 77.48% | 90.20% | 72.50% |
| | Qwen2.5-coder-14b-instruct Hui et al. (2024) | 2024 | 14B | 70.41% | 66.73% | 69.90% | 74.17% | 78.81% | 90.20% | 71.04% |
| | Qwen2.5-coder-7b-instruct Hui et al. (2024) | 2024 | 7B | 68.78% | 63.55% | 66.07% | 75.65% | 77.48% | 84.31% | 68.92% |
| | CodeLlama-7b-instruct Roziere et al. (2023) | 2023 | 7B | 53.33% | 44.41% | 49.74% | 49.82% | 54.30% | 41.18% | 50.15% |
| | CodeGemma-7b Team et al. (2024a) | 2024 | 7B | 52.20% | 53.78% | 52.30% | 65.31% | 62.91% | 58.92% | 55.40% |

Table 4: Performance of various models across programming languages (pass@5, temperature=0.2, top-p=0.95)

| Model Type | Model | Published Time | Size | Java ↑ | Python ↑ | Go ↑ | JavaScript ↑ | PHP ↑ | C++ ↑ | Mean ↑ | 95%CI |
|---|---|---|---|---|---|---|---|---|---|---|---|
| General Model | GPT-4o | 2024 | / | 82.38% | 76.30% | 78.79% | 85.97% | 87.69% | 89.76% | 81.21% | [79.52%, 82.84%] |
| | GPT-4o-mini | 2024 | / | 74.16% | 67.66% | 54.01% | 77.81% | 73.45% | 89.66% | 69.37% | [67.38%, 71.41%] |
| | Gemini-1.5-pro | 2024 | / | 85.86% | 68.32% | 69.69% | 77.39% | 80.83% | 87.75% | 76.72% | [74.87%, 78.52%] |
| | GLM4-plus | 2024 | / | 81.88% | 72.74% | 78.51% | 82.27% | 85.04% | 98.37% | 79.62% | [77.82%, 81.34%] |
| | DeepSeek-V3 | 2024 | 671B | 83.70% | 72.82% | 70.38% | 86.54% | 83.13% | 91.99% | 78.86% | [77.10%, 80.59%] |
| | Qwen3-235b-a22b | 2025 | 235B | 84.69% | 76.16% | 75.96% | 82.67% | 80.43% | 93.64% | 80.42% | [78.77%, 82.07%] |
| | Qwen3-32b | 2025 | 32B | 79.73% | 71.24% | 77.77% | 81.33% | 80.13% | 90.96% | 77.73% | [76.01%, 79.50%] |
| | Qwen3-30b-a3b | 2025 | 30B | 63.89% | 60.52% | 55.66% | 73.18% | 76.21% | 70.52% | 63.79% | [61.70%, 65.93%] |
| | Qwen3-14b | 2025 | 14B | 74.22% | 66.18% | 64.44% | 78.98% | 80.04% | 85.24% | 71.63% | [69.69%, 73.63%] |
| | Qwen3-8b | 2025 | 8B | 13.56% | 63.55% | 54.88% | 74.03% | 77.07% | 73.82% | 49.05% | [46.93%, 51.23%] |
| | Qwen3-4b | 2025 | 4B | 61.98% | 63.09% | 63.22% | 72.19% | 68.45% | 68.15% | 64.55% | [62.53%, 66.63%] |
| | GLM4-9b | 2024 | 9B | 77.06% | 65.81% | 67.30% | 71.36% | 75.97% | 86.17% | 70.93% | [68.96%, 72.86%] |
| | Lamma3-8b-instruct | 2024 | 70B | 63.54% | 68.98% | 54.81% | 77.88% | 67.54% | 53.87% | 65.21% | [63.34%, 67.11%] |
| | Gemma-7b | 2024 | 7B | 71.23% | 67.84% | 60.98% | 68.47% | 77.07% | 68.05% | 68.05% | [66.12%, 69.92%] |
| Code Model | CodeGeeX4 | 2024 | / | 76.93% | 73.15% | 66.95% | 76.26% | 79.27% | 89.28% | 74.44% | [72.62%, 76.26%] |
| | Qwen2.5-coder-32b-instruct | 2024 | 32B | 77.96% | 71.95% | 77.78% | 81.56% | 84.27% | 94.07% | 78.16% | [76.35%, 79.96%] |
| | Qwen2.5-coder-14b-instruct | 2024 | 14B | 78.04% | 74.40% | 75.97% | 81.41% | 84.73% | 96.08% | 78.14% | [76.38%, 79.84%] |
| | Qwen2.5-coder-7b-instruct | 2024 | 7B | 76.56% | 74.06% | 76.33% | 80.34% | 87.98% | 92.44% | 77.68% | [75.89%, 79.37%] |
| | CodeLlama-7b-instruct | 2023 | 7B | 80.88% | 74.99% | 80.89% | 74.02% | 80.98% | 86.56% | 78.60% | [77.03%, 80.22%] |
| | CodeGemma-7b | 2024 | 7B | 72.44% | 70.86% | 66.22% | 73.50% | 75.05% | 74.31% | 72.01% | [70.10%, 73.81%] |

**IPC-specific performance disparities**: Tables 5, 6, reveal significant variation in LLM performance across IPC subsets for CPL interoperating code. The *gRPC* subset achieves the highest results, with 10 LLMs surpassing 80% on Pass@1 and 14 on Pass@5; the top model reaches 90.70% and 93.17%, respectively. This success likely stems from *gRPC*'s standardized syntax, schema-driven design, and extensive representation in training data. In contrast, the low-level, platform-dependent *Pipe* subset underperforms, with 9 LLMs below 60% on Pass@1 and 10 below 70% on Pass@5; minimum scores are 33.91% and 44.23%. Similarly, the *HTTP* subset shows weaker results—10 LLMs under 60% Pass@1 and 14 under 70% Pass@5, likely due to its variable implementations and lack of strict interfaces, hindering LLM generalization. Fig. 17b illustrates the distribution of Pass@5 scores across different IPC technologies. The figure clearly highlights performance discrepancies among tasks in various IPC technologies.

Table 5: Performance of various models across IPC techniques (pass@1, greedy decode)

| Model Type | Model | Published Time | Size | HTTP ↑ | TCP ↑ | UDP ↑ | Websocket ↑ | Pipe ↑ | gRPC ↑ | Message Queue ↑ | Mean ↑ |
|---|---|---|---|---|---|---|---|---|---|---|---|
| | GPT-4o | 2024 | / | **71.63%** | 71.13% | 82.61% | 82.32% | 64.05% | 86.82% | 74.75% | 73.81% |
| | GPT-4o-mini | 2024 | / | 59.69% | 68.13% | 69.57% | 72.73% | 66.67% | 59.69% | 67.68% | 64.63% |
| | Gemini-1.5-pro | 2024 | / | 61.49% | 74.60% | 73.91% | 79.29% | **68.63%** | 86.82% | 72.22% | 69.98% |
| | GLM4-plus | 2024 | / | 68.68% | 77.83% | 84.78% | 88.38% | 67.32% | 89.92% | 80.81% | 75.88% |
| | DeepSeek-V3 | 2024 | 671B | 66.24% | 78.79% | 84.78% | 82.83% | 60.78% | 87.60% | 76.26% | 73.51% |
| | Qwen3-235b-a22b | 2025 | 235B | 67.01% | 77.60% | 83.70% | 74.75% | 65.36% | 86.82% | 72.22% | 72.55% |
| | Qwen3-32b | 2025 | 32B | 66.20% | 69.48% | 77.03% | 84.52% | 54.11% | 81.58% | 73.91% | 65.69% |
| General Model | Qwen3-30b-a3b | 2025 | 30B | 52.76% | 64.43% | 60.87% | 62.63% | 56.86% | 56.59% | 56.06% | 57.57% |
| | Qwen3-14b | 2025 | 14B | 59.44% | 72.98% | 70.65% | 72.73% | 51.63% | 76.74% | 67.17% | 65.54% |
| | Qwen3-8b | 2025 | 8B | 49.55% | 29.79% | 35.87% | 52.53% | 35.95% | 62.79% | 35.35% | 43.29% |
| | Qwen3-4b | 2025 | 4B | 54.43% | 62.12% | 63.04% | 54.04% | 48.37% | 68.22% | 59.09% | 57.37% |
| | GLM4-9b | 2024 | 9B | 56.74% | 60.28% | 60.87% | 60.61% | 60.13% | 81.40% | 68.18% | 61.10% |
| | Lamma3-8b-instruct | 2024 | 8B | 40.31% | 44.80% | 36.96% | 44.44% | 33.91% | 38.76% | 34.34% | 40.67% |
| | Gemma-7b | 2024 | 7B | 47.50% | 56.12% | 53.26% | 49.49% | 47.06% | 52.71% | 50.00% | 50.40% |
| | CodeGeeX4 | 2024 | / | 60.08% | 69.05% | 66.30% | 68.69% | 62.75% | 72.87% | 69.70% | 65.19% |
| | Qwen2.5-coder-32b-instruct | 2024 | 32B | 67.27% | 72.52% | 78.26% | 84.34% | 63.40% | **90.70%** | 73.74% | 72.50% |
| Code Model | Qwen2.5-coder-14b-instruct | 2024 | 14B | 65.47% | 76.21% | 67.39% | 82.32% | 62.09% | 86.82% | 69.70% | 71.04% |
| | Qwen2.5-coder-7b-instruct | 2024 | 7B | 64.31% | 71.36% | 70.65% | 75.25% | 62.09% | 81.40% | 71.72% | 68.92% |
| | CodeLlama-7b-instruct | 2023 | 7B | 46.98% | 57.74% | 55.43% | 44.95% | 45.10% | 60.47% | 45.96% | 50.15% |
| | CodeGemma-7b | 2024 | 7B | 50.83% | 63.05% | 61.96% | 62.63% | 45.75% | 54.26% | 54.55% | 55.40% |

Table 6: Performance of various models across IPC techniques (pass@5, temperature=0.2, top-p=0.95)

| Model Type | Model | Published Time | Size | HTTP ↑ | TCP ↑ | UDP ↑ | Websocket ↑ | Pipe ↑ | gRPC ↑ | Message Queue ↑ | Mean ↑ | 95%CI |
|---|---|---|---|---|---|---|---|---|---|---|---|---|
| | GPT-4o | 2024 | / | **75.58%** | 81.66% | 85.35% | **88.96%** | 75.40% | 91.75% | 82.74% | **81.21%** | [79.52%,82.84%] |
| | GPT-4o-mini | 2024 | / | 62.95% | 72.16% | 75.03% | 77.93% | 69.92% | 66.39% | 72.53% | 69.37% | [67.38%,71.41%] |
| | Gemini-1.5-pro | 2024 | / | 67.20% | 83.42% | 82.04% | 81.77% | 73.36% | 92.02% | 77.89% | 76.72% | [74.87%,78.52%] |
| | GLM4-plus | 2024 | / | 71.47% | 81.42% | 86.84% | 80.18% | 71.20% | 84.25% | 79.62% | 79.62% | [77.82%,81.34%] |
| | DeepSeek-V3 | 2024 | 671B | 70.06% | 84.92% | 87.99% | 86.91% | 68.58% | 91.20% | 80.84% | 78.86% | [77.10%,80.59%] |
| | Qwen3-235b-a22b | 2025 | 235B | 73.80% | 85.55% | 88.62% | 82.81% | 79.30% | 89.49% | 76.61% | 80.42% | [78.77%,82.07%] |
| | Qwen3-32b | 2025 | 32B | 72.12% | 80.70% | 81.68% | 88.14% | 65.45% | 86.00% | 77.88% | 77.73% | [76.01%,79.50%] |
| General Model | Qwen3-30b-a3b | 2025 | 30B | 57.73% | 70.75% | 63.62% | 67.08% | 64.04% | 63.84% | 63.13% | 63.79% | [61.70%,65.93%] |
| | Qwen3-14b | 2025 | 14B | 62.68% | 78.91% | 76.56% | 79.18% | 66.53% | 81.29% | 72.37% | 71.63% | [69.69%,73.63%] |
| | Qwen3-8b | 2025 | 8B | 53.91% | 36.66% | 39.25% | 58.91% | 44.23% | 68.85% | 37.08% | 49.05% | [46.93%,51.23%] |
| | Qwen3-4b | 2025 | 4B | 59.93% | 67.66% | 69.06% | 61.24% | 58.24% | 80.28% | 65.70% | 64.55% | [62.53%,66.63%] |
| | GLM4-9b | 2024 | 9B | 63.63% | 72.22% | 67.93% | 73.03% | 72.46% | 89.40% | 76.52% | 70.93% | [68.96%,72.86%] |
| | Lamma3-8b-instruct | 2024 | 8B | 59.75% | 73.19% | 62.68% | 59.85% | 62.28% | 80.30% | 62.14% | 65.21% | [63.34%,67.11%] |
| | Gemma-7b | 2024 | 7B | 43.31% | 57.26% | 60.24% | 51.23% | 52.28% | 65.82% | 46.68% | 68.05% | [66.12%,69.92%] |
| | CodeGeeX4 | 2024 | / | 68.49% | 79.62% | 76.58% | 73.73% | 73.73% | 80.32% | 76.04% | 74.44% | [72.62%,76.26%] |
| | Qwen2.5-coder-32b-instruct | 2024 | 32B | 71.07% | 79.52% | 79.56% | 88.49% | 73.13% | 95.18% | 77.72% | 78.16% | [76.35%,79.96%] |
| Code Model | Qwen2.5-coder-14b-instruct | 2024 | 14B | 71.07% | 82.77% | 77.87% | 87.11% | 69.81% | 91.35% | 77.69% | 78.14% | [76.38%,79.84%] |
| | Qwen2.5-coder-7b-instruct | 2024 | 7B | 72.75% | 78.32% | 79.16% | 83.85% | 67.12% | 89.81% | 81.73% | 77.68% | [75.89%,79.37%] |
| | CodeLlama-7b-instruct | 2023 | 7B | 50.41% | 66.99% | 74.54% | 63.97% | 54.68% | 78.20% | 55.02% | 78.60% | [77.03%,80.22%] |
| | CodeGemma-7b | 2024 | 7B | 64.38% | 79.31% | 75.47% | 78.09% | 65.98% | 75.03% | 74.63% | 72.01% | [70.10%,73.81%] |

### C.1.3 WHAT ARE THE ERROR DISTRIBUTIONS IN CASES OF FAILURE?

To further investigate the error distribution when LLMs generate IPC interaction code, we selected the best-performing model, GPT-4o, and the worst-performing model, Llama3-8b-instruct, for analysis. Tables 18 and 19 present the error distributions of GPT-4o and Llama3-8b-instruct in generating IPC code across various programming languages. From these figures, it is apparent that LLMs exhibit varying degrees of success in correctly covering the states within the corresponding FSM state transition chains when generating erroneous IPC code. Analyzing Table 18, we observe that GPT-4o's state coverage is relatively balanced, with errors predominantly occurring in the configuration of IPC protocol links and the setup for data transmission and reception. In contrast, Table 19 reveals that Llama3-8b-instruct's state coverage is heavily concentrated on the first two states,

indicating that it encounters errors as early as the configuration of necessary library files, preventing it from matching the corresponding state machine effectively.

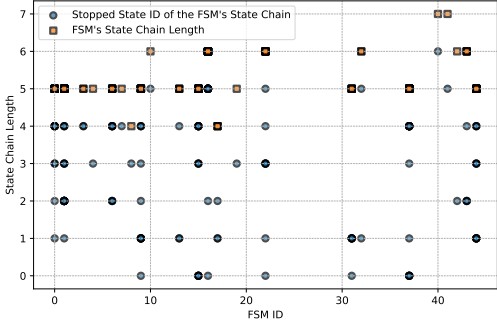

(a) Error Distribution of FSM state transitions observed in the Java IPC interaction code generated by GPT-4o.

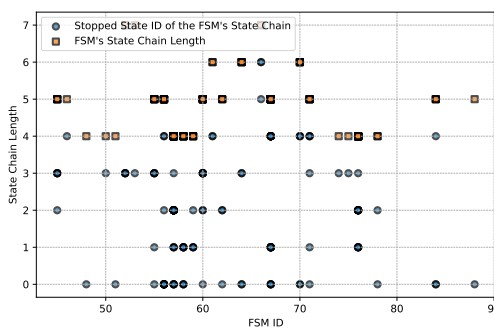

(b) Error Distribution of FSM state transitions observed in the Python IPC interaction code generated by GPT-4o.

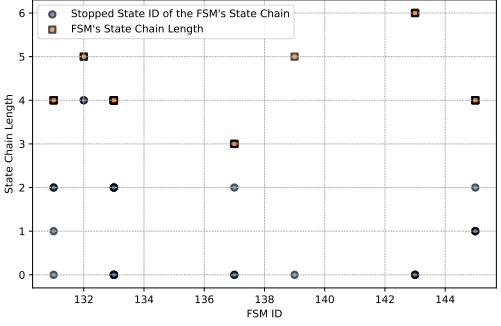

(c) Error Distribution of FSM state transitions observed in the PHP IPC interaction code generated by GPT-4o.

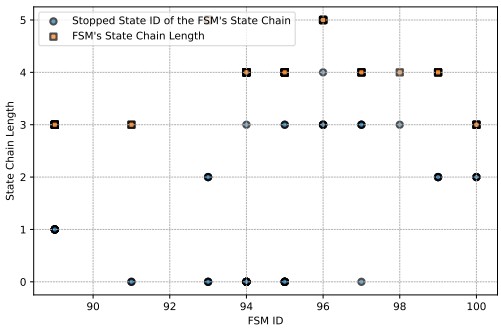

(d) Error Distribution of FSM state transitions observed in the JavaScript IPC interaction code generated by GPT-4o.

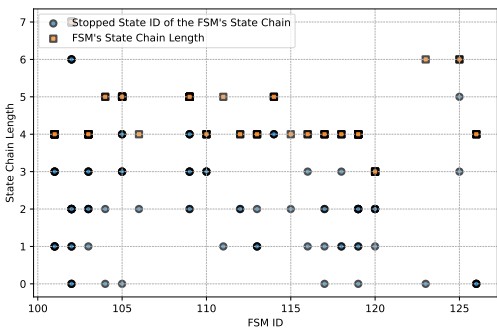

(e) Error Distribution of FSM state transitions observed in the Go IPC interaction code generated by GPT-4o.

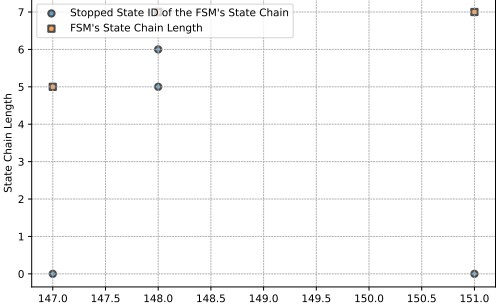

(f) Error Distribution of FSM state transitions observed in the C++ IPC interaction code generated by GPT-4o.

Figure 18: Error Distribution of FSM state transitions observed in the IPC interaction code generated by GPT-4o.

## C.2 RESULTS FOR ANSWER RQ2

Table 7 presents the Pass@1 and Pass@5 performance of various LLMs on the CrossPL-FFI benchmark. Among general-purpose models, GPT-4o achieves the highest performance (Pass@1: 19.54%, Pass@5: 26.46%), followed by Gemini-1.5-pro and DeepSeek-V3, indicating that large,

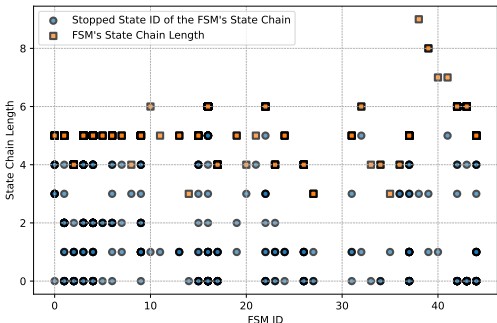

(a) Error Distribution of FSM state transitions observed in the Java IPC interaction code generated by Llama3-8b-instruct.

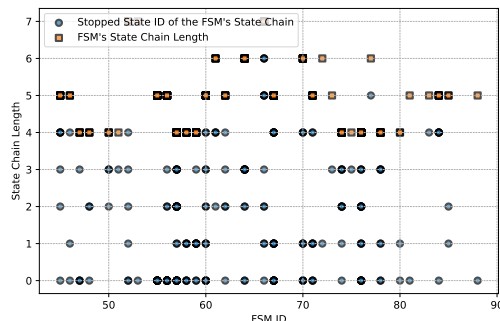

(b) Error Distribution of FSM state transitions observed in the Python IPC interaction code generated by Llama3-8b-instruct.

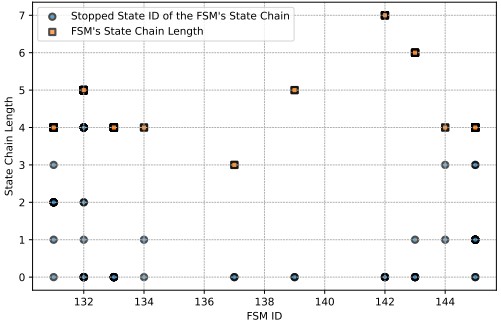

(c) Error Distribution of FSM state transitions observed in the PHP IPC interaction code generated by Llama3-8b-instruct.

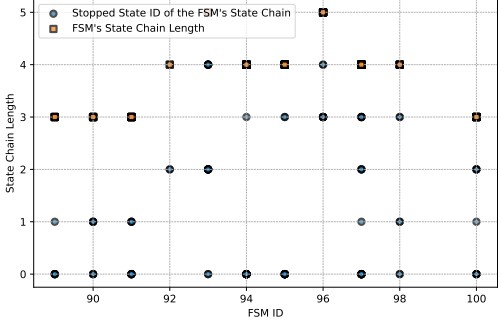

(d) Error Distribution of FSM state transitions observed in the JavaScript IPC interaction code generated by Llama3-8b-instruct.

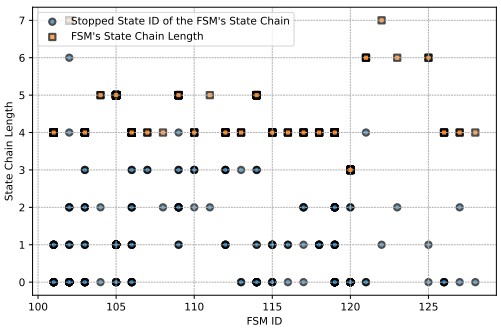

(e) Error Distribution of FSM state transitions observed in the Go IPC interaction code generated by Llama3-8b-instruct.

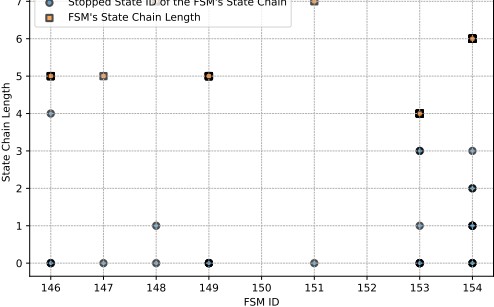

(f) Error Distribution of FSM state transitions observed in the C++ IPC interaction code generated by Llama3-8b-instruct.

Figure 19: Error Distribution of FSM state transitions observed in the IPC interaction code generated by Llama3-8b-instruct.

instruction-tuned models excel at generating Python-C FFI code. Smaller Qwen3 variants (e.g., 30B, 14B) exhibit moderate performance, while very small models (e.g., Qwen3-4b, GLM4-9b, Llama3-8b-instruct) achieve minimal success. Among code-specialized models, Qwen2.5-coder-32b-instruct outperforms other models, although their Pass@1 rates remain below 17%, highlighting the overall difficulty of FFI code generation. These results demonstrate that model size, instruction tuning, and specialization significantly influence performance, yet all models show limited absolute success, suggesting substantial room for improvement in LLM-based FFI generation.

Table 7: Pass@1 and Pass@5 results of various models on CrossPL-FFI

| Model Type | Model | Published Time | Size | Pass@1 ↑ | Pass@5 ↑ | 95%CI of Pass@5 |
|---|---|---|---|---|---|---|
| General Model | GPT-4o | 2024 | \ | **19.54%** | **26.46%** | [22.90%, 30.10%] |
| | GPT-4o-mini | 2024 | \ | 9.77% | 13.74% | [11.04%, 16.59%] |
| | Gemini-1.5-pro | 2024 | \ | 16.86% | 21.18% | [17.83%, 24.63%] |
| | GLM4-plus | 2024 | \ | 8.62% | 13.40% | [10.72%, 16.23%] |
| | DeepSeek-V3 | 2024 | 671B | 16.67% | 21.50% | [18.23%, 25.10%] |
| | Qwen3-235b-a22b | 2025 | 235B | 13.03% | 21.28% | [18.10%, 24.52%] |
| | Qwen3-32b | 2025 | 32B | 13.22% | 19.92% | [16.63%, 23.23%] |
| | Qwen3-30b-a3b | 2025 | 30B | 9.77% | 14.32% | [11.50%, 17.26%] |
| | Qwen3-14b | 2025 | 14B | 11.49% | 16.69% | [13.64%, 19.83%] |
| | Qwen3-8b | 2025 | 8B | 9.77% | 12.06% | [9.48%, 14.86%] |
| | Qwen3-4b | 2025 | 4B | 5.94% | 9.68% | [7.28%, 12.23%] |
| | GLM4-9b | 2024 | 9B | 4.60% | 7.92% | [5.79%, 10.20%] |
| | Lamma3-8b-instruct | 2024 | 8B | 0.77% | 3.95% | [2.59%, 5.48%] |
| | Gemma-7b | 2024 | 7B | 1.34% | 4.32% | [2.83%, 5.89%] |
| Code Model | CodeGeeX4 | 2024 | \ | 4.60% | 6.99% | [4.96%, 9.16%] |
| | Qwen2.5-coder-32b-insrtuct | 2024 | 32B | 7.28% | 14.38% | [11.50%, 17.29%] |
| | Qwen2.5-coder-14b-insrtuct | 2024 | 14B | 4.80% | 7.64% | [3.04%, 12.89%] |
| | Qwen2.5-coder-7b-instruct | 2024 | 7B | 4.80% | 8.36% | [6.19%, 10.73%] |
| | CodeLlama-7b-instruct | 2023 | 7B | 1.72% | 5.69% | [4.09%, 7.52%] |
| | CodeGemma-7b | 2024 | 7B | 4.41% | 8.20% | [6.10%, 10.49%] |

## C.3   RESULTS FOR ANSWER RQ3

As described in  subsection 4.3, this section provides a detailed analysis of the experimental data for RQ3. Tables 8–10 report the performance of LLMs on CrossPL.

**Model scale effect:** Larger models consistently outperform smaller ones across programming languages and network protocols. For instance, Qwen3-235B achieves the highest Pass@1 scores on Java (77.40%), C++ (88.24%), and gRPC (86.82%), whereas smaller models (4B–8B) show substantially lower performance.

**Think mode impact:** Enabling think mode yields limited or inconsistent improvement in IPC tasks. In some cases, it slightly decreases performance (e.g., Qwen3-235B Java: 75.12% vs 77.40%), suggesting that structured IPC patterns do not strongly benefit from additional intermediate reasoning.

**Language/protocol dependency:** The models perform better on mainstream languages (Java, Python, C++) and commonly used protocols (*TCP*, *gRPC*), while less common languages or protocols (Go, *Pipe*) exhibit lower scores.

**Overall difficulty:** Pass@1 scores for FFI tasks are substantially lower than for IPC, with the highest being  17–18%, reflecting the inherent challenges of library linking, type specification, and memory management in FFI.

**Think mode efficacy:** Think mode significantly enhances performance on FFI tasks. For example, Qwen3-32B improves from 13.22% to 17.81%, indicating that explicit intermediate reasoning helps manage the complexity of cross-language function calls and low-level operations.

**Model size effect:** Larger models benefit more from think mode in FFI tasks, demonstrating that scale amplifies the model's ability to utilize reasoning steps for complex scenarios.

**Conclusion:** The experimental results indicate that Qwen3 models achieve higher performance on IPC than FFI, with large models consistently outperforming smaller ones. Think mode confers a substantial advantage in FFI tasks, which require reasoning over library linking and memory management, but offers minimal benefit for IPC tasks, where communication patterns are more structured and captured adequately in the training data. Overall, model scale and task complexity modulate the effectiveness of intermediate reasoning, highlighting the need for task-specific strategies when generating cross-language interoperable code.

Table 8: pass@1 results across Qwen3 series models on IPC-related tasks

| Model | Published Time | Size | Java ↑ | Python ↑ | Go ↑ | JavaScript ↑ | PHP ↑ | C++ ↑ | Mean ↑ |
|---|---|---|---|---|---|---|---|---|---|
| Qwen3-235b-a22b-thinking | 2025 | 235B | 75.12% | 67.73% | 61.48% | 76.01% | 74.83% | 82.35% | 72.92% |
| Qwen3-32b-thinking | 2025 | 32B | 61.14% | 60.36% | 60.97% | 72.32% | 72.19% | 82.35% | 68.22% |
| Qwen3-30b-a3b-thinking | 2025 | 30B | 50.33% | 55.18% | 52.04% | 70.85% | 73.51% | 78.43% | 63.39% |
| Qwen3-14b-thinking | 2025 | 14B | 57.07% | 62.75% | 58.67% | 69.74% | 68.87% | 72.55% | 64.94% |
| Qwen3-8b-thinking | 2025 | 8B | 24.72% | 55.58% | 46.94% | 69.74% | 70.20% | 66.67% | 55.64% |
| Qwen3-4b-thinking | 2025 | 4B | 33.82% | 57.77% | 50.51% | 61.62% | 52.32% | 70.59% | 54.44% |
| Qwen3-235b-a22b | 2025 | 235B | **77.40%** | **67.93%** | **65.31%** | 76.38% | 74.83% | **88.24%** | **75.02%** |
| Qwen3-32b | 2025 | 32B | 70.89% | 62.75% | 49.74% | 74.17% | 75.50% | 80.39% | 68.91% |
| Qwen3-30b-a3b | 2025 | 30B | 57.24% | 51.99% | 49.75% | 70.48% | 71.52% | 66.67% | 61.28% |
| Qwen3-14b | 2025 | 14B | 67.64% | 57.57% | 59.44% | **77.12%** | **76.82%** | 70.59% | 68.20% |
| Qwen3-8b | 2025 | 8B | 10.73% | 55.78% | 48.47% | 68.63% | 68.21% | 64.71% | 52.76% |
| Qwen3-4b | 2025 | 4B | 54.96% | 57.97% | 52.81% | 66.79% | 57.62% | 64.71% | 59.14% |

Table 9: Performance of Qwen3 series models across network protocols (pass@1)

| Model | Published Time | Size | HTTP ↑ | TCP ↑ | UDP ↑ | Websocket ↑ | Pipe ↑ | gRPC ↑ | Message Queue ↑ | Mean ↑ |
|---|---|---|---|---|---|---|---|---|---|---|
| Qwen3-235b-a22b-think | 2025 | 235B | 64.96% | 76.67% | 79.35% | 73.74% | 64.05% | 82.17% | 72.22% | 73.31% |
| Qwen3-32b-think | 2025 | 32B | 58.28% | 70.44% | 70.65% | 71.72% | 45.10% | 80.62% | 63.64% | 65.78% |
| Qwen3-30b-a3b-think | 2025 | 30B | 56.10% | 60.28% | 65.22% | 64.14% | 48.37% | 66.67% | 54.04% | 59.26% |
| Qwen3-14b-think | 2025 | 14B | 55.58% | 66.51% | 58.70% | 72.73% | 54.25% | 79.07% | 61.62% | 64.07% |
| Qwen3-8b-think | 2025 | 8B | 50.19% | 41.80% | 45.65% | 59.60% | 35.29% | 54.26% | 44.44% | 47.32% |
| Qwen3-4b-think | 2025 | 4B | 51.86% | 49.19% | 43.48% | 50.51% | 32.68% | 61.24% | 46.46% | 47.92% |
| Qwen3-235b-a22b | 2025 | 235B | **67.01%** | **77.60%** | **83.70%** | 74.75% | **65.36%** | **86.82%** | 72.22% | **75.35%** |
| Qwen3-32b | 2025 | 32B | 66.20% | 69.48% | 77.03% | **84.52%** | 54.11% | 81.58% | **73.91%** | 72.40% |
| Qwen3-30b-a3b | 2025 | 30B | 52.76% | 64.43% | 60.87% | 62.63% | 56.86% | 56.59% | 56.06% | 58.60% |
| Qwen3-14b | 2025 | 14B | 59.44% | 72.98% | 70.65% | 72.73% | 51.63% | 76.74% | 67.17% | 67.33% |
| Qwen3-8b | 2025 | 8B | 49.55% | 29.79% | 35.87% | 52.53% | 35.95% | 62.79% | 35.35% | 43.12% |
| Qwen3-4b | 2025 | 4B | 54.43% | 62.12% | 63.04% | 54.04% | 48.37% | 68.22% | 59.09% | 58.47% |

### C.4 EFFECTS OF MODEL SCALE IN THE TWO BENCHMARK PIPELINES

Intuitively, in the IPC-related pipeline, smaller or less powerful LLMs could likely replace the Judger and "Instructor" components, as the associated reasoning and instruction-generation tasks are relatively simple. However, function-level and class-level code extraction still require stronger instruction-following and structural reasoning capabilities, which are better handled by large models such as DeepSeek-V3 used in our original framework.

In the IPC-related pipeline, the Judger module only needs to determine whether the input file contains IPC-related logic and produce a short categorical output (e.g., "Yes," "Function-level," or "Class-level"). This step requires very few output tokens and minimal reasoning depth. Although smaller models could, in principle, achieve similar decision accuracy, they are more prone to false positives. Such false positives would propagate into the downstream extraction stage and dramatically increase the number of unnecessary extraction attempts. For this reason, we recommend keeping the Judger as a stronger model to maintain pipeline efficiency and avoid computational overhead caused by spurious matches. In contrast, the FFI-related pipeline demands that the LLM simultaneously analyze low-level C code, design corresponding tasks, generate canonical solutions, and produce assertion-based test cases. These multi-step reasoning and synthesis tasks rely more heavily on advanced models.

As a potential extension, we conducted an ablation study to quantify the impact on the number and quality of generated samples across both IPC-related and FFI-related pipelines.

Table 10: Performance of Qwen3 series models on CrossPL-FFI (pass@1)

| Model | Published Time | Size | Pass@1 |
|---|---|---|---|
| Qwen3-235b-a22b-think | 2025 | 235B | 17.24% |
| Qwen3-32b-think | 2025 | 32B | **17.81%** |
| Qwen3-30b-a3b-think | 2025 | 30B | 11.69% |
| Qwen3-14b-think | 2025 | 14B | 12.46% |
| Qwen3-8b-think | 2025 | 8B | 13.22% |
| Qwen3-4b-think | 2025 | 4B | 8.05% |
| Qwen3-235b-a22b | 2025 | 235B | 13.03% |
| Qwen3-32b | 2025 | 32B | 13.22% |
| Qwen3-30b-a3b | 2025 | 30B | 9.77% |
| Qwen3-14b | 2025 | 14B | 11.49% |
| Qwen3-8b | 2025 | 8B | 9.77% |
| Qwen3-4b | 2025 | 4B | 5.94% |

For the ablation studies, we designed three experimental settings to isolate the impact of model size in specific pipeline components: ❶ "Extractor"-only replacement: We replace the "Extractor" with a smaller model (Qwen3-4B) while keeping all other components, including the "Instructor" and Judger, powered by DeepSeek-V3. ❷ "Instructor"-only replacement: We replace only the "Instructor" with Qwen3-4B, while maintaining DeepSeek-V3 for extraction, judging, and verification-related reasoning. ❸ In the FFI sample construction pipeline, we replaced the "Generator" with a smaller model (Qwen3-4B).

These ablations will quantify how much smaller models affect both the quantity and quality of generated benchmark samples across IPC and FFI pipelines.

From the experimental results, we observe that replacing the "Extractor" with Qwen3-4B substantially reduces the number of successfully constructed samples. The original pipeline produced 1,982 samples, whereas the replacement yielded only 1,723 samples—a decrease of 259. In contrast, replacing the "Instructor" with Qwen3-4B has a much smaller impact. The original pipeline also produced 1,982 samples, and the modified version generated 1,890 samples, reducing the total by only 72.

However, substituting the "Generator" with Qwen3-4B severely degrades performance in the FFI pipeline: the number of valid samples dropped from 522 to just 65. This result highlights that constructing high-quality FFI tasks requires a stronger and more consistent large-scale model.We sincerely appreciate the reviewer's insightful question, which helped us better analyze and clarify the role of model size in different components of the pipeline.

## C.5 RISKS OF DATA LEAKAGE IN CPL CODE GENERATION BENCHMARKS

To address potential concerns regarding temporal contamination, we clarify how time-based considerations were incorporated in the construction of both subsets of CrossPL.

Table 11: Instance counts before and after "Extractor"/"Instructor" replacement

| Language | Original | "Extractor" replacement (Replaced by Qwen3-4B) | "Instructor" replacement (Replaced by Qwen3-4B) |
|---|---|---|---|
| C++ | 51 | 52 | 52 |
| Go | 392 | 353 | 326 |
| Java | 615 | 606 | 620 |
| JavaScript | 271 | 157 | 262 |
| PHP | 151 | 159 | 162 |
| Python | 502 | 396 | 468 |

Table 12: Instance counts before and after "Generator" replacement

| Language Pair | Original | "Generator" replacement (Replaced by Qwen3-4B) |
|---|---|---|
| Python–C | 522 | 65 |

(1) Temporal analysis in the construction of CrossPL-IPC. The IPC subset is built from real-world open-source repositories on GitHub, and therefore requires explicit temporal validation. We categorized all IPC samples into four time buckets: before 2022, 2022, 2023, and 2024–2025. Notably, the majority of IPC instances (1377 samples, 69.48% of the IPC subset) originate from 2024–2025. Table 13 show the counts of CrossPL-IPC Samples with different Release Year.

We then evaluated model performance across these temporal slices. The results (table 14 and figure 20) show no evidence that models perform systematically better on the newer subsets; in fact, model scores on older samples do not exceed those on more recent ones. This suggests that temporal leakage does not materially inflate performance in the IPC tasks.

(2) Contamination mitigation in the construction of CrossPL-FFI. For the FFI subset, we deliberately designed the pipeline to avoid exposure to pre-existing code. All FFI samples are constructed in a self-contained and self-guided manner: the LLM is shown a standalone C function and is asked to ❶ generate a task description, ❷ design a Python wrapper using ctypes, and ❸ produce a canonical reference solution.

To further quantify the risk of training-data contamination, we measured the overlap between our prompts and the Pygsl codebase used during verification. As shown in the Table 15, the overlap is extremely low—0.61% for function names and 0.99% for class names—indicating minimal risk that LLMs benefited from prior exposure to these specific instances. We thank the reviewer for raising this question, as it prompted us to conduct a more systematic examination of temporal leakage and contamination across both IPC and FFI pipeline.

Table 13: Number of CrossPL-IPC Samples with different Release Year

| Number of samples before 2022 | Number of samples in 2022 | Number of samples in 2023 | Number of samples in 2024-2025 |
|---|---|---|---|
| 222 | 143 | 240 | 1377 |

## C.6 EFFECTS OF SAMPLING TEMPERATURE AND TOP-P ON CPL CODE GENERATION

To assess how decoding hyperparameters (temperature and top-p) influence cross-language code generation, we evaluated eight locally deployable models (Qwen3-4B, Qwen3-8B, GLM4-9B, Llama3-8B-Instruct, Gemma-7B, Qwen2.5-coder-7B, CodeLlama-7B-Instruct, CodeGemma-7B) on two CrossPL subsets: CrossPL-IPC (Java) and CrossPL-FFI. Two controlled experiments were conducted:

- A temperature sweep with top-p fixed at 0.95 and temperature $\in \{0.2, 0.4, 0.6, 0.8, 1.0, 1.2\}$;

- A top-p sweep with temperature fixed at 0.2 and top-p $\in \{0.1, 0.3, 0.5, 0.7, 0.9, 1.0\}$.

Table 14: Pass@1 Performance Across Release Years

| Model | Before 2022 | 2022 | 2023 | 2024–2025 |
|---|---|---|---|---|
| GPT-4o | 70.27% | 81.12% | 75.42% | 73.38% |
| GPT-4o-mini | 64.41% | 80.42% | 64.17% | 63.11% |
| Gemini-1.5-pro | 65.77% | 80.42% | 69.58% | 69.64% |
| GLM4-plus | 71.62% | 83.22% | 75.42% | 75.89% |
| DeepSeek-V3 | 72.97% | 81.82% | 72.08% | 72.98% |
| Qwen3-235b-a22b | 69.55% | 81.43% | 76.69% | 70.79% |
| Qwen3-32b | 63.64% | 77.86% | 64.83% | 64.34% |
| Qwen3-30b-a3b | 59.09% | 70.00% | 58.05% | 55.58% |
| Qwen3-14b | 64.09% | 77.14% | 66.53% | 64.19% |
| Qwen3-8b | 50.45% | 32.14% | 38.98% | 43.22% |
| Qwen3-4b | 55.00% | 68.57% | 62.29% | 55.43% |
| GLM4-9b | 59.91% | 78.32% | 61.25% | 59.48% |
| Lamma3-8b-instruct | 43.05% | 39.44% | 49.58% | 38.85% |
| Gemma-7b | 51.12% | 59.86% | 58.75% | 47.86% |
| CodeGeeX4 | 62.61% | 81.12% | 70.42% | 63.04% |
| Qwen2.5-coder-32b-instruct | 65.77% | 83.92% | 72.03% | 72.48% |
| Qwen2.5-coder-14b-instruct | 67.12% | 78.32% | 71.67% | 70.81% |
| Qwen2.5-coder-7b-instruct | 66.22% | 79.02% | 69.58% | 68.19% |
| CodeLlama-7b-instruct | 52.47% | 66.90% | 56.67% | 46.91% |
| CodeGemma-7b | 55.16% | 69.72% | 59.17% | 53.30% |
| Mean | 60.63% | 73.08% | 64.98% | 60.88% |

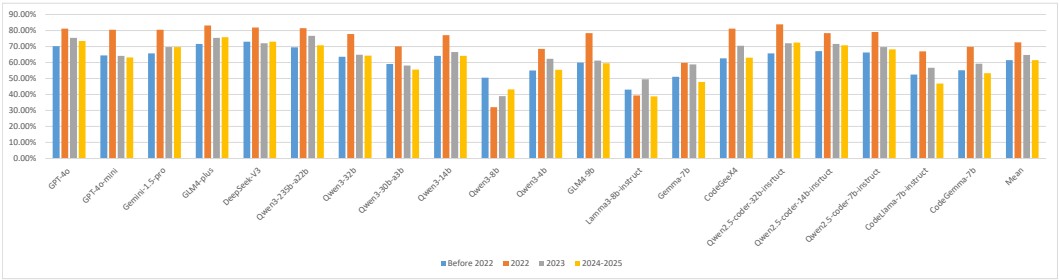

Figure 20: Classification of CrossPL-IPC Samples by Release Year

Table 15: Overlap statistics between pygsl and CrossPL-FFI

| Metric | Count and overlap Rate |
|---|---|
| Functions in pygsl | 4,510 |
| Functions in FFI-bench | 14,822 |
| Functions in both pygsl and FFI-bench | 90 |
| Function overlap rate | 0.61% |
| Classes in pygsl | 868 |
| Classes in FFI-bench | 304 |
| Classes in both pygsl and FFI-bench | 3 |
| Class overlap rate | 0.99% |

Performance was measured using Pass@1 (computed with $n = 5$ and $k = 1$), and the results are reported in Tables 16- 19.

Across both experiments, several consistent conclusions emerge. First, neither temperature nor top-p exhibits a linear or stable influence on model performance. Adjusting sampling parameters changes model behavior, but the effects are irregular and strongly model-dependent.

Second, all models perform substantially better on IPC than on FFI. IPC tasks follow well-defined communication protocols, resulting in a constrained solution space. In contrast, FFI tasks require low-level reasoning over function signatures, ABI conventions, type conversions, and memory layout; even small errors may lead to undefined behavior. This makes FFI generation considerably more difficult. For example, the average Pass@1 of Qwen3-4B decreases from 55.24% on IPC to only 3.08% on FFI across different temperatures.

Third, parameter tuning has limited impact on FFI performance. Increasing temperature or top-p does not enhance creativity in a beneficial way, and lowering them does not improve output stability. Both strategies fail to overcome the inherent difficulty of FFI generation. Meaningful improvements will likely require enhanced cross-language alignment and domain-specific fine-tuning with FFI-relevant knowledge.

Finally, optimal sampling settings vary across models. These differences reflect variations in training corpora, alignment procedures, and the extent to which each model has been exposed to cross-language code during training.

Table 16: Performance of different models on CrossPL-IPC (Java) under varying temperature (fixed top-p=0.95). Pass@1 computed with n = 5 and k = 1. The values in parentheses denote the 95% confidence intervals for pass@1.

| Model | T = 0.4 | T = 0.6 | T = 0.8 | T = 1.0 | T = 1.2 |
|---|---|---|---|---|---|
| Qwen3-4b | 54.50% | 58.41% | 53.92% | 55.25% | 53.82% |
| | ([51.28%, 57.79%]) | ([55.19%, 61.63%]) | ([50.47%, 57.24%]) | ([51.90%, 58.60%]) | ([50.44%, 57.14%]) |
| Qwen3-8b | 10.08% | 10.34% | 10.96% | 11.54% | 12.55% |
| | ([8.16%, 12.03%]) | ([8.42%, 12.36%]) | ([9.01%, 12.91%]) | ([9.59%, 13.56%]) | ([10.57%, 14.54%]) |
| GLM4-9b | 51.58% | 52.39% | 51.80% | 52.07% | 52.72% |
| | ([48.33%, 54.80%]) | ([49.23%, 55.58%]) | ([48.68%, 54.96%]) | ([48.91%, 55.25%]) | ([49.59%, 55.93%]) |
| Llama3-8B-Instruct | 33.37% | 32.94% | 32.29% | 33.37% | 33.40% |
| | ([30.67%, 36.00%]) | ([30.34%, 35.61%]) | ([29.69%, 34.93%]) | ([30.73%, 36.03%]) | ([30.73%, 36.10%]) |
| Gemma-7B | 52.20% | 50.31% | 49.95% | 51.06% | 50.44% |
| | ([48.85%, 55.48%]) | ([46.99%, 53.66%]) | ([46.73%, 53.30%]) | ([47.71%, 54.47%]) | ([47.06%, 53.79%]) |
| Qwen2.5-coder-7b | 66.89% | 66.11% | 66.80% | 66.67% | 67.77% |
| | ([63.74%, 70.02%]) | ([62.93%, 69.27%]) | ([63.61%, 69.92%]) | ([63.45%, 69.79%]) | ([64.52%, 70.96%]) |
| CodeLlama-7B-Instruct | 55.90% | 56.23% | 57.40% | 55.97% | 56.16% |
| | ([52.81%, 58.89%]) | ([53.17%, 59.19%]) | ([54.37%, 60.39%]) | ([52.98%, 59.06%]) | ([53.14%, 59.12%]) |
| CodeGemma-7B | 52.20% | 52.91% | 53.07% | 53.63% | 52.52% |
| | ([48.84%, 55.64%]) | ([49.50%, 56.29%]) | ([49.66%, 56.52%]) | ([50.24%, 57.14%]) | ([49.17%, 55.87%]) |

Table 17: Performance of different models on CrossPL-IPC (Java) under varying top-p (fixed temperature=0.2). Pass@1 computed with n = 5 and k = 1. The values in parentheses denote the 95% confidence intervals for pass@1.

| Model | Top-p = 0.3 | Top-p = 0.5 | Top-p = 0.7 | Top-p = 0.9 | Top-p = 1.0 |
|---|---|---|---|---|---|
| Qwen3-4b | 54.31% | 54.76% | 54.83% | 55.06% | 54.41% |
| | ([50.44%,58.05%]) | ([50.89%,58.57%]) | ([51.02%,58.60%]) | ([51.35%,58.86%]) | ([50.70%,58.02%]) |
| Qwen3-8b | 9.17% | 9.11% | 9.63% | 9.72% | 9.92% |
| | ([6.99%,11.45%]) | ([6.93%,11.28%]) | ([7.41%,11.87%]) | ([7.61%,11.90%]) | ([7.77%,12.10%]) |
| GLM4-9b | 65.59% | 65.37% | 65.24% | 64.72% | 64.07% |
| | ([61.79%,69.20%]) | ([61.53%,68.98%]) | ([61.46%,68.72%]) | ([61.10%,68.20%]) | ([60.62%,67.48%]) |
| Llama3-8B-Instruct | 34.63% | 31.80% | 32.94% | 34.05% | 33.17% |
| | ([32.07%,37.27%]) | ([29.33%,34.31%]) | ([30.31%,35.58%]) | ([31.32%,36.72%]) | ([30.54%,35.74%]) |
| Gemma-7B | 51.61% | 51.09% | 51.51% | 51.54% | 52.13% |
| | ([48.33%,54.86%]) | ([47.71%,54.41%]) | ([48.03%,54.89%]) | ([48.23%,54.93%]) | ([48.88%,55.51%]) |
| Qwen2.5-coder-7b | 67.87% | 67.67% | 68.13% | 68.03% | 68.39% |
| | ([64.10%,71.54%]) | ([63.97%,71.32%]) | ([64.55%,71.58%]) | ([64.62%,71.38%]) | ([65.01%,71.71%]) |
| CodeLlama-7B-Instruct | 57.17% | 56.23% | 55.35% | 56.33% | 56.29% |
| | ([54.18%,60.10%]) | ([53.20%,59.22%]) | ([52.36%,58.37%]) | ([53.30%,59.22%]) | ([53.20%,59.32%]) |
| CodeGemma-7B | 54.86% | 53.92% | 53.69% | 54.37% | 54.80% |
| | ([51.54%,58.24%]) | ([50.67%,57.30%]) | ([50.37%,57.01%]) | ([51.09%,57.63%]) | ([51.51%,58.24%]) |

Table 18: Performance of different models on CrossPL-FFI under varying temperature (fixed top-p=0.95). Pass@1 computed with n = 5 and k = 1. The values in parentheses denote the 95% confidence intervals for pass@1.

| Model | T=0.4 | T=0.6 | T=0.8 | T=1.0 | T=1.2 |
|---|---|---|---|---|---|
| Qwen3-4b | 2.72% | 3.72% | 2.80% | 3.22% | 2.84% |
| | ([1.88%, 3.72%]) | ([2.57%, 4.98%]) | ([1.92%, 3.75%]) | ([2.22%, 4.37%]) | ([1.88%, 3.95%]) |
| Qwen3-8b | 6.93% | 6.21% | 6.67% | 6.48% | 6.59% |
| | ([5.17%, 8.81%]) | ([4.48%, 8.05%]) | ([4.90%, 8.62%]) | ([4.22%, 9.57%]) | ([4.87%, 8.47%]) |
| GLM4-9b | 1.00% | 1.30% | 0.73% | 1.03% | 0.80% |
| | ([0.50%, 1.57%]) | ([0.65%, 2.07%]) | ([0.34%, 1.23%]) | ([0.54%, 1.61%]) | ([0.38%, 1.30%]) |
| Llama3-8B-Instruct | 1.11% | 1.15% | 1.11% | 0.96% | 1.07% |
| | ([0.61%, 1.69%]) | ([0.65%, 1.69%]) | ([0.61%, 1.72%]) | ([0.54%, 1.46%]) | ([0.61%, 1.57%]) |
| Gemma-7B | 1.38% | 1.88% | 1.57% | 1.69% | 1.84% |
| | ([0.73%, 2.15%]) | ([1.07%, 2.80%]) | ([0.88%, 2.41%]) | ([0.96%, 2.49%]) | ([1.03%, 2.72%]) |
| Qwen2.5-coder-7b | 4.25% | 4.06% | 4.18% | 3.79% | 3.75% |
| | ([3.10%, 5.52%]) | ([2.91%, 5.33%]) | ([3.03%, 5.44%]) | ([2.68%, 5.02%]) | ([2.72%, 4.87%]) |
| CodeLlama-7B-Instruct | 1.72% | 1.49% | 1.95% | 1.76% | 1.69% |
| | ([1.03%, 2.49%]) | ([0.92%, 2.15%]) | ([1.26%, 2.72%]) | ([1.07%, 2.57%]) | ([1.00%, 2.45%]) |
| CodeGemma-7B | 3.98% | 4.25% | 3.33% | 3.49% | 3.79% |
| | ([2.80%, 5.29%]) | ([2.99%, 5.67%]) | ([2.26%, 4.52%]) | ([2.34%, 4.71%]) | ([2.61%, 5.13%]) |

Table 19: Performance of different models on CrossPL-FFI under varying top-p (fixed temperature=0.2). Pass@1 computed with n = 5 and k = 1. The values in parentheses denote the 95% confidence intervals for pass@1.

| Model | Top-p = 0.3 | Top-p = 0.5 | Top-p = 0.7 | Top-p = 0.9 | Top-p = 1.0 |
|---|---|---|---|---|---|
| Qwen3-4b | 6.63% | 6.25% | 6.44% | 6.25% | 6.90% |
| | ([4.71%, 8.77%]) | ([4.37%, 8.31%]) | ([4.56%, 8.47%]) | ([4.41%, 8.20%]) | ([4.98%, 8.97%]) |
| Qwen3-8b | 9.66% | 10.00% | 9.66% | 9.50% | 9.77% |
| | ([7.20%, 12.34%]) | ([7.82%, 12.99%]) | ([7.28%, 12.26%]) | ([7.20%, 12.03%]) | ([7.43%, 12.30%]) |
| GLM4-9b | 1.23% | 0.80% | 0.96% | 1.07% | 1.42% |
| | ([0.69%, 1.84%]) | ([0.38%, 1.30%]) | ([0.46%, 1.53%]) | ([0.57%, 1.65%]) | ([0.73%, 2.26%]) |
| Llama3-8B-Instruct | 1.46% | 1.15% | 0.77% | 0.57% | 0.65% |
| | ([0.96%, 1.99%]) | ([0.65%, 1.76%]) | ([0.38%, 1.23%]) | ([0.27%, 0.96%]) | ([0.27%, 1.11%]) |
| Gemma-7B | 1.34% | 2.15% | 1.92% | 1.49% | 1.80% |
| | ([0.69%, 2.07%]) | ([1.23%, 3.18%]) | ([1.11%, 2.84%]) | ([0.80%, 2.30%]) | ([1.03%, 2.68%]) |
| Qwen2.5-coder-7b | 3.79% | 4.79% | 3.41% | 3.68% | 3.79% |
| | ([2.68%, 5.06%]) | ([3.49%, 6.21%]) | ([2.45%, 4.48%]) | ([2.64%, 4.87%]) | ([2.76%, 4.94%]) |
| CodeLlama-7B-Instruct | 1.26% | 1.92% | 1.38% | 1.57% | 1.84% |
| | ([0.73%, 1.88%]) | ([1.19%, 2.76%]) | ([0.80%, 2.03%]) | ([0.96%, 2.26%]) | ([1.11%, 2.64%]) |
| CodeGemma-7B | 3.75% | 3.83% | 3.64% | 3.98% | 2.95% |
| | ([2.61%, 5.06%]) | ([2.68%, 5.10%]) | ([2.49%, 4.94%]) | ([2.80%, 5.29%]) | ([1.95%, 4.06%]) |

# D    CASES STUDY

## D.1    CASES STUDY OF CROSSPL-IPC

Fig. 21 illustrates a sample alongside its corresponding "Instruction", "Canonical result", and an incorrect answer generated by DeepSeek-V3. The task involves implementing a *gRPC* server using the grpcio library. Based on the official documentation, we have summarized the general procedure for such problems, represented as the semantic descriptions of the various states in the FSM: (1) Import the necessary libraries required for server creation; (2) Create a *gRPC* server with a thread pool executor to support concurrency; (3) Bind the server to a port using the *add_insecure_port* method; (4) Start the server to begin listening for incoming *gRPC* requests.

From the "Canonical solution" shown on the middle part of Fig. 21, the implementation fully complies with this standard and passes validation by the predefined FSM. In contrast, the incorrect answer on the lower part lacks the critical step of binding the server to a port using *add_insecure_port*. Consequently, the server does not listen on any port, rendering it incapable of receiving requests and thus functionally ineffective.

Notably, although the instruction does not explicitly mention the binding operation, it states: "Initialize and start a *gRPC* server capable of handling concurrent *gRPC* requests." This implicitly requires binding the *gRPC* server to a port. In practice, software engineers seldom specify every detail, as enumerating all functional steps often takes longer than implementing the functionality itself.

## D.2    ERROR TYPE DISTRIBUTION ANALYSIS ON CROSSPL-FFI

As described in  subsection 4.2, this subsection presents an overall error type distribution analysis on CrossPL-FFI. Fig. 22 presents the distribution of the error types observed in GPT-4o-generated Python-C FFI interaction code. The histogram shows the frequency of different error categories, including symbol resolution errors, GSL runtime errors, Python-level calling errors, undefined symbols (NameError), assertion/test failures, memory/crash errors. GPT-4o, the best-performing model, exhibits errors primarily in symbol resolution errors (111), Python-level calling errors (84), and NameError/undefined symbols (94), while other errors such as GSL runtime errors (57), assertion/test failures (51), and memory/crash errors (54) occur less frequently. Overall, the error distribution is relatively balanced, without any single category dominating. Llama3-8b-instruct, the worst-performing model, shows a more extreme distribution: Python-level calling errors (212) and symbol resolution errors (166) account for the vast majority, with NameError/undefined symbols

"Instruction":
"Task Description:
Create a function to set up and tear down a gRPC server with configurable interceptors and thread pool workers.
Function Description: The function "_set_up " initializes and starts a gRPC server with optional interceptors, while "_tear_down" gracefully stops the server. The server is configured to handle gRPC requests concurrently using a thread pool.
Input:
- " interceptors " (Optional[List[grpc.ServerInterceptor]]): A list of gRPC server interceptors for request/response processing. Defaults to None.
- - " server " (grpc.Server): The gRPC server instance to be stopped (for "_tear_down").
- Output:
- - "_set_up ” returns the initialized and started "grpc.Server" instance.
- - "_tear_down" has no return value (stops the server).",

```python
import grpc
from concurrent import futures
from typing import List, Optional
def _set_up(interceptors: Optional[List[grpc.ServerInterceptor]] = None):
    server = grpc.server(
        futures.ThreadPoolExecutor(max_workers=2),
        interceptors=interceptors,
    )
    add_gRPCTestServiceServicer_to_server(TestService(), server)
    server.add_insecure_port("[::]:{}".format(PORT))
    server.start()
    return server
def _tear_down(server: grpc.Server):
    server.stop(None)
```

```python
import grpc
from concurrent import futures
from typing import List, Optional

def _set_up(interceptors: Optional[List[grpc.ServerInterceptor]] = None) -> grpc.Server:
    server = grpc.server(
        futures.ThreadPoolExecutor(max_workers=10),
        interceptors=interceptors if interceptors else []
    )
    server.start()
    return server

def _tear_down(server: grpc.Server) -> None:
    server.stop(0)
```

Figure 21: Case Analysis: the 495-th task in the CrossPL-IPC. The upper part is the task instruction; the middle part is the Canonical solution; the lower part is the wrong result generated by Deepseek-V3

(128) also significant. In contrast, GSL runtime errors (2), assertion/test failures (5), and memory/crash errors (5) are rare. This is likely occurs because Llama3-8b-instruct frequently fails during the basic function call and symbol resolution steps, preventing the code from successfully compiling and thus rarely triggering later assertion or runtime errors. Therefore, GPT-4o demonstrates greater robustness in FFI code generation, with errors relatively dispersed across categories, whereas Llama3-8b-instruct is particularly prone to errors in function calls and symbol resolution, highlighting its limitations in cross-language function interactions. Moreover, all LLMs exhibit suboptimal performance on the CrossPL-FFI benchmark, indicating substantial room for improvement.

### D.3 CASES STUDY OF CROSSPL-FFI

As described in subsection 4.2, this subsection presents a detailed case analysis of the Python-C FFI code generated by GPT-4o. We conducted a detailed case study on GPT-4o's outputs for Tasks 205 and 257 in the CrossPL-FFI benchmark. Fig. 23 presents the prompt and reference solution for Task 205, while Fig. 24 illustrates the corresponding assertion test cases and the erroneous outputs. Similarly, Fig. 25 shows the prompt and reference solution for Task 257, and Fig. 26 displays its assertion test cases along with the incorrect outputs. GPT-4o exhibited a symbol resolution failure on Task 205 and a GSL runtime error on Task 257.

#### D.3.1 TASK 205

For Task 205 in the CrossPL-FFI benchmark, GPT-4o generated a Python function intended to solve a triangular matrix system using the GSL library (`libgsl.so`) via `ctypes`. The provided implementation attempted to call the `cblas_ctrsm` function directly from `libgsl.so`. However, execution failed with the following undefined symbol error:

> **Execution Error**
>
> ```
> AttributeError:  /usr/local/gsl/lib/libgsl.so:  undefined
> symbol:  cblas_ctrsm
> ```

This failure indicates that `cblas_ctrsm` is not exported by `libgsl.so` itself but resides in the separate `libgslcblas.so` library. Despite loading `libgslcblas.so` with `RTLD_GLOBAL`, GPT-4o's solution did not correctly link the function, reflecting a limitation in the model's understanding of dynamic linking and cross-library symbol resolution in Python FFI contexts. This case illustrates that GPT-4o can produce syntactically plausible code while failing to account for library-level dependencies critical for correct execution.

#### D.3.2 TASK 257

For Task 257 in the CrossPL-FFI benchmark, GPT-4o generated a Python function intended to perform vector addition using the GSL library (`libgsl.so`) via `ctypes`. The implementation wraps Python lists into a custom `GSLVector` class and calls `gsl_vector_add` to compute the element-wise sum. During execution, the function triggered the following runtime error:

> **Execution Error**
>
> ```
> gsl:  oper_source.c:27:  ERROR: vectors must have same length
> Default GSL error handler invoked.
> ```

This failure indicates that GPT-4o's solution did not adequately validate the input vectors' lengths before calling the GSL function, which enforces strict dimension consistency. Although the function checks the Python lists' lengths at the Python level, a mismatch in the underlying GSL vectors or an improper mapping to `ctypes` arrays can still lead to runtime errors. This case highlights GPT-4o's limited handling of library-level preconditions and runtime constraints in Python–C FFI contexts, showing that syntactically correct code may still fail during execution when domain-specific invariants are not fully respected.

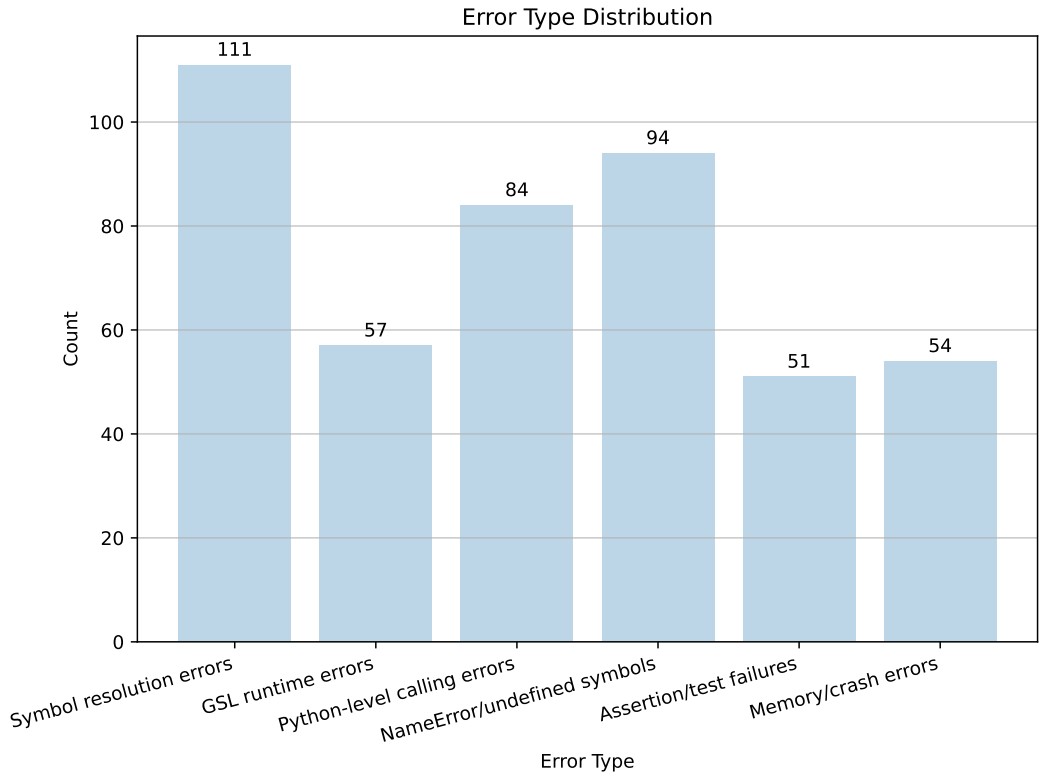

(a) Distribution of error types observed in the Python-C FFI interaction code generated by GPT-4o.

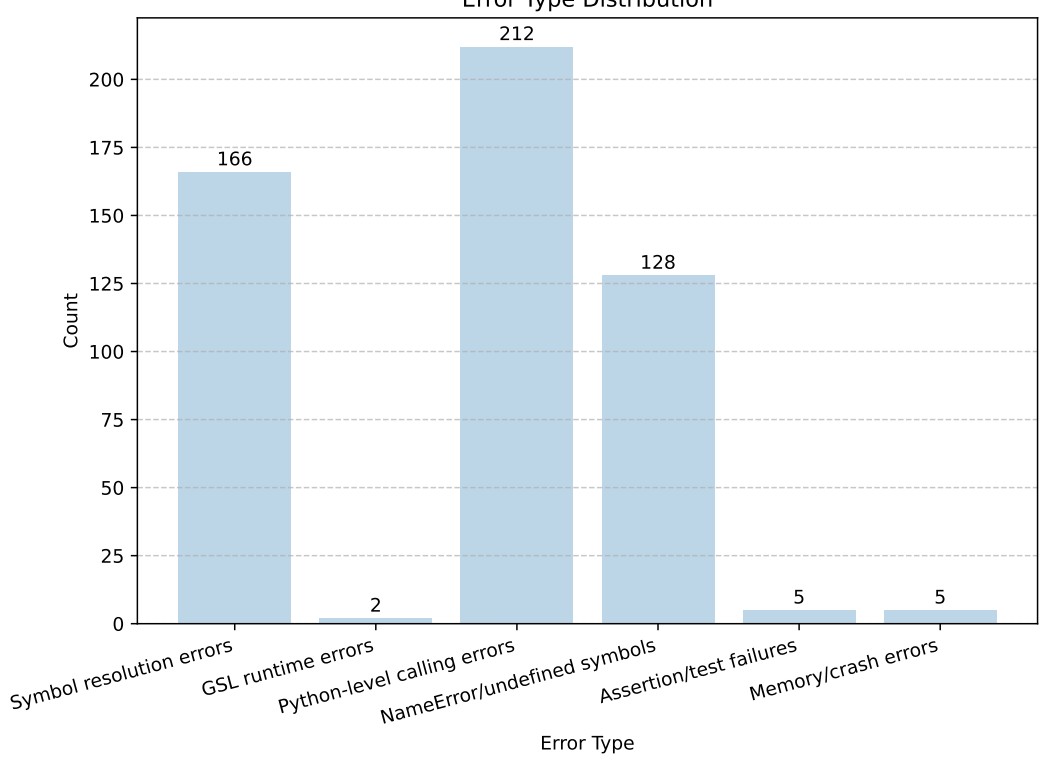

(b) Distribution of error types observed in the Python-C FFI interaction code generated by Llama3-8b-instruct.

Figure 22: Distribution of error types observed in the Python-C FFI interaction code generated by GPT-4o and Llama3-8b-instruct.

SUMMARY OF CASE STUDIES ON IPC AND FFI

The case studies presented in Sections D.1 and D.3 reveal several key limitations of current LLMs in generating cross-language interoperating code. In the IPC scenario, models often omit critical steps such as binding a *gRPC* server to a port, resulting in implementations that are syntactically valid but functionally incomplete. This reflects a tendency of LLMs to overlook implicit requirements that are not explicitly stated in the instruction but are essential for correct execution. In the FFI scenario, the observed issues include symbol resolution failures (e.g., attempting to access functions not exported by the linked library) and runtime errors due to insufficient validation of input data consistency (e.g., mismatched vector lengths in GSL operations). These cases highlight the gap between generating plausible-looking code and ensuring robust, semantically correct interoperability across programming languages.

Taken together, the findings underscore that current LLMs lack a deep understanding of language-specific APIs, library-level dependencies, and domain-specific preconditions that are indispensable in cross-language function calls.

### D.4  CASES STUDY OF THINK MODE

Figure 27 presents a representative failure case that highlights how think mode degrades model reliability in IPC-oriented code generation. Although both answers were produced by the same base model, the think-mode version consistently deviates from the protocol-specific constraints required by the task, whereas the non-think version adheres closely to the expected Boost.Asio UDP communication pattern. This contrast provides a clear window into the mechanism by which extended reasoning disrupts IPC code correctness.

In this case study, the think-mode model produced incorrect UDP client code by replacing the required send_to/receive_from datagram operations with connect/send/receive, effectively imposing a TCP-like workflow onto a UDP task. This violates both the Instruction and our FSM, which explicitly require message-oriented UDP semantics.

The underlying mechanism is that think mode encourages the model to generate longer, self-consistent reasoning chains before emitting code. While this can benefit FFI tasks—where multi-step reasoning about memory layout and type mapping is essential—it becomes harmful in IPC tasks. IPC implementations demand strict, protocol-specific API usage, but think mode tends to over-generalize and substitute higher-level "universal socket templates," introducing steps that appear logically coherent but are semantically incorrect.

As a result, excessive reasoning amplifies small early misconceptions (e.g., "UDP can use connect() like TCP"), causing the model to drift away from protocol constraints and break the precise communication logic required in IPC programming.

## E  USE OF LLMS

The LLM's role was strictly limited to improving the quality of the prose. Its contributions included:

- Assisting with grammar, spelling, and punctuation.
- Rephrasing sentences and paragraphs for better clarity, conciseness, and flow.

Write a Python function that uses ctypes to load and call the cblas_ctrsm function from the GNU Scientific Library's dynamic library. The function should perform a triangular matrix solve operation on complex matrices. The input should include the order, side, uplo, transpose, and diag parameters as integers, the dimensions M and N as integers, a complex alpha value, a complex matrix A, and a complex matrix B. The output should be the modified matrix B after the operation. Ensure the solution uses ctypes to load the libraries and call the function correctly. To accomplish the above task, the required functions along with their parameter lists are: [('triangular_matrix_solve', ['order', 'side', 'uplo', 'transA', 'diag', 'M', 'N', 'alpha', 'A', 'lda', 'B', 'ldb'])].

The recommended dynamic library loading order is as follows:
ctypes.CDLL("/usr/local/gsl/lib/libgslcblas.so", mode=ctypes.RTLD_GLOBAL)
gsl = ctypes.CDLL("/usr/local/gsl/lib/libgsl.so")

You must return ONLY a valid JSON object.
Do not include explanations, comments, or extra text.
Do not use triple quotes (""").

The JSON object must contain exactly one field:
"Candidate_solution": <string>

Inside "Candidate_solution", provide the complete code as a single string.
Escape all line breaks as \n.

```python
import ctypes
import numpy as np
# Load the libraries
ctypes.CDLL('/usr/local/gsl/lib/libgslcblas.so', mode=ctypes.RTLD_GLOBAL)
gsl = ctypes.CDLL('/usr/local/gsl/lib/libgsl.so')
# Define the cblas_ctrsm function from libgslcblas.so
cblas_ctrsm = ctypes.CDLL('/usr/local/gsl/lib/libgslcblas.so').cblas_ctrsm
cblas_ctrsm.argtypes = [ctypes.c_int,
        ctypes.c_int,
        ctypes.c_int,
        ctypes.c_int,
        ctypes.c_int,
                ctypes.c_int,
        ctypes.c_int,
        ctypes.POINTER(ctypes.c_float),
        ctypes.POINTER(ctypes.c_float),
                ctypes.c_int,
        ctypes.POINTER(ctypes.c_float),
        ctypes.c_int]
# Function to perform triangular matrix solve
def triangular_matrix_solve(order, side, uplo, transA, diag, M, N, alpha, A, lda, B, ldb):
    alpha_c = (ctypes.c_float * 2)(alpha.real, alpha.imag)
    A_c = (ctypes.c_float * (2 * lda * M))(*np.array(A, dtype=np.complex64).view(np.float32).flatten())
    B_c = (ctypes.c_float * (2 * ldb * N))(*np.array(B, dtype=np.complex64).view(np.float32).flatten())
    cblas_ctrsm(order, side, uplo, transA, diag, M, N, alpha_c, A_c, lda, B_c, ldb)
    return np.array(B_c).view(np.complex64).reshape((M, N))
```

Figure 23: Case Analysis: the 205-th task in the CrossPL-FFI. The upper part is the task instruction and the lower part is the Canonical solution.

```python
def assert_test():
    # Test case 1
    order = 101  # CblasRowMajor
    side = 141   # CblasLeft
    uplo = 121   # CblasUpper
    transA = 111 # CblasNoTrans
    diag = 131   # CblasNonUnit
    M, N = 2, 2
    alpha = complex(1.0, 0.0)
    A = [[complex(1.0, 0.0), complex(0.0, 0.0)],
         [complex(0.0, 0.0), complex(1.0, 0.0)]]
    B = [[complex(1.0, 0.0), complex(2.0, 0.0)],
         [complex(3.0, 0.0), complex(4.0, 0.0)]]
    lda, ldb = 2, 2
    result = triangular_matrix_solve(order, side, uplo, transA, diag, M, N, alpha, A, lda, B, ldb)
    expected = np.array(B)
    assert np.allclose(result, expected), f"Test case 1 failed: {result} != {expected}"
    # Test case 2
    order = 101
    side = 141
    uplo = 121
    transA = 111
    diag = 131
    M, N = 3, 3
    alpha = complex(2.0, 0.0)
    A = [[complex(1.0, 0.0), complex(0.0, 0.0), complex(0.0, 0.0)],
         [complex(0.0, 0.0), complex(1.0, 0.0), complex(0.0, 0.0)],
         [complex(0.0, 0.0), complex(0.0, 0.0), complex(1.0, 0.0)]]
    B = [[complex(1.0, 0.0), complex(2.0, 0.0), complex(3.0, 0.0)],
         [complex(4.0, 0.0), complex(5.0, 0.0), complex(6.0, 0.0)],
         [complex(7.0, 0.0), complex(8.0, 0.0), complex(9.0, 0.0)]]
    lda, ldb = 3, 3
    result = triangular_matrix_solve(order, side, uplo, transA, diag, M, N, alpha, A, lda, B, ldb)
    expected = np.array(B) * alpha
    assert np.allclose(result, expected), f"Test case 2 failed: {result} != {expected}"
if __name__ == "__main__":
    assert_test()
```

```python
import ctypes
import numpy as np

ctypes.CDLL('/usr/local/gsl/lib/libgslcblas.so', mode=ctypes.RTLD_GLOBAL)
gsl = ctypes.CDLL('/usr/local/gsl/lib/libgsl.so')
def triangular_matrix_solve(order, side, uplo, transA, diag, M, N, alpha, A, B):
    cblas_ctrsm = gsl.cblas_ctrsm
    cblas_ctrsm.argtypes = [ctypes.c_int, ctypes.c_int, ctypes.c_int, ctypes.c_int, ctypes.c_int,
            ctypes.c_int, ctypes.c_int, ctypes.POINTER(ctypes.c_float),
            ctypes.POINTER(ctypes.c_float), ctypes.c_int,
            ctypes.POINTER(ctypes.c_float), ctypes.c_int]
    cblas_ctrsm.restype = None
    A_ctypes = A.ctypes.data_as(ctypes.POINTER(ctypes.c_float))
    B_ctypes = B.ctypes.data_as(ctypes.POINTER(ctypes.c_float))
    alpha_ctypes = (ctypes.c_float * 2)(alpha.real, alpha.imag)
    lda = A.shape[1]
    ldb = B.shape[1]
    cblas_ctrsm(order, side, uplo, transA, diag, M, N, alpha_ctypes, A_ctypes, lda, B_ctypes, ldb)
    return B
```

Figure 24: Case Analysis: the 205-th task in the CrossPL-FFI. The upper part is the test cases and the lower part is the incorrect answer generated by the GPT-4o.

Write a Python function that uses ctypes to load and call a specific function from the GNU Scientific Library (GSL) compiled dynamic library. The function should perform a mathematical operation on a vector of double precision floating -point numbers. The input to the function should be a list of doubles, and the output should be a list of doubles representing the result of the operation. Ensure that libgslcblas.so is loaded globally before libgsl.so to resolve dependencies.To accomplish the above task, the required classes are: ['GSLVector'], and the required functions along with their parameter lists are: [('vector_add', ['a ', 'b'])].

The recommended dynamic library loading order is as follows:
ctypes.CDLL("/usr/local/gsl/lib/libgslcblas.so", mode=ctypes.RTLD_GLOBAL)
gsl = ctypes.CDLL("/usr/local/gsl/lib/libgsl.so")

You must return ONLY a valid JSON object.
Do not include explanations, comments, or extra text.
Do not use triple quotes (""").

The JSON object must contain exactly one field:
"Candidate_solution": <string>

Inside "Candidate_solution", provide the complete code as a single string.
Escape all line breaks as \n.

```python
import ctypes
import numpy as np
ctypes.CDLL('/usr/local/gsl/lib/libgslcblas.so', mode=ctypes.RTLD_GLOBAL)
gsl = ctypes.CDLL('/usr/local/gsl/lib/libgsl.so')
class GSLVector(ctypes.Structure):
    _fields_ = [
        ("size", ctypes.c_size_t),
        ("stride", ctypes.c_size_t),
        ("data", ctypes.POINTER(ctypes.c_double)),
        ("block", ctypes.c_void_p),
        ("owner", ctypes.c_int)
    ]
gsl.gsl_vector_alloc.argtypes = [ctypes.c_size_t]
gsl.gsl_vector_alloc.restype = ctypes.POINTER(GSLVector)
gsl.gsl_vector_free.argtypes = [ctypes.POINTER(GSLVector)]
gsl.gsl_vector_free.restype = None
gsl.gsl_vector_set_all.argtypes = [ctypes.POINTER(GSLVector), ctypes.c_double]
gsl.gsl_vector_set_all.restype = None
gsl.gsl_vector_add.argtypes = [ctypes.POINTER(GSLVector), ctypes.POINTER(GSLVector)]
gsl.gsl_vector_add.restype = ctypes.c_int

def vector_add(a, b):
    n = len(a)
    assert len(b) == n, "Vectors must be of the same length"
    vec_a = gsl.gsl_vector_alloc(n)
    vec_b = gsl.gsl_vector_alloc(n)
    vec_result = gsl.gsl_vector_alloc(n)
    try:
        for i in range(n):
            vec_a.contents.data[i] = a[i]
            vec_b.contents.data[i] = b[i]
        error_code = gsl.gsl_vector_add(vec_a, vec_b)
        if error_code != 0:
            raise RuntimeError("GSL vector addition failed with error code: {}".format(error_code ))
        result = [vec_a.contents.data[i] for i in range(n)]
        return result
    finally:
        gsl.gsl_vector_free(vec_a)
        gsl.gsl_vector_free(vec_b)
        gsl.gsl_vector_free(vec_result)
```

Figure 25: Case Analysis: the 257-th task in the CrossPL-FFI. The upper part is the task instruction and the lower part is the Canonical solution.

```
def assert_test():
    # Test case 1: Simple addition
    a = [1.0, 2.0, 3.0]
    b = [4.0, 5.0, 6.0]
    expected = [5.0, 7.0, 9.0]
    assert vector_add(a, b) == expected, "Test case 1 failed"

    # Test case 2: Zero vector
    a = [0.0, 0.0, 0.0]
    b = [1.0, 2.0, 3.0]
    expected = [1.0, 2.0, 3.0]
    assert vector_add(a, b) == expected, "Test case 2 failed"

    # Test case 3: Negative numbers
    a = [-1.0, -2.0, -3.0]
    b = [1.0, 2.0, 3.0]
    expected = [0.0, 0.0, 0.0]
    assert vector_add(a, b) == expected, "Test case 3 failed"

    print("All test cases passed!")

if __name__ == "__main__":
    assert_test()
```

```
import ctypes
import numpy as np

class GSLVector:
    def __init__(self, data):
        self.data = np.array(data, dtype=np.float64)
        self.size = len(self.data)
        self.c_array = self.data.ctypes.data_as(ctypes.POINTER(ctypes.c_double))
def vector_add(a, b):
    if len(a) != len(b):
        raise ValueError("Vectors must be of the same length ")
    ctypes.CDLL("/usr/local/gsl/lib/libgslcblas.so", mode=ctypes.RTLD_GLOBAL)
    gsl = ctypes.CDLL("/usr/local/gsl/lib/libgsl.so")
    gsl.gsl_vector_add.argtypes = [
        ctypes.POINTER(ctypes.c_double),
        ctypes.POINTER(ctypes.c_double),
        ctypes.POINTER(ctypes.c_double),
        ctypes.c_size_t
    ]
    gsl.gsl_vector_add.restype = None
    vec_a = GSLVector(a)
    vec_b = GSLVector(b)
    result = np.zeros(vec_a.size, dtype=np.float64)
    result_c_array = result.ctypes.data_as(ctypes.POINTER(ctypes.c_double))
    gsl.gsl_vector_add(vec_a.c_array, vec_b.c_array, result_c_array, ctypes.c_size_t(vec_a.size ))
    return result.tolist()
```

Figure 26: Case Analysis: the 257-th task in the CrossPL-FFI. The upper part is the test cases and the lower part is the incorrect answer generated by the GPT-4o.

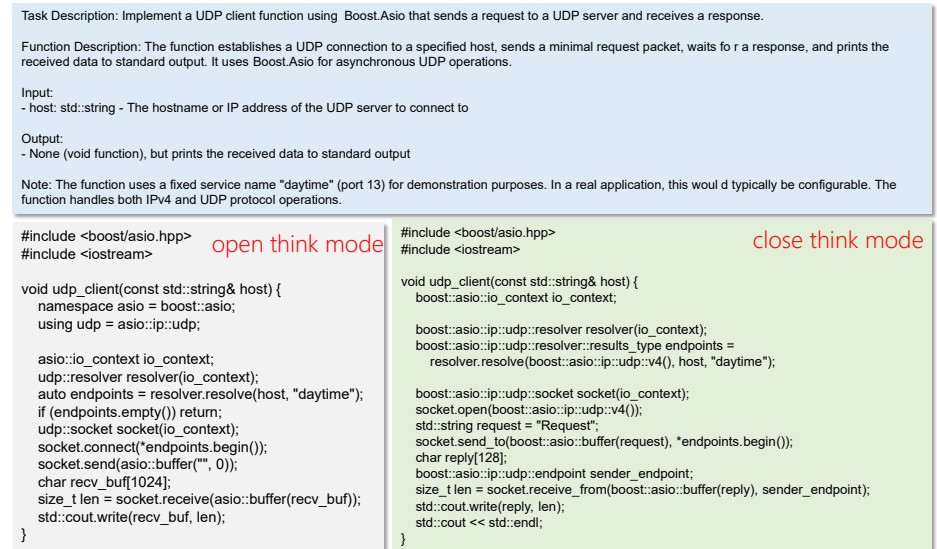

Figure 27: Case Analysis: the 37-th task in the CrossPL-IPC (C++). The left part is the incorrect answer generated by the Qwen3-235b-a22b with think mode. The right part is the correct answer generated by the Qwen3-235b-a22b without think mode.

