# OpenReview forum: "CrossPL: Systematic Evaluation of Large Language Models for Cross Programming Language Interoperating Code Generation"
_ICLR.cc/2026/Conference — ICLR 2026 Poster_

### Official Review · Reviewer_cZK6 · 2025-10-24

**Soundness:** 2
**Presentation:** 3
**Contribution:** 2
**Rating:** 4
**Confidence:** 5

**Summary:**

The goal of this paper is to build a benchmark to evaluate the ability of LLMs
to generate code that spans multiple programming languages. It evaluates both
language interoperability using both IPC and FFI. For IPC, it considers several
techniques and language pairs. For FFI, it only considers FFI between Python
and the GNU Scientific Library (GSL) which is written in C.

The benchmark has 2,000+ tasks, based on existing GitHub repositories. It
achieves this scale with LLM-authored prompts.

**Strengths:**

- An important task to benchmark. To the best of my knowledge, this is not
  a well-studied task.

**Weaknesses:**

I really think the paper is on to something by working on this problem. However,
I am not convinced by the benchmark construction methodology. There are two
main weaknesses:

- Very high likelihood of benchmark contamination: the paper starts with
  existing repositories, which makes contamination likely. I am aware that
  there are other, popular LLM benchmarks with the same flaw, but that doesn't
  mean that they are good benchmarks, even if they are very popular.

  To be concrete, consider that the all the Python-C FFI tasks involve the
  GNU Scientific Library (GSL). Unsurprisingly, there is a well-established
  Python FFI to the GSL on GitHub: https://github.com/pygsl/pygsl

  So, to what extent is the benchmark just measuring the willingness of LLM
  developers to train on GPL code?

- Model generated prompts can be unnatural: the problem with an LLM generated
  prompt is that it can leak very peculiar implementation details unless it is
  carefully crafted. For example, in my opinion, the prompt on Page 34 (Appendix)
  literally specifies all implementation details. In fact, the prompt is
  longer than the canonical solution.

**Questions:**

See the weaknesses listed -- there are two primary concerns that I have with the paper.

---

> ### Author Response · Authors · 2025-11-23
>
> Dear reviewer, we sincerely thank you for the time and effort dedicated to reviewing our paper. We have carefully considered each of your comments and provide detailed responses below.
>
> **Q1**:  Very high likelihood of benchmark contamination: the paper starts with existing repositories, which makes contamination likely. I am aware that there are other, popular LLM benchmarks with the same flaw, but that doesn't mean that they are good benchmarks, even if they are very popular. To be concrete, consider that all the Python-C FFI tasks involve the GNU Scientific Library (GSL). Unsurprisingly, there is a well-established Python FFI to the GSL on GitHub: [https://github.com/pygsl/pygsl](https://github.com/pygsl/pygsl). So, to what extent is the benchmark just measuring the willingness of LLM developers to train on GPL code?
>
> **A1**: Thank you for the insightful question. Benchmark contamination is indeed a crucial issue when evaluating LLMs. To address this, we carefully designed our FFI subset to minimize potential data leakage from existing repositories such as pygsl. Specifically, our FFI samples are constructed via a self-contained, self-instruction approach — the LLM is first shown a C-language function and then asked to generate a question that designs a Python wrapper using `ctypes` to call that function.
>
> To further assess possible contamination, we quantitatively analyzed function name overlaps between our prompts and those in the pygsl repository. The function overlap rate is found to be only 0.61% and the class overlap rate is found to be only 0.99%. These measures collectively greatly reduce the risk of data contamination. We hope it clarifies your concern and will add these discussions in the revised version.
>
> **Table 1: Overlap statistics between pygsl and CrossPL-FFI**
>
> | Metric                                | Count / Rate |
> |---------------------------------------|--------------|
> | Functions in pygsl                    | 4,510        |
> | Functions in FFI-bench                | 14,822       |
> | Functions in both pygsl and FFI-bench | 90           |
> | Function overlap rate                 | 0.61%        |
> | Classes in pygsl                      | 868          |
> | Classes in FFI-bench                  | 304          |
> | Classes in both pygsl and FFI-bench   | 3            |
> | Class overlap rate                    | 0.99%        |
>
>
> **Q2**:   Model generated prompts can be unnatural: the problem with an LLM generated prompt is that it can leak very peculiar implementation details unless it is carefully crafted. For example, in my opinion, the prompt on Page 34 (Appendix) literally specifies all implementation details. In fact, the prompt is longer than the canonical solution.
>
> **A2**: Thank you for the insightful question. We agree that overly specific prompts could, in principle, introduce implementation bias if not handled carefully. However, in our benchmark, the level of prompt detail is intentionally standardized across all models to ensure fairness, reproducibility, and consistent evaluation conditions.
>
> As emphasized in prior works such as SWE-bench [R1], detailed prompts are not a flaw but rather a design choice to minimize ambiguity in task interpretation and guarantee deterministic evaluation. Our goal was not to make the tasks easier, but to ensure that each model receives the same, fully specified instruction context, preventing discrepancies caused by underspecified or ambiguous queries. Therefore, while the prompts may appear longer than the canonical solution, this reflects our commitment to clarity and fairness, not over-specification. The overall evaluation framework remains unbiased and consistent across all models.
>
> **References**
>
> [R1] Jimenez, Carlos E., et al. "SWE-bench: Can Language Models Resolve Real-world Github Issues?." ICLR, 2024.

---

> > ### Comment · Reviewer_cZK6 · 2025-11-27
> >
> > I have read this response and will maintain my score.

---

### Official Review · Reviewer_re9k · 2025-10-27

**Soundness:** 2
**Presentation:** 2
**Contribution:** 2
**Rating:** 4
**Confidence:** 4

**Summary:**

This paper presents CrossPL, a benchmark suite for systematically evaluating large language models (LLMs) for generating cross-programming language interoperating code (CPL interoperating code). The benchmark covers two major interoperability mechanisms: IPC and FFI, encompassing 2,534 tasks (1,982 IPC tasks and 522 Python–C FFI tasks). The authors modeled interfaces based on 156 FSMs, mined 19,169 multi-language GitHub repositories, and designed two LLM-driven build and evaluation workflows (one each for IPC and FFI), ultimately evaluating 20 representative LLMs. Key findings: On the FFI subset, the best model achieved only Pass@1=19.54% and Pass@5=26.46%, in stark contrast to the strong performance of single-language code generation, revealing the significant shortcomings of current LLMs in generating CPL interoperating code.

**Strengths:**

- The problem definition is novel and important: this is the first systematic focus on "cross-language interoperable code generation" rather than translation/monolinguistic operation.
- The experiments are comprehensive, making good use of FSM formalization, large-scale repository mining, LLM pipelines, and FFI executable environments for experiments.
- The experimental results can reveal the difficulties of FFI and the heterogeneity of IPC in the cross-model/language/protocol panoramic comparison.

**Weaknesses:**

- The experiments and methods limit the scope of generalization, making it difficult to prove that the current approach is effective for multiple libraries, platforms, and languages simultaneously.
- IPC correctness is determined only by FSM matching and runtime robustness is not measured, which lacks important evaluation metrics.
- There is no systematic exploration of the impact of hyperparameters and decoding strategies on the overall method. Only the impact of fixed temperature and top-p on the experimental method is given.
- The paper doesn't propose a novel technical path or implementation; its technical paths rely on elements of established, mature implementations. Furthermore, the FFI is limited to Python-C + GSL, and the IPC covers only six languages/seven technology categories, limiting the originality of this "first system benchmark" in broad CPL interoperability.
- The paper's most important conclusion is a negative result (Pass@1 ≈ 19.5% in the FFI subset), but this finding is primarily confirmatory rather than surprising: given the narrow Python-C/GSL setting and the IPC caliber of FSM-compliant simulation, the conclusions are difficult to extrapolate to the broader CPL ecosystem.

**Questions:**

- IPC currently defines "correct" based on FSM matching. Can you provide end-to-end run pass rates and abnormal scenario results?
- Whether statistical significance tests (e.g., paired tests/confidence intervals) were performed for the improvement or decrease in Fig. 3/4
- Why does think mode have a "limited or even negative" effect on IPC but is beneficial for FFI? I think you need to provide a mechanistic analysis of the failure example.

---

> ### Author Response · Authors · 2025-11-23
>
> Dear reviewer, we sincerely thank you for the time and effort dedicated to evaluating our work. We have carefully addressed each of your points in detail below.
>
> **Q1**: IPC currently defines "correct" based on FSM matching. Can you provide end-to-end run pass rates and abnormal scenario results?
>
> **A1**: All IPC tasks in our benchmark are validated through FSM-based static analysis. For IPC, static protocol matching is sufficient, because IPC interactions across languages typically follow deterministic, pattern-driven communication structures (e.g., socket connect–send–recv loops, REST request–response cycles). In contrast, building a unified dynamic execution environment for heterogeneous IPC programs—spanning multiple languages, runtimes, and system configurations is prohibitively complex and offers limited additional benefit for protocol-level evaluation. Our 156 IPC FSMs are constructed from widely used implementations and official documentation, covering 8,487 interaction scenarios (average 54.40 per FSM).

---

> > ### Author Response · Authors · 2025-11-23
> >
> > **Q2**: Whether statistical significance tests (e.g., paired tests/confidence intervals) were performed for the improvement or decrease in Fig.3/4?
> >
> > **A2**: Thank you for the insightful question. We did not perform paired significance tests for Fig. 3/4, as the primary goal of this work is benchmark construction rather than hypothesis testing across models. For greedy decoding, pass@1 is deterministic (temperature = 0), so confidence intervals are not meaningful.
> >
> > For pass@5, each task includes 10 sampled outputs (n = 10, k = 5). To quantify statistical uncertainty, we computed 95% confidence intervals using a non-parametric bootstrap over all 2,504 tasks (5,000 resamples). For each bootstrap sample, we recomputed the macro-average pass@5 and used the 2.5th–97.5th percentiles as the CI bounds. These intervals are now reported for all non-greedy pass@k results in the revised manuscript.
> >
> > In future versions of CrossPL, we plan to incorporate paired tests to further strengthen model-to-model comparisons.
> >
> > Table 1: Pass@5 results of various models on CrossPL-IPC
> >
> > | Model                           | Pass@5   | 95CI%                  |
> > |---------------------------------|---------|-------------------------|
> > | GPT-4o                          | 81.21%  | [79.52%, 82.84%]        |
> > | GPT-4o-mini                     | 69.37%  | [67.38%, 71.41%]        |
> > | Gemini-1.5-pro                  | 76.72%  | [74.87%, 78.52%]        |
> > | GLM4-plus                       | 79.62%  | [77.82%, 81.34%]        |
> > | DeepSeek-V3                     | 78.86%  | [77.10%, 80.59%]        |
> > | Qwen3-235b-a22b                 | 80.42%  | [78.77%, 82.07%]        |
> > | Qwen3-32b                       | 77.73%  | [76.01%, 79.50%]        |
> > | Qwen3-30b-a3b                   | 63.79%  | [61.70%, 65.93%]        |
> > | Qwen3-14b                       | 71.63%  | [69.69%, 73.63%]        |
> > | Qwen3-8b                        | 49.05%  | [46.93%, 51.23%]        |
> > | Qwen3-4b                        | 64.55%  | [62.53%, 66.63%]        |
> > | GLM4-9b                         | 70.93%  | [68.96%, 72.86%]        |
> > | Lamma3-8b-instruct              | 65.21%  | [63.34%, 67.11%]        |
> > | Gemma-7b                        | 68.05%  | [66.12%, 69.92%]        |
> > | CodeGeeX4                       | 74.44%  | [72.62%, 76.26%]        |
> > | Qwen2.5-coder-32b-instruct      | 78.16%  | [76.35%, 79.96%]        |
> > | Qwen2.5-coder-14b-instruct      | 78.14%  | [76.38%, 79.84%]        |
> > | Qwen2.5-coder-7b-instruct       | 77.68%  | [75.89%, 79.37%]        |
> > | CodeLlama-7b-instruct           | 78.60%  | [77.03%, 80.22%]        |
> >
> > Table 2: Pass@5 results of various models on CrossPL-FFI
> >
> > | Model                       | Pass@5  | 95% CI                 |
> > |-----------------------------|---------|-------------------------|
> > | GPT-4o                      | 26.46%  | [22.90%, 30.10%]       |
> > | GPT-4o-mini                 | 13.74%  | [11.04%, 16.59%]       |
> > | Gemini-1.5-pro              | 21.18%  | [17.83%, 24.63%]       |
> > | GLM4-plus                   | 13.40%  | [10.72%, 16.23%]       |
> > | DeepSeek-V3                 | 21.50%  | [18.23%, 25.10%]       |
> > | Qwen3-235b-a22b             | 21.28%  | [18.10%, 24.52%]       |
> > | Qwen3-32b                   | 19.92%  | [16.63%, 23.23%]       |
> > | Qwen3-30b-a3b               | 14.32%  | [11.50%, 17.26%]       |
> > | Qwen3-14b                   | 16.69%  | [13.64%, 19.83%]       |
> > | Qwen3-8b                    | 12.06%  | [9.48%, 14.86%]        |
> > | Qwen3-4b                    | 9.68%   | [7.28%, 12.23%]        |
> > | GLM4-9b                     | 7.92%   | [5.79%, 10.20%]        |
> > | Llama3-8b-instruct          | 3.95%   | [2.59%, 5.48%]         |
> > | Gemma-7b                    | 4.32%   | [2.83%, 5.89%]         |
> > | CodeGeeX4                   | 6.99%   | [4.96%, 9.16%]         |
> > | Qwen2.5-coder-32b-instruct  | 14.38%  | [11.50%, 17.29%]       |
> > | Qwen2.5-coder-14b-instruct  | 7.64%   | [3.04%, 12.89%]        |
> > | Qwen2.5-coder-7b-instruct   | 8.36%   | [6.19%, 10.73%]        |
> > | CodeLlama-7b-instruct       | 5.69%   | [4.09%, 7.52%]         |
> > | CodeGemma-7b                | 8.20%   | [6.10%, 10.49%]        |

---

> > > ### Author Response · Authors · 2025-11-23
> > >
> > > **Q3**: Why does think mode have a “limited or even negative” effect on IPC but is beneficial for FFI? I think you need to provide a mechanistic analysis of the failure example.
> > >
> > > **A3**: Thank you for the insightful question. We have added a detailed discussion section in the revised manuscript to analyze the mechanism behind this phenomenon.
> > >
> > > In summary, think mode encourages models to generate longer reasoning chains before producing code. While this can help in FFI tasks, where reasoning about cross-language type mapping and memory management is necessary, it tends to negatively affect IPC tasks, which require precise and concise protocol handling (e.g., socket setup, message synchronization). In these cases, excessive reasoning often introduces redundant or inconsistent steps that break the communication logic. Appendix D.4  provides a representative failure example of think mode in IPC tasks.
> > >
> > >
> > > **Response to weaknesses:**
> > >
> > > We would like to further clarify the third weakness you mentioned regarding dataset imbalance.
> > >
> > > The primary goal of this work is to establish a systematic fair benchmark framework rather than to optimize decoding strategies or hyperparameters for specific models. As is common practice in benchmark studies (e.g., Multiple-E [R1], Humaneval-X [R2] and Domaineva [R3]), we fixed the temperature and top-p settings across all models to ensure fair and consistent comparison.
> > >
> > > Given the high computational cost of running closed-source models, we additionally conducted a focused supplementary experiment on eight representative open-source models — Qwen3-4b, Qwen3-8b, GLM4-7b, Llama3-8B-Instruct and Gemma-7B (general models), as well as Qwen2.5-coder-7b, CodeLlama-7B-Instruct and CodeGemma-7B (code models). We varied the temperature parameter in {0.0, 0.2, 0.4, 0.6, 0.8, 1.0, 1.2}and top-p parameter in {0.1, 0.3, 0.5, 0.7, 0.9, 1.0} following the HumanEval [R4] setup, and evaluated pass@1 metric.
> > >
> > > The experiment results are recorded in tables. The experiment results show three key observations:
> > >
> > > **1. Both sampling parameters affect model behavior, yet in nonlinear ways.**
> > > Performance varies irregularly across settings, and the optimal configuration differs by model.
> > >
> > > **2. IPC tasks are largely insensitive to sampling variation.**
> > > Their interaction patterns follow fixed communication protocols and deterministic state transitions, yielding a narrow solution space where stochasticity provides little benefit. Consequently, temperature and top-p adjustments lead to only marginal performance shifts.
> > >
> > > **3. FFI tasks remain fundamentally constrained by low-level correctness.**
> > >
> > > FFI requires strict adherence to function signatures, memory layouts, and type coercions; minor deviations result in undefined behavior. Higher sampling diversity increases the likelihood of violating these constraints, while more conservative sampling does not substantively improve reliability.
> > >
> > > This explains the consistently large performance gap—for example, Qwen3-4B averages 55.24% on IPC but only 3.08% on FFI across temperatures, and Qwen2.5-coder-7B shows 68.05% on IPC but only 3.89% across top-p.

---

> > > > ### Author Response · Authors · 2025-11-23
> > > >
> > > > **Table 1: Performance of different models on CrossPL-IPC (Java) under varying temperature (fixed top-p=0.95). Pass@1 computed with n = 5 and k = 1. Values in parentheses denote 95% confidence intervals.**
> > > >
> > > > | Model | T = 0.4 | T = 0.6 | T = 0.8 | T = 1.0 | T = 1.2 |
> > > > |-------|---------|---------|---------|---------|---------|
> > > > | Qwen3-4b | 54.50% ([51.28%, 57.79%]) | 58.41% ([55.19%, 61.63%]) | 53.92% ([50.47%, 57.24%]) | 55.25% ([51.90%, 58.60%]) | 53.82% ([50.44%, 57.14%]) |
> > > > | Qwen3-8b | 10.08% ([8.16%, 12.03%]) | 10.34% ([8.42%, 12.36%]) | 10.96% ([9.01%, 12.91%]) | 11.54% ([9.59%, 13.56%]) | 12.55% ([10.57%, 14.54%]) |
> > > > | GLM4-9b | 51.58% ([48.33%, 54.80%]) | 52.39% ([49.23%, 55.58%]) | 51.80% ([48.68%, 54.96%]) | 52.07% ([48.91%, 55.25%]) | 52.72% ([49.59%, 55.93%]) |
> > > > | Llama3-8B-Instruct | 33.37% ([30.67%, 36.00%]) | 32.94% ([30.34%, 35.61%]) | 32.29% ([29.69%, 34.93%]) | 33.37% ([30.73%, 36.03%]) | 33.40% ([30.73%, 36.10%]) |
> > > > | Gemma-7B | 52.20% ([48.85%, 55.48%]) | 50.31% ([46.99%, 53.66%]) | 49.95% ([46.73%, 53.30%]) | 51.06% ([47.71%, 54.47%]) | 50.44% ([47.06%, 53.79%]) |
> > > > | Qwen2.5-coder-7b | 66.89% ([63.74%, 70.02%]) | 66.11% ([62.93%, 69.27%]) | 66.80% ([63.61%, 69.92%]) | 66.67% ([63.45%, 69.79%]) | 67.77% ([64.52%, 70.96%]) |
> > > > | CodeLlama-7B-Instruct | 55.90% ([52.81%, 58.89%]) | 56.23% ([53.17%, 59.19%]) | 57.40% ([54.37%, 60.39%]) | 55.97% ([52.98%, 59.06%]) | 56.16% ([53.14%, 59.12%]) |
> > > > | CodeGemma-7B | 52.20% ([48.84%, 55.64%]) | 52.91% ([49.50%, 56.29%]) | 53.07% ([49.66%, 56.52%]) | 53.63% ([50.24%, 57.14%]) | 52.52% ([49.17%, 55.87%]) |
> > > >
> > > > ---
> > > >
> > > > **Table 2: Performance of different models on CrossPL-IPC (Java) under varying top-p (fixed temperature=0.2). Pass@1 computed with n = 5 and k = 1. Values in parentheses denote 95% confidence intervals.**
> > > >
> > > > | Model | Top-p = 0.3 | Top-p = 0.5 | Top-p = 0.7 | Top-p = 0.9 | Top-p = 1.0 |
> > > > |-------|------------|------------|------------|------------|------------|
> > > > | Qwen3-4b | 54.31% ([50.44%, 58.05%]) | 54.76% ([50.89%, 58.57%]) | 54.83% ([51.02%, 58.60%]) | 55.06% ([51.35%, 58.86%]) | 54.41% ([50.70%, 58.02%]) |
> > > > | Qwen3-8b | 9.17% ([6.99%, 11.45%]) | 9.11% ([6.93%, 11.28%]) | 9.63% ([7.41%, 11.87%]) | 9.72% ([7.61%, 11.90%]) | 9.92% ([7.77%, 12.10%]) |
> > > > | GLM4-9b | 65.59% ([61.79%, 69.20%]) | 65.37% ([61.53%, 68.98%]) | 65.24% ([61.46%, 68.72%]) | 64.72% ([61.10%, 68.20%]) | 64.07% ([60.62%, 67.48%]) |
> > > > | Llama3-8B-Instruct | 34.63% ([32.07%, 37.27%]) | 31.80% ([29.33%, 34.31%]) | 32.94% ([30.31%, 35.58%]) | 34.05% ([31.32%, 36.72%]) | 33.17% ([30.54%, 35.74%]) |
> > > > | Gemma-7B | 51.61% ([48.33%, 54.86%]) | 51.09% ([47.71%, 54.41%]) | 51.51% ([48.03%, 54.89%]) | 51.54% ([48.23%, 54.93%]) | 52.13% ([48.88%, 55.51%]) |
> > > > | Qwen2.5-coder-7b | 67.87% ([64.10%, 71.54%]) | 67.67% ([63.97%, 71.32%]) | 68.13% ([64.55%, 71.58%]) | 68.03% ([64.62%, 71.38%]) | 68.39% ([65.01%, 71.71%]) |
> > > > | CodeLlama-7B-Instruct | 57.17% ([54.18%, 60.10%]) | 56.23% ([53.20%, 59.22%]) | 55.35% ([52.36%, 58.37%]) | 56.33% ([53.30%, 59.22%]) | 56.29% ([53.20%, 59.32%]) |
> > > > | CodeGemma-7B | 54.86% ([51.54%, 58.24%]) | 53.92% ([50.67%, 57.30%]) | 53.69% ([50.37%, 57.01%]) | 54.37% ([51.09%, 57.63%]) | 54.80% ([51.51%, 58.24%]) |
> > > >
> > > > ---
> > > >
> > > > **Table 3: Performance of different models on CrossPL-FFI under varying temperature (fixed top-p=0.95). Pass@1 computed with n = 5 and k = 1. Values in parentheses denote 95% confidence intervals.**
> > > >
> > > > | Model | T = 0.4 | T = 0.6 | T = 0.8 | T = 1.0 | T = 1.2 |
> > > > |-------|---------|---------|---------|---------|---------|
> > > > | Qwen3-4b | 2.72% ([1.88%, 3.72%]) | 3.72% ([2.57%, 4.98%]) | 2.80% ([1.92%, 3.75%]) | 3.22% ([2.22%, 4.37%]) | 2.84% ([1.88%, 3.95%]) |
> > > > | Qwen3-8b | 6.93% ([5.17%, 8.81%]) | 6.21% ([4.48%, 8.05%]) | 6.67% ([4.90%, 8.62%]) | 6.48% ([4.22%, 9.57%]) | 6.59% ([4.87%, 8.47%]) |
> > > > | GLM4-9b | 1.00% ([0.50%, 1.57%]) | 1.30% ([0.65%, 2.07%]) | 0.73% ([0.34%, 1.23%]) | 1.03% ([0.54%, 1.61%]) | 0.80% ([0.38%, 1.30%]) |
> > > > | Llama3-8B-Instruct | 1.11% ([0.61%, 1.69%]) | 1.15% ([0.65%, 1.69%]) | 1.11% ([0.61%, 1.72%]) | 0.96% ([0.54%, 1.46%]) | 1.07% ([0.61%, 1.57%]) |
> > > > | Gemma-7B | 1.38% ([0.73%, 2.15%]) | 1.88% ([1.07%, 2.80%]) | 1.57% ([0.88%, 2.41%]) | 1.69% ([0.96%, 2.49%]) | 1.84% ([1.03%, 2.72%]) |
> > > > | Qwen2.5-coder-7b | 4.25% ([3.10%, 5.52%]) | 4.06% ([2.91%, 5.33%]) | 4.18% ([3.03%, 5.44%]) | 3.79% ([2.68%, 5.02%]) | 3.75% ([2.72%, 4.87%]) |
> > > > | CodeLlama-7B-Instruct | 1.72% ([1.03%, 2.49%]) | 1.49% ([0.92%, 2.15%]) | 1.95% ([1.26%, 2.72%]) | 1.76% ([1.07%, 2.57%]) | 1.69% ([1.00%, 2.45%]) |
> > > > | CodeGemma-7B | 3.98% ([2.80%, 5.29%]) | 4.25% ([2.99%, 5.67%]) | 3.33% ([2.26%, 4.52%]) | 3.49% ([2.34%, 4.71%]) | 3.79% ([2.61%, 5.13%]) |

---

> > > > > ### Author Response · Authors · 2025-11-23
> > > > >
> > > > > **Table 4: Performance of different models on CrossPL-FFI under varying top-p (fixed temperature=0.2). Pass@1 computed with n = 5 and k = 1. Values in parentheses denote 95% confidence intervals.**
> > > > >
> > > > > | Model | Top-p = 0.3 | Top-p = 0.5 | Top-p = 0.7 | Top-p = 0.9 | Top-p = 1.0 |
> > > > > |-------|------------|------------|------------|------------|------------|
> > > > > | Qwen3-4b | 6.63% ([4.71%, 8.77%]) | 6.25% ([4.37%, 8.31%]) | 6.44% ([4.56%, 8.47%]) | 6.25% ([4.41%, 8.20%]) | 6.90% ([4.98%, 8.97%]) |
> > > > > | Qwen3-8b | 9.66% ([7.20%, 12.34%]) | 10.00% ([7.82%, 12.99%]) | 9.66% ([7.28%, 12.26%]) | 9.50% ([7.20%, 12.03%]) | 9.77% ([7.43%, 12.30%]) |
> > > > > | GLM4-9b | 1.23% ([0.69%, 1.84%]) | 0.80% ([0.38%, 1.30%]) | 0.96% ([0.46%, 1.53%]) | 1.07% ([0.57%, 1.65%]) | 1.42% ([0.73%, 2.26%]) |
> > > > > | Llama3-8B-Instruct | 1.46% ([0.96%, 1.99%]) | 1.15% ([0.65%, 1.76%]) | 0.77% ([0.38%, 1.23%]) | 0.57% ([0.27%, 0.96%]) | 0.65% ([0.27%, 1.11%]) |
> > > > > | Gemma-7B | 1.34% ([0.69%, 2.07%]) | 2.15% ([1.23%, 3.18%]) | 1.92% ([1.11%, 2.84%]) | 1.49% ([0.80%, 2.30%]) | 1.80% ([1.03%, 2.68%]) |
> > > > > | Qwen2.5-coder-7b | 3.79% ([2.68%, 5.06%]) | 4.79% ([3.49%, 6.21%]) | 3.41% ([2.45%, 4.48%]) | 3.68% ([2.64%, 4.87%]) | 3.79% ([2.76%, 4.94%]) |
> > > > > | CodeLlama-7B-Instruct | 1.26% ([0.73%, 1.88%]) | 1.92% ([1.19%, 2.76%]) | 1.38% ([0.80%, 2.03%]) | 1.57% ([0.96%, 2.26%]) | 1.84% ([1.11%, 2.64%]) |
> > > > > | CodeGemma-7B | 3.75% ([2.61%, 5.06%]) | 3.83% ([2.68%, 5.10%]) | 3.64% ([2.49%, 4.94%]) | 3.98% ([2.80%, 5.29%]) | 2.95% ([1.95%, 4.06%]) |
> > > > >
> > > > > Detailed discussion and analysis of these experiments can be found in the Discussion and Appendix C.6 of the revised manuscript.
> > > > >
> > > > > **Reference:**
> > > > >
> > > > > [R1] Cassano, Federico, et al. "Multipl-E: A scalable and polyglot approach to benchmarking neural code generation." IEEE Transactions on Software Engineering 49.7 (2023): 3675-3691.
> > > > >
> > > > > [R2] Zheng, Qinkai, et al. "Codegeex: A pre-trained model for code generation with multilingual benchmarking on humaneval-x." Proceedings of the 29th ACM SIGKDD Conference on Knowledge Discovery and Data Mining. 2023.
> > > > >
> > > > > [R3] Zhu, Qiming, et al. "Domaineval: An auto-constructed benchmark for multi-domain code generation." Proceedings of the AAAI Conference on Artificial Intelligence. Vol. 39. No. 24. 2025.
> > > > >
> > > > > [R4] Chen, Mark. "Evaluating large language models trained on code." arXiv preprint arXiv:2107.03374 (2021).

---

> > > > > > ### Comment · Reviewer_re9k · 2025-11-27
> > > > > >
> > > > > > Thanks for the response, I will keep my score.

---

### Official Review · Reviewer_aE3T · 2025-11-02

**Soundness:** 3
**Presentation:** 3
**Contribution:** 3
**Rating:** 6
**Confidence:** 2

**Summary:**

CrossPL introduces a benchmark for cross-programming-language (CPL) interoperating code generation, covering two modes: IPC (1,982 tasks across 6 languages and 7 techniques) and Python↔C FFI (522 tasks). The benchmark is largely auto-constructed with 156 FSMs plus two LLM-driven workflows (for mining/templating IPC snippets and for building/validating FFI tasks from GSL). Evaluation of 20 LLMs shows stark gaps versus single-language coding: best Pass@1 is ~80%+ on some IPC slices but only 19.5% Pass@1 on FFI (GPT-4o), highlighting interoperability as an unsolved frontier. The paper also provides error taxonomies and an analysis of “think” mode (helpful for FFI, mixed for IPC). Overall, this is a timely benchmark with solid engineering, though several choices (Python-C-only FFI, FSM-only IPC validation, LLM-generated instructions) raise concerns about coverage, validity, and potential bias/contamination.

**Strengths:**

Clear problem framing: Shifts evaluation from translation/single-language coding to true cross-language interaction.

Scale & breadth: 2,534 tasks; IPC spans 6 languages × 7 techniques; rare in this area.

Methodological novelty: Use of 156 FSMs to mine/validate IPC protocols is systematic and reusable.

**Weaknesses:**

FFI narrowness: FFI limited to Python↔C (GSL); may not generalize to JNI, CFFI, Rust FFI, JS↔C++ addons, Swift bridging, etc.

IPC validation fidelity: FSM conformance ≠ functional correctness; many IPC tasks are not executed end-to-end (risk of false positives).

Dataset skew: Language/technique distribution is uneven (e.g., heavy Java/HTTP), which can bias aggregate metrics.

Contamination risk: Mining popular repos and widely documented patterns without a contamination analysis could inflate results.

**Questions:**

PC validation fidelity: What fraction of IPC tasks are executed end-to-end vs only FSM-matched, and what are the FSM false-accept/false-reject rates?

Contamination/time split: Did you create a time-based holdout (repos/docs post-training) to estimate training-data leakage and its impact on scores?

FFI generalization: Why restrict to Python↔C (GSL)? Any pilots for JNI/Rust/Node add-ons/CFFI—and a plan to broaden FFI coverage in v2?

---

> ### Author Response · Authors · 2025-11-23
>
> Dear reviewer, we sincerely appreciate your thoughtful review and valuable feedback. Our point-by-point responses to your questions are provided below.
>
> **Q1**: IPC validation fidelity: What fraction of IPC tasks are executed end-to-end vs only FSM-matched, and what are the FSM false-accept/false-reject rates?
>
> **A1**: All IPC tasks in our benchmark are validated through FSM-based static analysis. For IPC, static protocol matching is sufficient, because IPC interactions across languages typically follow deterministic, pattern-driven communication structures (e.g., socket connect–send–recv loops, REST request–response cycles). In contrast, building a unified dynamic execution environment for heterogeneous IPC programs—spanning multiple languages, runtimes, and system configurations is prohibitively complex and offers limited additional benefit for protocol-level evaluation.
>
> Our 156 IPC FSMs are constructed from widely used implementations and official documentation, covering 8,487 interaction scenarios (average 54.40 per FSM). Manual inspection of 100 sampled tasks shows a 5% false-accept rate and a 9% false-reject rate, indicating that FSM-based validation provides reliable protocol checking at specific tasks.
>
> In the future work, we will develop a unified end to end execution and verification framework to evaluate LLM-generated cross-language code beyond static FSM validation.

---

> > ### Author Response · Authors · 2025-11-23
> >
> > **Q2**: Contamination/time split: Did you create a time-based holdout (repos/docs post-training) to estimate training-data leakage and its impact on scores?
> >
> > **A2**: Thank you for the insightful question. To address potential concerns regarding temporal contamination, we clarify how time-based considerations were incorporated in the construction of both subsets of CrossPL. Here is our response to your question; the full explanation can be found in the revised manuscript’s Discussion section and Appendix C.5.
> >
> > **1. Temporal analysis in the construction of CrossPL-IPC**
> >
> > The IPC subset is built from real-world open-source repositories on GitHub, and therefore requires explicit temporal validation. We categorized all IPC samples into four time buckets: before 2022, 2022, 2023, and 2024–2025. Notably, the majority of IPC instances (1,377 samples, 69.48% of the IPC subset) originate from 2024–2025.
> >
> > **Table 1: Number of CrossPL-IPC Samples with Different Release Years**
> >
> > | Before 2022 | 2022 | 2023 | 2024–2025 |
> > |-------------|------|------|-----------|
> > | 222         | 143  | 240  | 1377      |
> >
> > We then evaluated model performance across these temporal slices. The results (Table 2 as below and Figure 20 in the revised manuscript) show no evidence that models perform systematically better on the newer subsets; in fact, model scores on older samples do not exceed those on more recent ones.
> >
> >
> > **Table 2: Pass@1 Performance Across Release Years**
> >
> > | Model                         | Before 2022 | 2022   | 2023   | 2024–2025 |
> > |-------------------------------|------------|--------|--------|-----------|
> > | GPT-4o                         | 70.27%    | 81.12% | 75.42% | 73.38%    |
> > | GPT-4o-mini                     | 64.41%    | 80.42% | 64.17% | 63.11%    |
> > | Gemini-1.5-pro                  | 65.77%    | 80.42% | 69.58% | 69.64%    |
> > | GLM4-plus                       | 71.62%    | 83.22% | 75.42% | 75.89%    |
> > | DeepSeek-V3                     | 72.97%    | 81.82% | 72.08% | 72.98%    |
> > | Qwen3-235b-a22b                 | 69.55%    | 81.43% | 76.69% | 70.79%    |
> > | Qwen3-32b                       | 63.64%    | 77.86% | 64.83% | 64.34%    |
> > | Qwen3-30b-a3b                   | 59.09%    | 70.00% | 58.05% | 55.58%    |
> > | Qwen3-14b                       | 64.09%    | 77.14% | 66.53% | 64.19%    |
> > | Qwen3-8b                        | 50.45%    | 32.14% | 38.98% | 43.22%    |
> > | Qwen3-4b                        | 55.00%    | 68.57% | 62.29% | 55.43%    |
> > | GLM4-9b                         | 59.91%    | 78.32% | 61.25% | 59.48%    |
> > | Lamma3-8b-instruct              | 43.05%    | 39.44% | 49.58% | 38.85%    |
> > | Gemma-7b                        | 51.12%    | 59.86% | 58.75% | 47.86%    |
> > | CodeGeeX4                        | 62.61%    | 81.12% | 70.42% | 63.04%    |
> > | Qwen2.5-coder-32b-instruct      | 65.77%    | 83.92% | 72.03% | 72.48%    |
> > | Qwen2.5-coder-14b-instruct      | 67.12%    | 78.32% | 71.67% | 70.81%    |
> > | Qwen2.5-coder-7b-instruct       | 66.22%    | 79.02% | 69.58% | 68.19%    |
> > | CodeLlama-7b-instruct           | 52.47%    | 66.90% | 56.67% | 46.91%    |
> > | CodeGemma-7b                     | 55.16%    | 69.72% | 59.17% | 53.30%    |
> > | **Mean**                        | 60.63%    | 73.08% | 64.98% | 60.88%    |
> >
> > **2. Contamination mitigation in the construction of CrossPL-FFI**
> >
> > For the FFI subset, we deliberately designed the pipeline to avoid exposure to pre-existing code. All FFI samples are constructed in a self-contained and self-guided manner: the LLM is shown a standalone C function and is asked to (i) generate a task description, (ii) design a Python wrapper using ctypes, and (iii) produce a canonical reference solution.
> >
> > To further quantify the risk of training-data contamination, we measured the overlap between our prompts and the Pygsl codebase used during verification. The overlap is extremely low—0.61% for function names and 0.99% for class names—indicating minimal risk that LLMs benefited from prior exposure to these specific instances.
> >
> > **Table 3: Overlap statistics between pygsl and CrossPL-FFI**
> >
> > | Metric                                | Count / Rate |
> > |---------------------------------------|--------------|
> > | Functions in pygsl                    | 4,510        |
> > | Functions in FFI-bench                | 14,822       |
> > | Functions in both pygsl and FFI-bench | 90           |
> > | Function overlap rate                 | 0.61%        |
> > | Classes in pygsl                      | 868          |
> > | Classes in FFI-bench                  | 304          |
> > | Classes in both pygsl and FFI-bench   | 3            |
> > | Class overlap rate                    | 0.99%        |
> >
> > We thank the reviewer for raising this question, as it prompted us to conduct a more systematic examination of temporal leakage and contamination across both IPC and FFI construction pipelines.

---

> > > ### Author Response · Authors · 2025-11-23
> > >
> > > **Q3**: FFI generalization: Why restrict to Python↔C (GSL)? Any pilots for JNI/Rust/Node add-ons/CFFI—and a plan to broaden FFI coverage in v2?
> > >
> > > **A3**: Thank you for the insightful question. Designing a unified framework for validation, installation, execution, and testing across diverse programming language combinations remains a complex challenge and most existing studies on cross-language interaction [R1-R4] also focus on only one specific pair such as Python–C or Java–C.
> > >
> > > We restricted FFI evaluation to Python–C because both are among the ten most widely used programming languages: C provides strong performance and fine-grained memory control, while Python offers simplicity and a rich ecosystem essential for AI and scientific computing (e.g., NumPy, SciPy, PyTorch, TensorFlow).
> > >
> > > For v2 of the benchmark, we plan to:
> > > 1. Develop a unified execution and verification framework to evaluate LLM-generated cross-language code beyond static FSM validation.
> > > 2. Extend FFI coverage to additional combinations, such as Java–C (JNI), Rust–Python (PyO3), and Node.js–C++ (N-API).
> > >
> > > **Response to weaknesses:**
> > >
> > > We would like to further clarify the third weakness you mentioned regarding dataset imbalance.
> > >
> > > Our benchmark is the first to focus specifically on evaluating LLMs’ ability to generate cross-programming-language interaction code. To ensure the quality and correctness of collected samples, we prioritized tasks with stable implementations and comprehensive FSM coverage. As a result, some language or technique categories (e.g., Java/HTTP) are slightly overrepresented. However, this work focuses on the 0 to 1 gap of a benchmark to evaluate cross-programming-language interaction code generation for LLMs. In future work, we plan to progressively expand the dataset to include underrepresented language pairs and IPC/FFI techniques to improve balance and coverage.
> > >
> > > **References**
> > >
> > > [R1] Yang, Haoran, et al. "Learning to detect and localize multilingual bugs." Proceedings of the ACM on Software Engineering. FSE (2024): 2190–2213.
> > >
> > > [R2] Li, Wen, et al. "PolyCruise: A Cross-Language dynamic information flow analysis." 31st USENIX Security Symposium (USENIX Security 22). 2022.
> > >
> > > [R3] Li, Wen, et al. "PolyFuzz: Holistic Greybox Fuzzing of Multi-Language Systems." 32nd USENIX Security Symposium (USENIX Security 23). 2023.
> > >
> > > [R4] Bae, Sora, Sungho Lee, and Sukyoung Ryu. "Towards understanding and reasoning about Android interoperations." 2019 IEEE/ACM 41st International Conference on Software Engineering (ICSE). IEEE, 2019.

---

### Official Review · Reviewer_LuTN · 2025-11-02

**Soundness:** 2
**Presentation:** 3
**Contribution:** 2
**Rating:** 6
**Confidence:** 2

**Summary:**

This paper introduces CrossPL, the first benchmark for systematically assessing LLM performance in Cross Programming Language (CPL) interoperating code generation.

**Strengths:**

- The writing is fluent and logically clear.
﻿
- The dataset and evaluation code are open-sourced, facilitating replication of the work and future use by the community.
﻿
- The first benchmark for CPL interoperating code generation
﻿
- It highlights the widespread inadequacy of large models in this task, providing a new perspective for improving LLM capabilities.
﻿
- The benchmark is large-scale and provides a methodology for automated benchmark construction.
﻿
- The use of FSM-based validation simplifies the evaluation process.

**Weaknesses:**

- CrossPL-FFI is limited to Python–C tasks, and even focuses on the GSL library, raising concerns that the conclusions might be overly assertive or not generalizable.

- There remains a gap between FSM-based validation and actual execution, which may affect the accuracy of performance assessment.

Tips

- The abbreviations IPC and FFI are not fully defined upon their first appearance in the abstract, which could hinder readability.

**Questions:**

- Was there any manual review process for the 156 FSMs? Could insufficient coverage of the FSMs lead to an underestimation of model performance?

- For IPC tasks, the FSM validation focuses on state transitions, but does it overlook performance aspects (such as latency or concurrency) in the evaluation?

---

> ### Author Response · Authors · 2025-11-23
>
> Dear reviewer, thank you very much for your time and constructive feedback. We have carefully addressed your comments and provide our detailed responses below.
>
> **Q1**: Was there any manual review process for the 156 FSMs? Could insufficient coverage of the FSMs lead to an underestimation of model performance?
>
> **A1**: Yes. All 156 FSMs were manually designed and reviewed by the authors using official documentation and widely adopted open-source IPC implementations. Each FSM was checked for correctness and to ensure that it captures the dominant usage patterns of its corresponding mechanism.
>
> While no finite FSM set can exhaustively cover all IPC cases, our design targets the most common and practically relevant patterns. The dataset construction pipeline is itself driven by these FSMs, ensuring that all IPC samples in the benchmark are aligned with their coverage, which keeps evaluation fair across models.
>
> To quantify coverage, the 156 FSMs collectively cover 8,487 distinct interaction settings (average 54.40 per FSM), providing broad and representative coverage and yielding the largest collection of such interaction patterns so far. We will continue active expansion of the FSM library as new IPC patterns emerge in the future.
>
>
> **Q2**: For IPC tasks, the FSM validation focuses on state transitions, but does it overlook performance aspects (such as latency or concurrency) in the evaluation?
>
> A2: Thank you for the insightful question. This is true but a shared limitation by prior cross-language analysis frameworks [R1-R3], which likewise rely on static FSM validation due to the complexity of building a unified runtime system for heterogeneous IPC implementations.
>
> Despite the FSM being essentially a static analysis tool, the framework still highlights meaningful observations. We leave such performance aspects evaluation to future work.
>
>
> **Q3**: The abbreviations IPC and FFI are not fully defined upon their first appearance in the abstract, which could hinder readability.
>
> **A3**: Thank you for the helpful note. We have carefully fixed these.
>
> **References**
> [R1] Li, Wen, Li Li, and Haipeng Cai. "PolyFax: A toolkit for characterizing multi-language software." Proceedings of the 30th ACM Joint European Software Engineering Conference and Symposium on the Foundations of Software Engineering. 2022.
>
> [R2] Yang, Haoran, et al. "Learning to detect and localize multilingual bugs." Proceedings of the ACM on Software Engineering 1.FSE (2024): 2190-2213.
>
> [R3] Yang, Haoran, and Haipeng Cai. "Dissecting Real-World Cross-Language Bugs." Proceedings of the ACM on Software Engineering 2.FSE (2025): 1270-1292.

---

> > ### Comment · Reviewer_LuTN · 2025-11-26
> >
> > Thank you for your response and clarification. However, due to the weaknesses of the paper itself, I maintain my original score.

---

### Official Review · Reviewer_DQS3 · 2025-11-02

**Soundness:** 2
**Presentation:** 2
**Contribution:** 3
**Rating:** 6
**Confidence:** 4

**Summary:**

This work presents a cross-programming-language benchmark designed to systematically evaluate large language models (LLMs) on cross-language code generation tasks. The benchmark includes 2,534,982 interprocess communication tasks across six programming languages, as well as 522 Python–C foreign function interface (FFI) tasks. The study is motivated by the growing prevalence of real-world systems that integrate two or more programming languages. Through extensive experimentation, the authors provide insightful analyses of LLM performance, highlighting differences in capabilities across languages and programming techniques.

**Strengths:**

* The benchmark is large in scale, with the IPC task subset covering multiple real-world repositories and the FFI subset derived from a high-quality open-source repository.
* The benchmark construction pipeline is fully automated, leveraging established software engineering methods to ensure correctness.
* Although the benchmark is synthetically generated, the authors validate correctness through stochastic verification, including compilation and execution tests and a comprehensive suite of test cases.
* The evaluation spans a diverse set of LLMs, encompassing both proprietary and large open-source models, providing a balanced comparative analysis.

**Weaknesses:**

* The manuscript employs an excessive number of acronyms within individual paragraphs across multiple sections, which negatively affects readability. Including a dedicated background or terminology section early in the paper would help introduce key terms and abbreviations, improving overall clarity.
* The literature review omits several major code benchmarks, including SWE-bench, Multi-SWE-bench, and MBPP, which are essential for contextualizing the contribution.
* Although a manual review was conducted, the methodology and outcomes are not reported, limiting transparency and reproducibility.
* The pipeline’s performance appears to rely heavily on large, high-capacity LLMs, which may limit its reproducibility and scalability when using smaller open-source models.
* Despite identifying outliers, most LLMs achieve similarly high scores on both IPC and FFI tasks, suggesting the need for further task annotation and filtering to better capture differences in task complexity.

**Questions:**

* Benchmark design: Multi-programming-language tasks can often be represented within repository-level benchmarks. Why do you believe your benchmark design offers a superior approach compared to, for instance, extending SWE-bench to include GitHub repositories that natively incorporate multiple programming languages?
* FSM creation: Based on the main paper, I understand that the finite state machines (FSMs) were not generated by LLMs. Could you please confirm this?
* Manual review details: What was the size of the manually sampled subsets used for human review (referenced in Lines 254 and 287)? Could you describe the review process in more detail and specify how many reviewers participated?
* Filtering and model accuracy: Could you provide detailed percentages of instances that were filtered or pruned at each LLM-judging step? Additionally, do you believe that smaller or less powerful LLMs could achieve comparable accuracy on these tasks?

---

> ### Author Response · Authors · 2025-11-23
>
> Dear reviewer, we appreciate your time for review our paper and great feedback! We have incorporated the rebuttal content into our revision to answer your questions and concerns. We revised our manuscript as follows:
> (1) Clarified terminology and emphasized contributions (Related work)
> (2) We added comprehensive statistics for each key filtering stage in the pipeline (Appendix B.2) and analyzed the impact of model size on the overall pipeline behavior (Discussion and Appendix C.4).
>
> **Q1**: Multi-programming-language tasks can often be represented within repository-level benchmarks. Why do you believe your benchmark design offers a superior approach compared to, for instance, extending SWE-bench to include GitHub repositories that natively incorporate multiple programming languages?
>
> **A1**: Thank you for your time and thoughtful comments. In real-world software systems, multi-programming-language projects constitute a substantial portion of codebases, and CPL interaction plays a critical role in such systems (e.g. Numpy, SciPy, Pytorch, and Tensorflow). Our work specifically focuses on evaluating LLMs’ ability to generate correct cross-language interaction code, which is a distinct and practically significant challenge.
>
> The original SWE-bench[R1] was designed as a Python-centric code repair benchmark, and more recent work Multi-SWE-bench [R2] extends this scope to seven programming languages. Although Multi-SWE-bench includes some cross-language repair tasks, our manual inspection revealed that out of 1,682 repair problems, only 28 involve cross-language interaction, which is the focus of our work. Thus, we believe our work will be an important complementary to the existing Multi-SWE-bench.
>
>
> **Q2**: Based on the main paper, I understand that the finite state machines (FSMs) were not generated by LLMs. Could you please confirm this?
>
>
> **A2**: Yes. All FSMs in this study were manually constructed and verified through meticulous inspection to ensure accuracy and reliability.
>
> **Q3**: What was the size of the manually sampled subsets used for human review (referenced in Lines 254 and 287)? Could you describe the review process in more detail and specify how many reviewers participated?
>
> **A3**: Thank you for the comment. Our study involves human review of two kinds:
>
> (1) IPC-subset construction: manual verification of whether the instructor-generated task instructions were accurate and appropriate.
> (2) FFI-subset construction: manual review of the generated instructions, assertion test cases, and canonical solutions.
>
> We randomly sampled 200 IPC examples and 50 FFI examples. The review was conducted by two authors, who discussed each case together. Only samples that received unanimous approval were retained.
>
> All 200 IPC samples were retained, indicating that LLMs performed well in understanding code snippets and generating clear, detailed, and accurate task instructions — as illustrated in Figure 20 (Appendix). We attribute this to the effectiveness of our instruction-generation prompt, which was carefully designed based on [R3–R5]. In contrast to [R3–R5], where task instructions were directly generated from canonical solutions without human verification, we incorporated manual review to ensure higher quality.
>
> For the 50 FFI samples, only two samples were removed. This high success rate is largely because the LLM generated Python tasks (via ctypes) after directly observing the underlying C function. Moreover, prior to manual review, these samples had already passed execution-based validation, which filtered out unreasonable cases.

---

> > ### Author Response · Authors · 2025-11-23
> >
> > **Q4**: Filtering and model accuracy: Could you provide detailed percentages of instances that were filtered or pruned at each LLM-judging step?
> >
> > **A4**: Thank you for this insightful comment. The related statistics are as follows, and we will include them in the revised version of the manuscript:
> >
> > Table 1: Detailed number statistics of instances filtered at each key step in CrossPL-IPC construction pipeline
> >
> > | Language   | Total MPL Projects | File Counts (FSM Filtered) | File Counts (Token Limit) | Instance Counts (Judger Filtered) | Instance Counts (FSM Filtered) |
> > |------------|--------------------|-----------------------------|-----------------------------|------------------------------------|----------------------------------|
> > | C++        | 1462               | 70                          | 70                          | 58                                 | 51                               |
> > | Go         | 1534               | 2239                        | 989                         | 497                                | 392                              |
> > | Java       | 8807               | 1038                        | 966                         | 669                                | 615                              |
> > | JavaScript | 7560               | 3271                        | 777                         | 368                                | 271                              |
> > | PHP        | 704                | 921                         | 292                         | 252                                | 151                              |
> > | Python     | 5219               | 1690                        | 913                         | 522                                | 502                              |
> >
> >
> > Table 2: Detailed number statistics of instances filtered at each key step in CrossPL-FFI construction pipeline
> >
> > | Language | Total C Functions | File Counts (Execution Filtered) |
> > |----------|--------------------|----------------------------------|
> > | C++      | 1061               | 522                              |

---

> > > ### Author Response · Authors · 2025-11-23
> > >
> > > **Q5**: Additionally, do you believe that smaller or less powerful LLMs could achieve comparable accuracy on these tasks?
> > > **A5**: Thank you for the insightful comment. To answer the question, we conducted ablations replacing DeepSeek-V3 with Qwen3-4B in three components: (1) Extractor in CrossPL-IPC construction pipeline, (2) Instructor in CrossPL-IPC construction pipeline, and (3) Generator in CrossPL-FFI construction pipeline. Results can conclude as follows:
> > >
> > > 1. Extractor replacement in CrossPL-IPC construction pipeline: valid samples dropped from 1,982 to 1,723.
> > >
> > > 2. Instructor replacement in CrossPL-IPC construction pipeline: a small drop 1,982 to 1,890.
> > >
> > > 3. Generator replacement in CrossPL-FFI construction pipeline:  valid samples severely degrade from 522 to 65.
> > >
> > > The observation is that smaller models can therefore replace only lightweight components such as the Instructor. Components requiring structural code understanding or cross-language synthesis—Extractor and especially the FFI Generator—depend critically on powerful models. Although the Judger produces valid categorical outputs, smaller models increase false positives and cause substantial downstream overhead, so a powerful model is still preferred. We will add these results to the revised version.
> > >
> > > Table 3: Instance counts before and after “Extractor” / “Instructor” replacement
> > >
> > > | Language   | Original | “Extractor” replacement (Qwen3-4B) | “Instructor” replacement (Qwen3-4B) |
> > > |-----------|----------|-------------------------------------|--------------------------------------|
> > > | C++        | 51       | 52                                  | 52                                   |
> > > | Go         | 392      | 353                                 | 326                                  |
> > > | Java       | 615      | 606                                 | 620                                  |
> > > | JavaScript | 271      | 157                                 | 262                                  |
> > > | PHP        | 151      | 159                                 | 162                                  |
> > > | Python     | 502      | 396                                 | 468                                  |
> > >
> > >
> > > Table 4: Instance counts before and after **“Generator”** replacement
> > >
> > > | Language Pair | Original | “Generator” replacement (Qwen3-4B) |
> > > |---------------|----------|-------------------------------------|
> > > | Python–C      | 522      | 65                                  |
> > >
> > >
> > > **Response to weaknesses**:
> > >
> > > We have also revised the manuscript according to the first two weaknesses you mentioned:
> > > 1. We thoroughly reviewed all abbreviations and provided their full forms upon first appearance.
> > > 2. We expanded the Related Work section to include detailed discussions of SWE-bench, Multi-SWE-bench, and MBPP, highlighting how our work differs from theirs and further emphasizing our contributions.
> > >
> > > These revisions have been marked in blue in the updated manuscript. Finally, we sincerely thank you for your time and thoughtful feedback, which have helped us improve the quality and clarity of our paper.
> > >
> > > **References**
> > > [R1] Jimenez, Carlos E., et al. "SWE-bench: Can Language Models Resolve Real-world Github Issues?." ICLR, 2024.
> > > [R2] Zan, Daoguang, et al. "Multi-swe-bench: A multilingual benchmark for issue resolving." arXiv preprint arXiv:2504.02605 (2025).
> > > [R3] Zhu, Qiming, et al. "Domaineval: An auto-constructed benchmark for multi-domain code generation." Proceedings of the AAAI Conference on Artificial Intelligence, Vol. 39, No. 24, 2025.
> > > [R4] Li, Jia, et al. "EvoCodeBench: an evolving code generation benchmark with domain-specific evaluations." Proceedings of the 38th International Conference on Neural Information Processing Systems, 2024.
> > > [R5] Xie, Yiqing, et al. "Codebenchgen: Creating scalable execution-based code generation benchmarks." arXiv preprint arXiv:2404.00566, 2024.

---

### Author Response · Authors · 2025-12-02

**General response:**

We thank the reviewers for their time and constructive feedback. All rebuttal clarifications have been integrated into the revised manuscript.

This work is motivated by a key observation: although LLMs perform well on single-language code benchmarks, real-world software is inherently multilingual as over 80% of modern projects use multiple programming languages. **Existing multilingual benchmarks emphasize translation rather than evaluating whether LLMs can generate cross-programming-language (CPL) interoperating code. This leaves a critical gap in assessing LLMs’ practical CPL capabilities.**

To address this gap, we introduce CrossPL, **the first** benchmark designed to evaluate LLMs’ ability to generate CPL interoperating code. CrossPL contains two subsets: CrossPL-IPC, containing 1,982 Inter-Process Communication (IPC) tasks across six languages, and CrossPL-FFI, containing 522 Foreign Function Interface (FFI) tasks. We further present two LLM-based workflows for IPC and FFI tasks construction. Finally, we conduct a large-scale evaluation of 20 LLMs, providing **the first** systematic assessment of LLM-based CPL code generation.

Based on the reviewers’ comments, we have thoroughly addressed their major concerns as follows:

**Q1: Compared with repository-level benchmarks built on SWE-bench and its extensions, what distinguishes your benchmark and what advantages does it offer? (reviewer DQS3)**

**A1**: The original SWE-bench is Python-centric, and Multi-SWE-bench expands to seven languages but still contains only 28 CPL repair tasks out of 1,682 tasks. Since CPL interaction is our focus, our benchmark serves as a crucial complement to Multi-SWE-bench.

**Q2: Requests for greater transparency in the benchmark construction pipeline. (reviewer DQS3)**

**A2**: We added comprehensive statistics for each key filtering stage in the pipeline (Appendix B.2) and analyzed the impact of model size on the overall pipeline behavior (Discussion and Appendix C.4).

**Q3: Need for deeper analysis of decoding and reasoning configurations. (reviewer re9k)**

**A3**: We now provide a detailed examination of how temperature, top-p, and think-mode influence model performance in generating CPL interoperating code (Discussion; Appendix C.6; Appendix D.4).

**Q4: Concerns that IPC correctness relies only on FSM matching without runtime execution. (reviewer re9k, aE3T, LuTN)**

**A4**: We use static analysis for IPC tasks as IPC interactions across languages typically follow deterministic communication structures (e.g., socket connect–send–recv loops). In contrast, building a unified dynamic execution environment for heterogeneous IPC programs—spanning multiple languages, runtimes, and system configurations is prohibitively complex. Our 156 IPC FSMs are constructed from widely used implementations and official documentation. Furthermore, we perform dynamic execution testing for all FFI tasks using unit tests to ensure functional correctness.

**Q5: Concerns regarding possible dataset contamination. (reviewers cZK6, aE3T)**

**A5**: We have conducted a comprehensive and systematic examination of temporal leakage and potential training-data contamination across both IPC and FFI subsets, and the results show no evidence of contamination that would inflate model performance.

We performed an explicit time-based analysis for all IPC samples, grouping them by release year and evaluating model performance across independent temporal slices. The results—reported in the revised manuscript (Discussion; Appendix C.5)—show no evidence indicating that models perform better on older samples.This demonstrates that temporal contamination does not affect the Cross-IPC evaluation much.

For the FFI subset, we purposefully designed the construction pipeline to avoid any dependence on external code bases. All FFI tasks are self-contained: the LLM only sees a standalone C function during task generation, and all reference solutions and wrappers are produced within this controlled environment. In addition, we quantified potential overlaps with the verification codebase and found only negligible overlap. These results, presented in the revised manuscript (Discussion; Appendix C.5), confirm that FFI tasks are not meaningfully exposed to pre-training data.

**Note on reviews re9k and cZK6 (for AC’s consideration).** We appreciate all reviewers’ efforts, but we believe some concerns raised in reviews *re9k* and *cZK6* stem from misunderstandings of our problem setting and evaluation methodology, especially regarding IPC correctness and contamination. These points do not fully reflect the clarifications and analyses already provided in the main text and appendices, which we have further strengthened in the revision with experiments and data analysis. We kindly remind to take this into account when weighing these reviews against those that more closely engage with the stated scope and design of CrossPL.

---

### Meta-Review · Area_Chair_7SFo · 2026-01-08

**Summary:**

The authors introduced a new benchmark called CrossPL, focusing on evaluating cross-programming language (CPL) interaction code generation. CrossPL evaluates LLMs over two primary interoperation modes: Inter-process Communication (IPC) and Foreign Function Interface (FFI). The construction process involves multiple stages of reviewing CPL documentation, finite state machines, and review of multi-language Github repositories. The authors evaluated 20 LLMs on the benchmark and found that most of the current models perform badly.

**Reviewer Concerns:**

- There are some presentation and writing issues (too many acronyms), affecting the readability of the paper.
    - To address this, the authors updated the paper to provide the full forms of all abbreviations.
- There are missing details in the manual review, including the methodology and outcomes
    - To respond to this concern, the authors described in details the study with Hunan review for both IPC and FFI subsets
- The authors didn’t compare their work to prior released benchmarks such as SWE-Bench as repository-level benchmarks can naturally contain multi-programming language tasks
    - In their rebuttal, the authors explained that SWE-Benchmark and Multi-SWEBench contain only a small portion that involves cross-language interaction.
- The proposed benchmark is limited to Python-C tasks and the findings here might not generalise well to other cross-language tasks
- The validation of IPC tasks is mainly based on static analysis tool, and might not reflect different desired qualities e.g. latency, functional correctness
    - The authors partially addressed this concern by explaining the static analysis tool and their reliability with manual inspection
- There are some risks of data contamination as some samples might be leaked to LLMs during training
    - The authors conducted temporal analysis and reported results based on samples by their release years. The results are convincing to me and show no significantly leakage in the current major LLMs

**Reviewer Scores:**

- Reviewer DQS3 might increase their score from 6 to 7 given the substantial responses from the authors to address the reviewer concerns
- Reviewer LuTN would keep the same positive score of 6 as their concerns still persist
- Reviewer aE3T might keep the same positive score of 6
- Reviewer re9k might keep the same score of 4 given their main concerns still persist
- Reviewer cZk6 might keep the same score of 4 as their concerns are only partially addressed

---

### Decision · Program_Chairs · 2026-01-26

Accept (Poster)